# The application of dendrometers to alpine dwarf shrubs – a case study to investigate stem growth responses to environmental conditions

Svenja Dobbert[1], Roland Pape[2], Jörg Löffler [1]

[1]Department of Geography, University of Bonn, Meckenheimer Allee 166, D‑53115 Bonn, Germany

[2]Department of Natural Sciences and Environmental Health, University of South-Eastern Norway, Gullbringvegen 36, N-3800 Bø, Norway

*Correspondence to*: Jörg Löffler (joerg.loeffler@uni‑bonn.de)

**Abstract.** Considering the recent widespread greening and browning trends associated with shrubs in arctic‑alpine ecosystems, further understanding of how these shrubs respond to rapidly changing environmental conditions is of crucial importance. We here monitor shrub growth, using high-precision dendrometers, which have not been used on shrubs before.

We present the first fine‑scale intra-annual growth patterns derived from hourly radial stem variability of a common evergreen species (*Empetrum nigrum* ssp. *hermaphroditum*) at a micrometer scale for the period 2015 till 2018 on exposed and mostly snow-free ridge positions. With the same temporal resolution, we collected near-ground environmental data and identified on‑site controls of growth behavior.

We found high inter-plant variability in growth-defining parameters, but strong similarities in response to the local environment. Radial growth starting in spring and early summer exhibited high sensitivity to winter temperatures and prolonged ground freezing, suggesting that the evergreen species remains partly photosynthetically active during winter, which facilitates carbohydrate accumulation for early-season physiological activities. We discovered a phase of radial stem shrinkage during the winter months, which can be attributed to an active cell water reduction to protect the plant from frost damage.

We identified soil moisture availability and winter freezing conditions as the main drivers of radial stem variability, thus forwarding the ongoing debate on the functional mechanisms of greening and browning in arctic and alpine regions.

## 1 Introduction

Arctic and alpine ecosystems are especially sensitive to recent climate variability, with temperatures increasing thrice as much as the global average in the past decades, caused by a rising concentration of atmospheric $CO_2$ and accompanied by a

substantial lengthening of the growing period (e.g., Stocker, 2013; Post et al., 2019; AMAP, 2021). This trend has favored growth, abundance, and biomass production of numerous shrub species, resulting in a widespread, yet spatially heterogenic, greening of the affected areas - with potentially global effects (Myers-Smith et al., 2011; Gough et al., 2015; Brodie et al., 2019; Myers-Smith et al., 2020). The observed greening has been verified using remote sensing techniques (e.g., Carlson et al., 2017) and is caused by both evergreen and broadleaved species, although in different ways (Vowles and Björk, 2019; Weijers and Löffler, 2020). In general, shrubs are considered one of the most responsive plant functional groups to climate variability (Elmendorf et al., 2012). Their expanding trend, in turn, has been associated with climatic feedbacks, such as influence on surface albedo and frozen-ground processes (Sturm et al, 2001; Chapin et al., 2005; Blok et al., 2011; Aartsma et al., 2021). Therefore, an understanding of shrub growth-physiology and its environmental controls is of crucial importance.

Over the past decade, dendroecological studies have identified temperatures and soil moisture as the most important drivers in controlling shrub growth, with the strongest effects caused by conditions within the current year (Van der Wal and Stien, 2014), especially during the growing season (Elmendorf et al., 2012; Hollesen et al., 2015; Ackerman et al., 2017; Weijers et al., 2017). Additionally, most recent studies have suggested that snow cover and winter warming may play an important role in promoting shrub growth (Hollesen et al., 2015; Weijers et al. 2018a; Francon et al., 2020), as well as spring warming (Weijers et al., 2018a). Yet, an increased frequency of spring freezing events might counteract these positive effects (Choler, 2018). Collectively, these studies agree on the fine-scale complexity of growth behavior, niche shifts, and local adaptation of shrubs in arctic and alpine regions, with a multitude of still little-understood, site-related environmental drivers (Graae et al., 2017; Pape and Löffler, 2017; Löffler and Pape, 2020).

High-resolution data as provided by dendrometer measurements have the potential to bridge this knowledge-gap and to provide valuable insights into fine-scale response mechanisms towards a changing environment. In tree physiology and forest sciences, dendrometers have proven useful to monitor tree responses to environmental fluctuations (Breitsprecher and Bethel, 1990; Duchesne et al., 2012; Ježík et al., 2016; Van der Maaten et al., 2018; Smiljanic and Wilmking, 2018), because recent dendrometers can detect radial stem dimensions at hourly or even shorter intervals (Drew and Downes, 2009; Liu et al., 2018). Starting with the early designs, first described in the 1930s and the 1940s (Reineke, 1932; Daubenmire, 1945), dendrometers have been widely used, focusing on long-term monitoring of growth responses to environmental variables (e.g., Duchesne et al., 2012; Liu et al., 2018; Van der Maaten et al., 2018). Recently, a first study using band-dendrometers to monitor radial stem growth of tree-like shrubs was presented (González-Rodríguez et al., 2017). Because current dendrometers are designed to measure at a micrometer scale, they have the potential to be used on shrubs to provide fine-scale, intra-annual, continuous, and highly comparable information. Past studies of radial growth in shrubs, the first of which were published in 2006 and 2007 (Bär et al., 2006; Bär et al., 2007), had to rely solely on shrub-ring series (Macias-Fauria,

et al., 2012; Shetti, 2018; Le Moullec et al., 2019), extraction of micro-cores, or wood anatomical analyses (Rossi et al., 2006; Weijers et al., 2010; Liang et al., 2012; Francon et al., 2020). Because trees and shrubs are both woody plants, transferring methodical approaches from one to the other has proven successful in the past (e.g., Bär et al., 2006; Liang et al., 2012; Macias-Fauria et al., 2012). Yet, these approaches may have to be adapted for the special morphology, growth behavior and high internal variability of multi-stemmed shrubs (Bär et al., 2007; Buras and Wilmking, 2014; Myers-Smith et al., 2015). Because cambial activity occurs at time scales ranging from hours to days (Deslauriers et al., 2007; Köcher et al., 2012; Liu et al., 2018), the fine temporal resolution that is gained by dendrometers has the potential to provide valuable additional insights compared to traditional methods. These include intra-annual and seasonal growth behavior of shrubs in alpine environments, thereby bridging existing knowledge gaps regarding plant productivity in remote ecosystems (Le Moullec et al., 2019).

Additionally, the time series derived from dendrometer measurements offer information not only on radial stem gro wth, but also on stem water relationships with higher quality and resolution than previously attainable (Fritts, 1976; Steppe et al., 2015; Zweifel, 2016; González-Rodríguez et al., 2017). Stem diameter increase can be a result of both swelling tissue and growth, and both can contain valuable information on shrub responses to external factors and cambial activity. Thus, it can be considered one of the strengths of this approach that these factors are both visible in the data and can be simultaneously assessed (Drew and Downes, 2009; Chan et al., 2016; Zweifel, 2016). Monitoring these intra-annual patterns could provide the opportunity to study fine-scale, eco-physiological mechanisms for the first time in shrubs. What has become a widely used practice in dendroecological studies on trees (Deslauriers et al., 2007), has the potential to become available for shrubs as well.

In this context, we monitored intra- and inter-annual variability of stem radial variations in an alpine environment, testing this novel approach using high-precision dendrometer data derived from dwarf shrubs to 1) explain major growth patterns and their variation between years and specimens, 2) identify the most important on-site environmental drivers controlling these patterns, and 3) gain insights into potential response to environmental change. The main objective of our work is thus to gain detailed understanding of the growth patterns of one common arctic-alpine dwarf shrub (*Empetrum nigrum* ssp. *hermaphroditum)* and their relation to the micro-environment. With this, we hope to bridge the gap between observed large-scale vegetational shifts, and the fine-scale physiological mechanisms driving these complex changes within the highly relevant arctic-alpine ecosystems.

 **2 Material and Methods**

**2.1 Study sites**

We conducted our study in two alpine mountain regions of central Norway. To the west, the Geiranger/Møre og Romsdal region (62°03′N; 7°15′E) is located within the slightly to markedly oceanic climatic section (O1–O2; Moen, 1999) of the inner fjords. It is characterized by humid conditions, with total annual precipitation of 1,500–2,000 mm in the valleys (Aune, 95    1993) and a mean annual ambient air temperature of 1.9 °C (range: -23.2 °C–17.2 °C) (Löffler, 2003). To the east, the Vågåmo/Innlandet region (61°53′N; 9°15′E) is located within the continental climatic section (C1; Moen, 1999). The total annual precipitation is low, approximately 300–500 mm in the valleys (Kleiven, 1959) and the mean annual ambient air temperature is -1.2 °C (range: -29.2 °C–16.7 °C) (Löffler, 2003). Our own measurements in the alpine parts of the studied regions indicated that the annual liquid precipitation was 900 mm in the west and 375 mm in the east. The additional amount 100  of snow and its water equivalent remains unknown, but snowdrift leads to an uneven distribution of the snowpack within the complex alpine topography (Löffler, 2007).

Across both regions, we used exposed alpine ridge sites that were - within the framework of our long-term alpine ecosystem research project (LTAER; e.g., Löffler and Finch, 2005; Hein et al., 2014; Frindte et al., 2019; Löffler et al., 2021), 105  stratified-randomly chosen along the elevational gradient. The elevational gradient was stratified into six elevational bands from the tree line upwards, shifted by 100 m between regions to account for slightly different conditions and a diverging position of the tree line. In the oceanic region, we used 900, 1000, 1100, 1200, 1300, and 1400 m a.s.l. (above sea level), in accordance with the tree line in this region, which is located at about 750 to 800 m a.s.l. In the continental region, we used 1000, 1100, 1200, 1300, 1400, and 1500 m a.s.l. Here, the tree line is situated slightly higher, at about 1000 m a.s.l. (Rößler 110  et al., 2008; Rößler and Löffler, 2007). Thus, all of our studied sites were located above the tree line. As study sites, we chose micro-topographical positions at wind-blown ridges, likely representing the most extreme thermal regimes, with discontinuous snow cover and deeply frozen ground during winter. Our study design resulted in a total of two regions × six elevational bands = 12 sites, with one specimen monitored per site (N = 12), resulting in 12 × four years = 48 annual dendrometer curves. A summary of total stem diameter variation and environmental conditions measured at each site is 115  presented in Fig. A1.

**2.2 Species**

We focused on the shrub species *Empetrum nigrum* ssp. *hermaphroditum* (hereafter *E. hermaphroditum*), an evergreen shrub almost circumpolar in distribution (Bell and Tallis, 1973) and abundant in the Scandes mountain chain. *E. hermaphroditum* has been identified as a niche constructor species with strong direct effects on tundra communities, including a potential 120  slowing of process rates and lowering of biodiversity with *E. hermaphroditum* encroachment (Bråthen et al., 2018). Because

of its complex response to variation in snow cover, it is most common at positions with either shallow or relatively deep snow cover (Bienau et al., 2014; Bienau et al., 2016). Additionally, *E. hermaphroditum* is comparatively resistant to low winter temperatures (Stushnoff and Junttila, 1986; Ogren, 2001) and usually not affected by grazing (Weijers and Löffler, 2020). Löffler and Pape (2020) found its occurrence promoted by temperatures of >15.5 °C in the shoot zone and >0.7 °C in the root zone. Generally, the species occurs in a wide phytogeographic range at various sites along the alpine elevational gradient. Its frequency is high at different micro-topographic positions and at high elevations, the species occurs as exclusive shrub within a matrix of debris and graminoids. The species belongs to the Empetraceae family of heathlike shrubs. Its stem anatomy was described by Carlquist in 1989 and is characterized by a narrow vessel diameter, which can be interpreted as a form of adaptation to drought or physiological drought due to cold, as it impedes embolism formation (Fig. A2). In general, the family is known to match extreme environments by adapting stem anatomy (Carlquist, 1989). *E. hermaphroditum* was the first shrub species for which a chronology was successfully derived using its annual growth rings (Bär et al., 2007).

## 2.3 Dendrometric data and monitoring setup

Here, we applied a technological approach, commonly used for trees, to our multi-stemmed specimens of *E. hermaphroditum*, taking radial stem measurements using dendrometers. The general idea was to apply well-established methods from dendroecology and tree growth analysis within a novel setting, to assess intra- and inter-annual variation in growth patterns and its environmental controls for a shrub species. We mounted our dendrometers on one major above-ground stem of a randomly chosen specimen per site, horizontal to the ground surface and as close to the assumed root collar as possible. During this process, we removed the dead outer bark (periderm) to place the sensor as close to the living tissue as possible, following a common practice for dendrometer measurements of trees (Oberhuber et al., 2020; Wang et al., 2020; Grams et al., 2021). This ensures that hygroscopic shrinkage and swelling of dead tissues from the outer bark does not influence the diameter measurements. Such processes have been previously addressed in trees (Zweifel and Häsler, 2000; Gall et al., 2002; Ilek et al., 2016), and comparative studies revealed a complex interplay of xylem as well as phloem growth and pressure induced size changes, which simultaneously affect radial stem change and are thus captured by the dendrometers (Turcotte et al., 2011; Zweifel et al., 2014b; Oberhuber et al., 2020) (Fig. A2). Additionally, we avoided specific micro-positions near stones and depressions, inside the radius of other larger shrub species, and near patches of wind erosion (Fig. A3). Stem diameter data were measured at 1 min intervals using dendrometers (type DRO; Ecomatik, Dachau/Germany). The sensor has a temperature coefficient of <0.2 μm/K. To ensure that the dendrometers produce meaningful data, unaffected by the mounting process and bark removal, we tested the study design for several years, before presenting the final study period here. Based on these data, we calculated daily stem diameter variation as the mean stem diameter measured each day, following the "daily mean approach" (Deslauriers et al., 2007), which represents a combination of water- and growth-induced radius expansion (Oberhuber and Gruber, 2010). For the calculations, we used the

'dendrometeR' package (Van der Maaten et al., 2016), developed for the R statistical software (R Development Core Team, 2020).

**2.4 Analysis of seasonal growth patterns**

To assess growth variation and patterns, we defined specific parameters and dates of annual stem growth, such as a) total growth, defined as growth-induced stem expansion, b) peak growth (maximum daily growth rate), c) growth initiation (start of the growing season), d) growth cessation (end of the growing season), and e) peak shrinking during the winter months. There are several methods currently available to separate growth from water-related expansion and contraction of the stem. To define total annual growth, we chose the approach proposed by Zweifel (2016), which excludes reversible shrinking and

swelling associated with stem water fluctuations, assuming zero growth during periods of stem shrinkage. Total annual growth is, thus, derived from the original measured data by calculating the cumulative maxima *(*Zweifel et al., 2014a; Zweifel, 2016). We recorded additional stem increment before the start of the growing season in spring. Because this increment did not exceed the previous year's maximum stem diameter, we assumed that it might be related to refilling processes rather than formation of new xylem and cambial growth (see Mayr et al., 2006) and therefore, in accordance with

the previously described zero growth assumption, did not define those processes as growth. Following this approach, we were able to define growth-induced stem increment during the main growing season. To further define this growing season and derive accurate dates for growth start and end, we fitted sigmoid Gompertz models to the resulting growth curves. Although multiple models have been used to describe growth, the Gompertz equation is the most widely used in dendrochronological studies and has been proven to explain the variations in dendrometer measurements (for trees) well

(e.g., Rossi et al., 2003; Rossi et al., 2006; Duchesne et al., 2012; Van der Maaten et al., 2018; Liu et al., 2019). The equation used for the model was Eq (1):

$$y(t) = \alpha \times \exp(-\beta \times \exp(-k \times t),$$

(1)

where alpha is the upper asymptote, beta is the x-axis placement parameter, and k is the growth rate. We calculated these

175 input parameters from our original data using the equations defined by Fekedulegn et al. (1999). To assess how well the models fit our data, we calculated a goodness-of-fit (GoF) measure using the least-squares method with the formula Eq (2):

$$GoF = 1 - (\Sigma(f - f2)^2/\Sigma f^2$$

(2)

where f is the original and f2 is the modeled stem diameter.

We determined growth initiation (onset) and cessation (offset) from this modeled curve to ensure that the main growth phase was being captured. Both dates were defined as the time when 20% and 90% of the total annual modeled growth occurred, respectively. We chose these thresholds following careful testing and, in accordance with our data, they were slightly higher

than the thresholds used in similar studies for trees (e.g., Van der Maaten-Theunissen et al., 2013; Van der Maaten et al., 2018; Drew and Downes, 2018). We calculated these growth-defining parameters for each individual curve to assess the variability between sampled specimens. These values entered into the following statistical analysis individually.

Some of our specimens did not experience any growth or irreversible stem expansion in specific years. Years with no or little growth have been detected in other shrubs, for example, in *Salix arctica* by Polunin (1955). Buchwal et al. (2013) assumed such mechanisms to be related to carbon allocation and to occur irregularly along the stem because growth is not homogenously allocated within the different plant segments. In accordance with their findings for *Salix polaris*, the specimens might preferentially allocate resources to less exposed parts (e.g., roots) in these years. In general, such partial dormancy (Preece et al., 2012) might reflect insufficient resources for homogenous growth across the entire plant. We did not calculate a growing season for these years, and the analyses proceeded separately, excluding them from most of our calculations.

## 2.5 Environmental data collection and growth conditions

To identify the thermal constraints of our species at both, the critical location and time scale of action (cf. Körner and Hiltbrunner, 2018), we measured soil temperatures (°C) at a depth of 15 cm below the ground surface (i.e., within the root zone: hereafter "$T_{RZ}$") and air temperatures 15 cm above the ground surface (i.e., within the shoot zone: hereafter "$T_{SZ}$"), at all sites. Temperatures were measured at 1 min intervals and recorded as hourly means using ONSET's HOBO loggers (type H21-002) and type S-TMB-002 temperature sensors (±0.2 °C accuracy). For the $T_{SZ}$ measurements, the sensors were equipped with passively ventilated radiation shields. Moreover, to identify the soil moisture constraints in the root zone of our specimens, we measured the volumetric soil water content (m³/m³) 15 cm below the soil surface (hereafter $SM_{RZ}$) at all sites. The uncalibrated $SM_{RZ}$ was measured at 1 min intervals and recorded as hourly means using ONSET's HOBO type S-SMD-M005 soil moisture sensors (±3% accuracy). Additionally, we measured the shoot zone global radiation (W/m²) at 1 cm above the ground surface in close proximity to the plant (hereafter $GR_{SZ}$) using ONSET's HOBO type S-LIB-M003 silicon pyranometers (±10 W/m² accuracy). We made sure that those measurements were not affected by the canopy. Our data covered a period of four full calendar years from January 1, 2015, to December 31, 2018. Missing data did not occur at the chosen sites. The different near-surface regimes of $T_{RZ}$, $T_{SZ}$, $SM_{RZ}$, and $GR_{SZ}$ at our micro-topographical sites are illustrated in Fig. 1. From these raw data, we calculated a set of variables defining the micro-environmental conditions for each specimen based on the expected effects on different growth mechanisms. Averaged values for all sites (N=12) are summarized in Table 1.

**Table 1: Summary of collected environmental data: Annual mean, minimum and maximum values as well as mean values for each meteorological season. Numbers in parentheses represent site variability (standard deviation, +/- SE).**

| Environmental parameter | Year | Mean | Min | Max | Mean spring | Mean summer | Mean autumn | Mean winter |
|---|---|---|---|---|---|---|---|---|
| Shoot zone temperature [°C] | 2015 | 0.10 (0.34) | -13.66 (0.70) | 21.24 (0.59) | -2.44 (0.34) | 6.95 (0.36) | 1.29 (0.37) | -5.50 (0.43) |
| | 2016 | 0.13 (0.30) | -19.53 (0.74) | 21.10 (0.46) | -1.39 (0.33) | 8.40 (0.33) | 0.30 (0.30) | -6.86 (0.34) |
| | 2017 | 0.04 (0.37) | -15.70 (0.40) | 21.97 (0.40) | -1.68 (0.35) | 7.28 (0.34) | -0.07 (0.45) | -5.48 (0.48) |
| | 2018 | 0.55 (0.28) | -19.373 (1.19) | 25.156 (0.42) | -1.071 (0.28) | 10.029 (0.33) | 0.214 (0.32) | -7.782 (0.51) |
| Root zone temperature [°C] | 2015 | 0.98 (0.33) | -7.01 (0.56) | 14.27 (0.33) | -1.80 (0.31) | 7.04 (0.39) | 2.48 (0.32) | -3.88 (0.37) |
| | 2016 | 0.96 (0.29) | -13.16 (0.58) | 14.37 (0.42) | -1.48 (0.31) | 8.76 (0.34) | 2.03 (0.24) | -5.52 (0.36) |
| | 2017 | 0.93 (0.32) | -9.76 (0.52) | 13.57 (0.43) | -1.67 (0.28) | 7.71 (0.31) | 1.87 (0.32) | -4.30 (0.49) |
| | 2018 | 1.31 (0.25) | -11.03 (0.65) | 16.79 (0.48) | -1.76 (0.25) | 10.45 (0.34) | 1.67 (0.23) | -5.27 (0.47) |
| Soil moisture [m³/m³] | 2015 | 0.15 (0.01) | 0.05 (0.01) | 0.28 (0.02) | 0.09 (0.03) | 0.23 (0.04) | 0.15 (0.17) | 0.06 (0.05) |
| | 2016 | 0.14 (0.01) | 0.06 (0.01) | 0.27 (0.02) | 0.12 (0.03) | 0.23 (0.04) | 0.12 (0.15) | 0.06 (0.03) |
| | 2017 | 0.15 (0.01) | 0.05 (0.01) | 0.27 (0.01) | 0.10 (0.02) | 0.24 (0.03) | 0.14 (0.15) | 0.07 (0.05) |
| | 2018 | 0.15 (0.01) | 0.05 (0.00) | 0.312 (0.01) | 0.13 (0.03) | 0.21 (0.04) | 0.14 (0.15) | 0.06 (0.03) |
| Global radiation [W/m²] | 2015 | 73.11 (3.90) | 0 | 809.34 (24.50) | 74.38 (11.49) | 169.70 (4.32) | 41.82 (1.85) | 4.72 (1.10) |
| | 2016 | 75.80 (3.10) | 0 | 823.93 (28.48) | 93.94 (9.32) | 165.04 (5.49) | 39.51 (2.70) | 3.53 (0.78) |
| | 2017 | 70.86 (4.34) | 0 | 791.81 (37.21) | 78.45 (12.55) | 166.59 (5.38) | 32.27 (1.30) | 4.26 (0.94) |
| | 2018 | 85.57 (3.60) | 0 | 846.38 (46.05) | 117.15 (7.73) | 190.84 (7.12) | 28.59 (1.93) | 3.25 (0.80) |

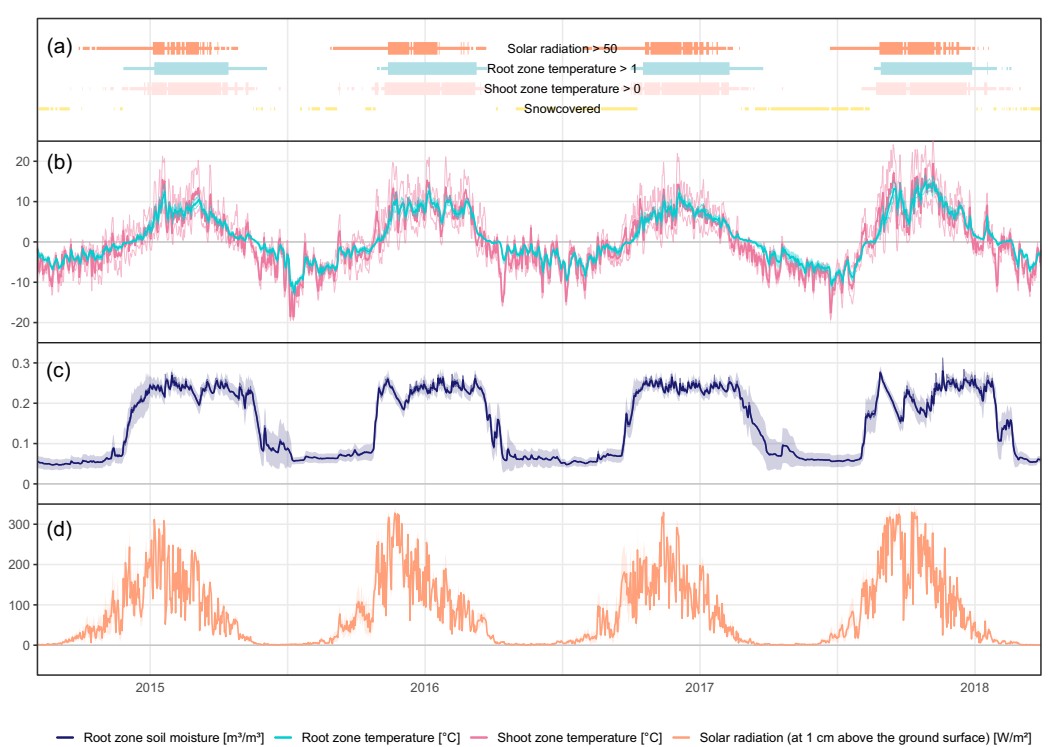

**Figure 1: Micro-environmental data.** Daily mean values of shoot and root zone temperatures (b), soil moisture (c), and global radiation (d). Measurements were taken at each site individually, but averaged here over all sites for visualization. Shaded areas indicate standard deviations. Bars (a) show time spans for certain environmental conditions, with narrow bars marking the time at which the given condition was present at least at one of the monitored sites, whereas broad bars mark the time for which the given condition was detected at all sites.

In all four monitored years, our sampled specimens experienced the highest temperatures during summer (Fig. 2). In 2018, the highest mean temperatures were expressed because of exceptionally high summer temperatures (number of days with TSZ > 10, growing degree days, GDD10 = 57), whereas 2015 and 2017 both experienced shorter periods of high temperatures. Temperatures varied slightly among sites (Table 1 and Fig. A1), with a noticeable but not linear temperature decline with elevation, and a slightly lower difference between the lowest and highest study sites than proposed by the linearly regressed adiabatic lapse rate (Löffler et al. 2006). The reasons are daily variations in the adiabatic lapse rate induced by nocturnal inversions, and variations over specific weather phenomena and seasonality (e.g. Löffler et al. 2006; Pape et al. 2009; Wundram et al. 2010). As expected, the shoot and root zone temperature curves were well coupled. Additionally, we detected slight variability between the two studied regions, but overall similar seasonal temperature patterns on the measured micro-scale (Fig. 2 and A1), which differs from the regional climate signal. This is because these measured temperature signals reflect local, near-ground conditions, which are decoupled from the regional signal. Global radiation showed similar patterns at all sites as well, following the course of the astronomic sun angle, with a mid-summer maximum; however, there were large variations according to cloud coverage. As such, 2018 experienced a short period of temperature and radiation decrease during summer (Fig. 1).

We did not explicitly measure data regarding snow cover, but calculated snow cover from the daily shoot zone temperature amplitude and validated those calculations using radiation sensor measurements. We assumed that a daily amplitude of less than 5% of the maximum amplitude reached throughout the year indicated that a layer of snow restricted daily air temperature fluctuations at the measured height of 15 cm. The respective periods were therefore defined as snow-covered. The period in which our specimens were snow-covered varied considerably between the monitored winters and between sites. The winter of 2015/2016 had comparatively little snow, whereas 2017/2018 was snow-covered for the longest period. However, because of the chosen positions on wind-blown ridges, most of our monitored sites did not experience long periods of snow cover. Nonetheless, snow and its presumed effects, such as mitigating extreme negative temperatures by acting as an isolating barrier and, hence, reducing the effects of frost (Körner, 2021; Bienau et al., 2014) might play a role in influencing the growth response and was therefore included in our analysis.

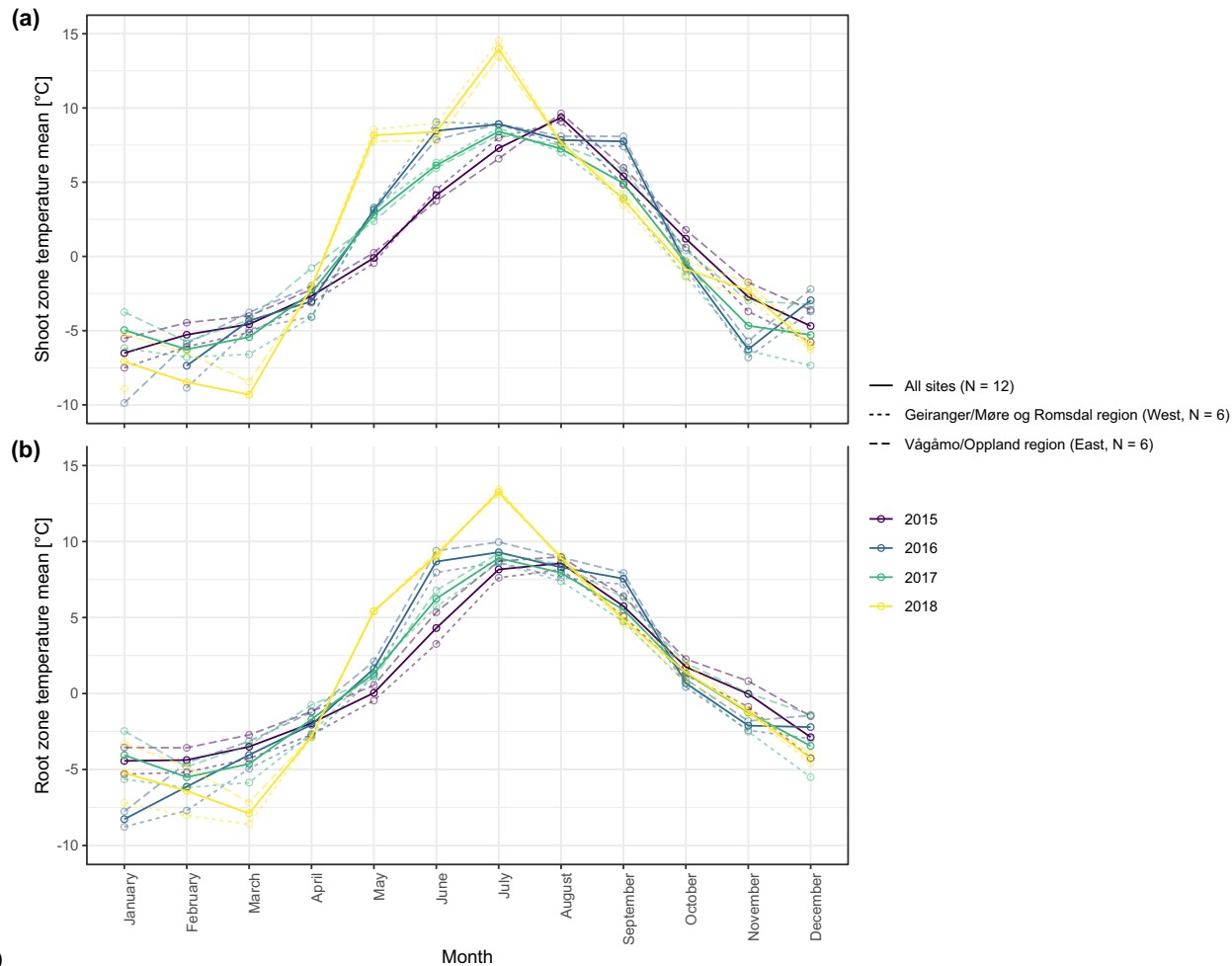

**Figure 2: Mean monthly shoot (a) and root (b) zone temperatures from 2015 to 2018, for visualization purposes averaged i), over the measured sites per study region, and ii), over all measured sites.**

## 2.6 Potential micro-environmental drivers of seasonal growth patterns

To analyze the effect of the site-specific environmental conditions on the observed growth patterns, we performed a set of multiple and partial regressions alongside correlation analyses, calculating partial R squared and Pearson's correlation coefficients. Regressions were performed in R (R Development Core Team, 2020) and tested for multicollinearity among the independent variables that would have affected the regression outcome using several measures of collinearity, which are implemented in the mctest package (Imdad and Aslam, 2018). This included the determinant of the correlation matrix (Cooley and Lohnes, 1971), Farrar test of chi-square for presence of multicollinearity (Farrar and Glauber, 1967), Red

indicator (Kovàcs et al., 2005), Sum of lambda inverse (Chatterjee and Price, 1977) values, Theil's indicator (Theil, 1971), and condition number (Belsley et al., 1980). A regression model was discarded if more than two of the six calculated measures indicated collinearity among the independent variables. For these, as well as the following statistical analysis, we used site-specific values for each sampled specimen to overcome common misconceptions regarding the utilization of, e.g., temperature data (cf. Körner and Hiltbrunner, 2018), We carefully chose physiologically meaningful variables without averaging, specifically to enable a meaningful analysis (see also Löffler and Pape, 2020).

After testing our chosen growth parameters and their relationship to total annual growth by performing a partial regression analysis (Table 2), we analyzed the environment-growth relationship by correlating the growth parameters (total annual growth, growth initiation, peak growth, growth cessation, start of the shrinking period, peak shrinking, and day of year (DOY) at which peak shrinking occurred) with the potential micro-environmental drivers listed in Table 3, using Pearson's correlation coefficient. For each group of potential drivers (means, maxima, days at which the maxima were reached for shoot and root zone temperatures, global radiation, and soil moisture), we performed a multiple regression analysis, predicting each of the growth parameters from a fitted linear regression model. Additionally, we correlated our raw environmental data with the daily stem change for each season, performing rolling correlations by aggregating our hourly data and calculating moving averages for preceding periods, with length ranging from days to years. Here, we chose to include the complete dendrometer series, rather than reducing the data to the main growing season (Deslauriers et al., 2007), to capture seasonal stem shrinking and swelling throughout the year.

**Table 2: Results of multiple partial regression analysis for total annual growth (measured cumulative stem diameter increment in comparison to the previous year's maximum stem diameter) and annual peak shrinking against growth parameters (growth initiation, growth cessation, and peak growth).**

| Variable | Independent variable | Pearson's correlation coefficient | Partial $R^2$ | Semipartial (Part) correlation | Coefficient | Standard Error | T-statistic | p-value |
|---|---|---|---|---|---|---|---|---|
| Total annual growth | Growth initiation | -0.169 | -0.399 | -0.162 | -0.349 | 0.144 | -2.423 | 0.021 |
| | Growth cessation | -0.245 | 0.448 | 0.187 | 0.428 | 0.153 | 2.793 | 0.009 |
| | Peak growth | 0.909 | 0.922 | 0.890 | 25.228 | 1.900 | 13.278 | <0.001 |
| | *Total* | | *0.930* | | | | | *<0.001* |
| Annual peak shrinking | Growth initiation | 0.133 | 0.144 | 0.129 | 0.209 | 0.259 | 0.808 | 0.425 |
| | Growth cessation | 0.016 | -0.037 | -0.033 | -0.057 | 0.275 | -0.057 | 0.838 |
| | Peak growth | 0.402 | 0.395 | 0.383 | 8.167 | 3.411 | 2.394 | 0.023 |
| | *Total* | | *0.790* | | | | | *<0.001* |

**Table 3: Summary of calculated micro-environmental parameters (means). Numbers in parentheses represent inter-stem variability (standard deviation, +/- SE).**

| Parameter | 2015 | 2016 | 2017 | 2018 |
|---|---|---|---|---|
| ***Shoot zone temperature ($T_{SZ}$)*** | | | | |
| Day, when maximum $T_{SZ}$ is reached [DOY] | 183 (5.36) | 203 (7.41) | 203 (0.29) | 208 (0) |
| First day, at which $T_{SZ} > 0$ [DOY] | 100 (9.24) | 74 (7.28) | 25 (12.41) | 104 (0.56) |
| Annual $T_{SZ}$ sum [°C] | 37.28 (125.36) | 47.96 (109.48) | 16.21 (133.57) | 199.96 (102.65) |
| $T_{SZ}$ sum at growth initiation [°C] | -291.87 (57.03) | -421.91 (134.73) | -189.45 (96.27) | -384.76 (139.80) |

| | | | | |
|---|---|---|---|---|
| | (1155.10) | (1678.66) | (1929.40) | (2882.40) |
| *Snow* | | | | |
| Number of snow-free days [days] | 361 (4.90) | 361 (5.53) | 358 (13.79) | 356 (9.48) |
| First autumn frost ($T_{RZ}$ <0°C) [DOY] | 320 (3.60) | 296 (2.84) | 310 (4.58) | 301 (4.70) |

## 3 Results

### 3.1 Intra-annual growth patterns

In general, the seasonal variability in stem diameter was well explained by non-linear, sigmoid regressions (Gompertz curves) with a GoF between 0.90 and 0.94 (Fig. 3), and all specimens experienced distinct growing seasons. Moreover, our data revealed a distinct phase of stem radial contraction (shrinking) following the growing phase towards the end of the year, starting in October, with remarkably little variation in timing between years (on average from the 287[th] to the 311[th] Julian day). In most cases, the stem radius remained below the previously achieved maximum for the entire winter and started to increase again with the following year´s growing season (Fig. 3). The timing of this shrinking period was significantly linked to the day when peak growth occurred (R = 0.50, p = 0.004), as well as to growth initiation (R = 0.40, p = 0.023) and cessation (R = 0.52, p = 0.0023).

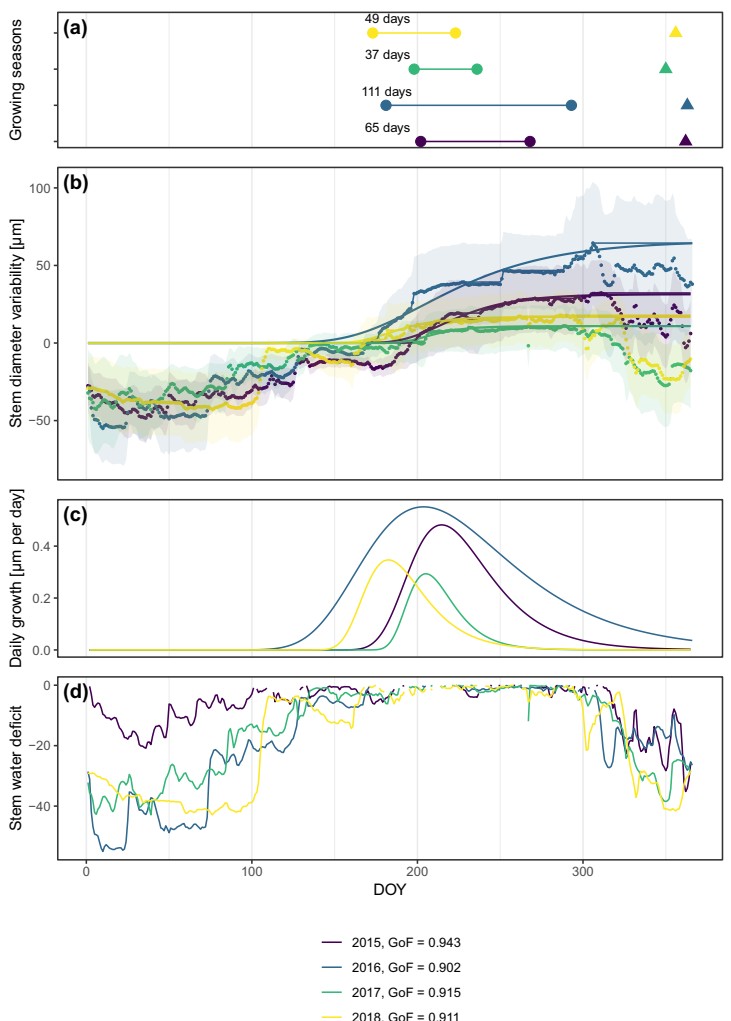

**Figure 3: Observed and modelled variability in stem diameter. (a) Growing seasons derived from fitted Gompertz models and timing of peak shrinking defined as the minimum of the observed shrinking phase toward the end of the year (triangles). (b) Averaged measured daily stem diameter variability and fitted Gompertz models (a goodness-of-fit (GoF) measure was calculated using the least-squares method). Models were fitted to zero growth curves derived from the original measurements as cumulative maxima (thin lines), assuming zero growth during phases of prolonged stem shrinkage. In this way annual growth and, consequently, growth start is directly linked to growth during the previous year and additional rehydrating processes before the start of the main growing season are excluded. Shaded areas indicate standard deviation showing the variability among the specimens. (c) Daily growth derived from Gompertz models. (d) Stem water deficit defined as reversible radial stem variability at times when no growth-induced irreversible stem expansion was measured.**

| | | | | |
|---|---|---|---|---|
| Number of days with $T_{SZ}$ >0 (Growing degree days, GDD0) [days] | 175 (6.91) | 169 (4.48) | 165 (5.09) | 173 (5.38) |
| Number of days with $T_{SZ}$ >5 (Growing degree days, GDD5) [days] | 82 (5.50) | 117 (3.93) | 100 (6.74) | 121 (3.73) |
| Number of days with $T_{SZ}$ >10 (Growing degree days, GDD10) [days] | 21 (2.53) | 32 (4.57) | 12 (2.80) | 57 (3.27) |

**Root zone temperature ($T_{RZ}$)**

| | | | | |
|---|---|---|---|---|
| Day, when maximum $T_{RZ}$ is reached [DOY] | 186 (0.08) | 206 (0) | 204 (0.11) | 197 (1.6) |
| First day, at which $T_{RZ}$ >1 [DOY] | 146 (3.95) | 140 (1.73) | 139 (1.43) | 127 (1) |
| Annual $T_{RZ}$ sum [°C] | 358.52 (121.54) | 351.23 (104.54) | 339.15 (117.34) | 477.58 (90.09) |
| $T_{RZ}$ sum at growth initiation [°C] | -127.05 (79.27) | -326.05 (140.50) | -116.97 (90.72) | -303.43 (149.08) |
| Number of days with $T_{RZ}$ >0 (Growing degree days, GDD0) [days] | 182 (8.01) | 170 (4.19) | 176 (5.78) | 196 (5.99) |
| Number of days with $T_{RZ}$ >5 (Growing degree days, GDD5) [days] | 96 (5.75) | 124 (2.14) | 117 (5.27) | 119 (2.74) |
| Number of days with $T_{RZ}$ >10 (Growing degree days, GDD10) [days] | 7 (3.70) | 14 (7.48) | 10 (3.68) | 57 (4.10) |

**Soil moisture**

| | | | | |
|---|---|---|---|---|
| Day, when maximum soil moisture is reached [DOY] | 197 (21.28) | 219 (17.38) | 193 (18.85) | 222 (22.02) |
| Day, when minimum soil moisture is reached in autumn [DOY] | 351 (3.25) | 342 (3.16) | 351 (2.32) | 353 (2.88) |
| Annual soil moisture sum [m³/m³] | 54.65 (4.09) | 52.78 (3.53) | 55.52 (3.46) | 54.51 (3.47) |
| First day, at which soil moisture >0.15 [DOY] | 135 (3.37) | 128 (2.53) | 126 (6.51) | 110 (3.28) |

**Global radiation**

| | | | | |
|---|---|---|---|---|
| Day, when maximum global radiation is reached [DOY] | 171 (4.53) | 156 (1.12) | 181 (3.04) | 177 (4.55) |
| First day, when global radiation >50 W/m² [DOY] | 74 (9.62) | 75 (7.93) | 85 (6.71) | 76 (6.50) |
| Annual global radiation sum [W/m²] | 26686.66 (1423.77) | 27742.87 (1133.43) | 25863.3 (1584.93) | 31229 (1313.83) |
| Global radiation sum at growth initiation [W/m²] | 16342.47 | 15435.76 | 16197.58 | 15928.75 |

Total annual growth ranged from 11 µm in 2017 to 65 µm in 2016, on average, with the growing season starting in May or June (Table 4 and Fig. 3). Interestingly, year-to-year variability and patterns in total growth were similar in the two studied regions (Fig. A4). Furthermore, while our data showed slight differences in seasonal growth patterns between the two regions, there were no clear overall patterns beyond the high inter-specimen variability observed in the whole dataset (Fig. A5). Our chosen growth parameters, growth initiation, peak growth, and growth cessation, together explained 93% of the variance in total annual growth, with peak growth having by far the greatest influence. For stem contraction (shrinking), the same parameters explained 79% of the variance (Table 2 and Fig. A6), suggesting that the observed winter shrinking in *E. hermaphroditum* might be linked to growth during the growing season.

**Table 4: Growth parameters (means). Numbers in parentheses represent variability between specimens (standard deviation, +/- SE).**

| Year | 2015 | 2016 | 2017 | 2018 |
|---|---|---|---|---|
| Stem diameter variation (measured variation between start and end of the year) [µm] | 31.74 (12.73) | 63.86 (11.62) | 11.35 (20.79) | 19.91 (11.08) |
| Stem radial growth (growth-induced, irreversible stem increment compared to the previous year) [µm] | 31.81 (11.53) | 64.66 (21.82) | 14.85 (6.27) | 17.45 (9.46) |
| Stem radial shrinking (stem shrinking during the winter months, after the end of the main growing season) [µm] | 35.18 (10.35) | 28.13 (14.44) | 38.45 (12.77) | 41.61 (13.47) |
| Day peak shrinking occurs [DOY] | 362 (16.62) | 363 (4.38) | 350 (17.16) | 356 (13.63) |
| Day shrinking starts [DOY] | 311 (26.38) | 306 (14.07) | 289 (10.35) | 287 (12.81) |
| Growth initiation [DOY] | 202 (12.56) | 181 (12.12) | 198 (8.49) | 173 (13.65) |
| Growth cessation [DOY] | 267 (14.06) | 292 (14.87) | 235 (12.24) | 222 (15.47) |
| Growth duration [days] | 65 (7.93) | 111 (7.14) | 37 (10.70) | 49 (9.04) |
| Peak growth (maximum daily growth rate) [µm] | 0.48 (0.40) | 0.55 (0.85) | 0.29 (0.20) | 0.35 (0.50) |
| Day peak growth occurs [DOY] | 215 (13.68) | 204 (9.54) | 205 (8.24) | 182 (13.85) |
| Mean daily growth [µm] | 0.09 (0.03) | 0.18 (0.06) | 0.03 (0.02) | 0.05 (0.03) |

### 3.2 Micro-environmental drivers of growth patterns

We investigated how the specific on-site environmental conditions were correlated with the growth parameters described above. A set of 25 chosen environmental parameters (Table 3) was able to explain total annual growth to a very large extent,

even though not all parameters contributed significantly. Separate regression models for each environmental variable revealed an overall high explanatory power for global radiation and soil moisture on total annual growth and peak growth, compared to temperatures (Fig. 4). However, correlating temporarily aggregated data revealed that all measured environmental parameters influenced stem radial variation, but at different time scales, with radiation taking a longer time to show effects on stem increment. Comparisons of seasons also showed that on-site environmental conditions had strong explanatory power during spring, when many of the specimens started stem increment. Conversely, the explanatory power was comparatively low during autumn, when stem shrinking was detected (Fig. 5).

The correlations (Fig. 6) indicated that a multitude of environmental drivers influenced growth, yet, to a very low degree and with high inter-annual variation. Maximum soil moisture was significantly correlated with total annual growth (r = 0.47). Minimum soil moisture was positively correlated with growth cessation (r = 0.56), as were annual mean soil moisture (r = 0.40) and soil moisture sum (r = 0.39), indicating a strong overall influence of soil moisture on growth processes throughout all four seasons. Growth initiation, on the other hand, was linked to spring radiation (r = -0.37) and winter temperatures (r = 0.40). Furthermore, growth cessation was negatively correlated with maximum shoot zone temperatures (r = -0.46), mean summer temperatures (r = -0.40), and GDD5 (r = -0.40).

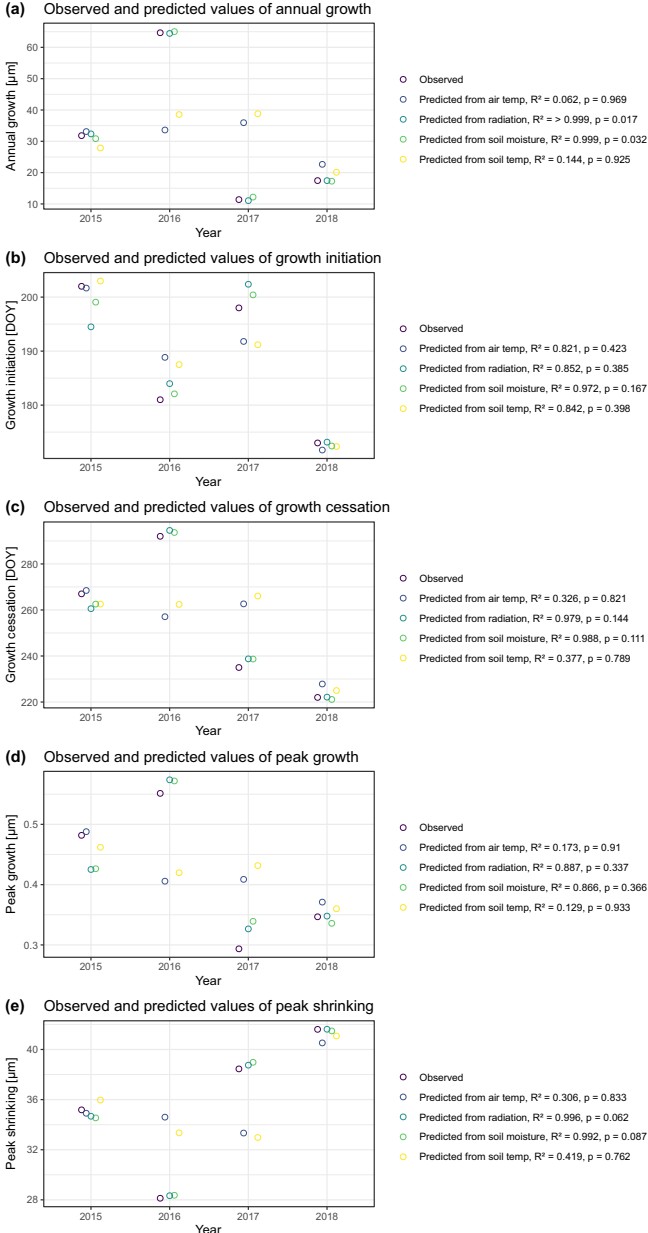

**Figure 4: Observed and predicted values of growth parameters (total annual growth, defined as growth-induced, irreversible stem increment compared to the previous year, timing of growth (growth initiation and cessation), peak growth, defined as the maximum daily growth rate, and peak shrinking, defined as the maximum stem shrinking during the winter months, after the end of the growing season). Predictions were obtained from the respective mean of four micro-environmental drivers and the Julian day at which their maximum occurred.**

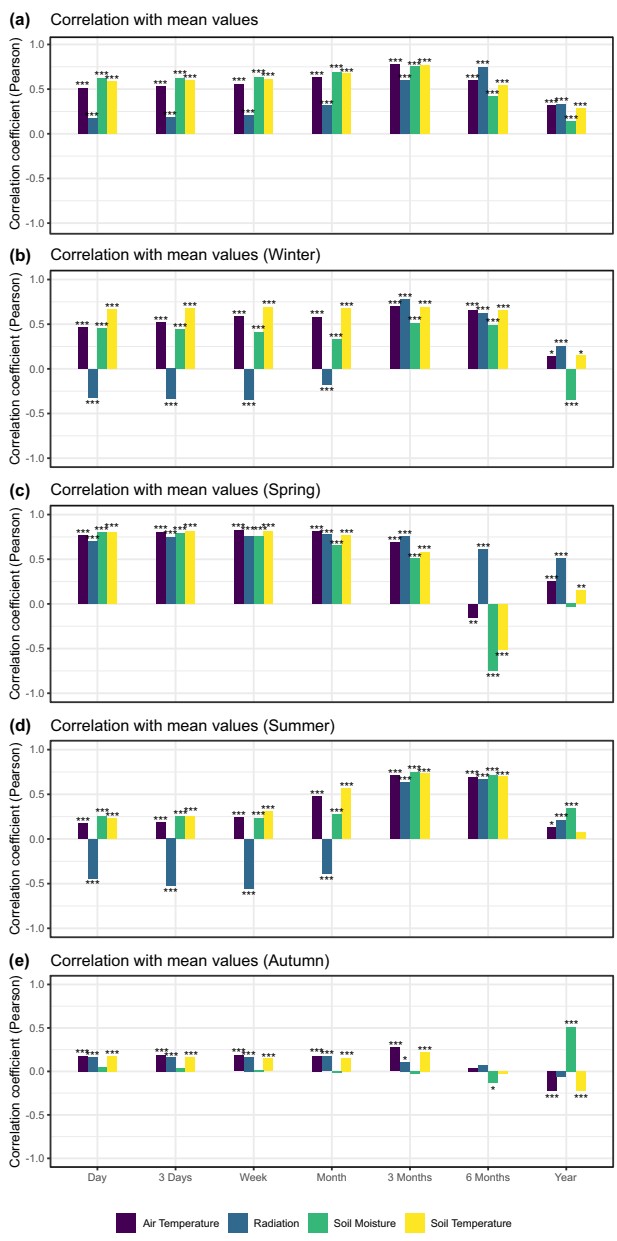

**Figure 5: Pearson correlation coefficients for radial stem diameter against aggregated environmental data (shoot and root zon e temperatures, soil moisture, and global radiation). Each daily value of stem variability was correlated with the environmental mean of the prior period of up to one year. All 12 monitored specimens entered into the correlation analysis with individual va lues for both stem diameter and environmental parameters. * indicate significance (*** = p -value < 0.001, ** = p-value < 0.01, * = p-value < 0.05).**

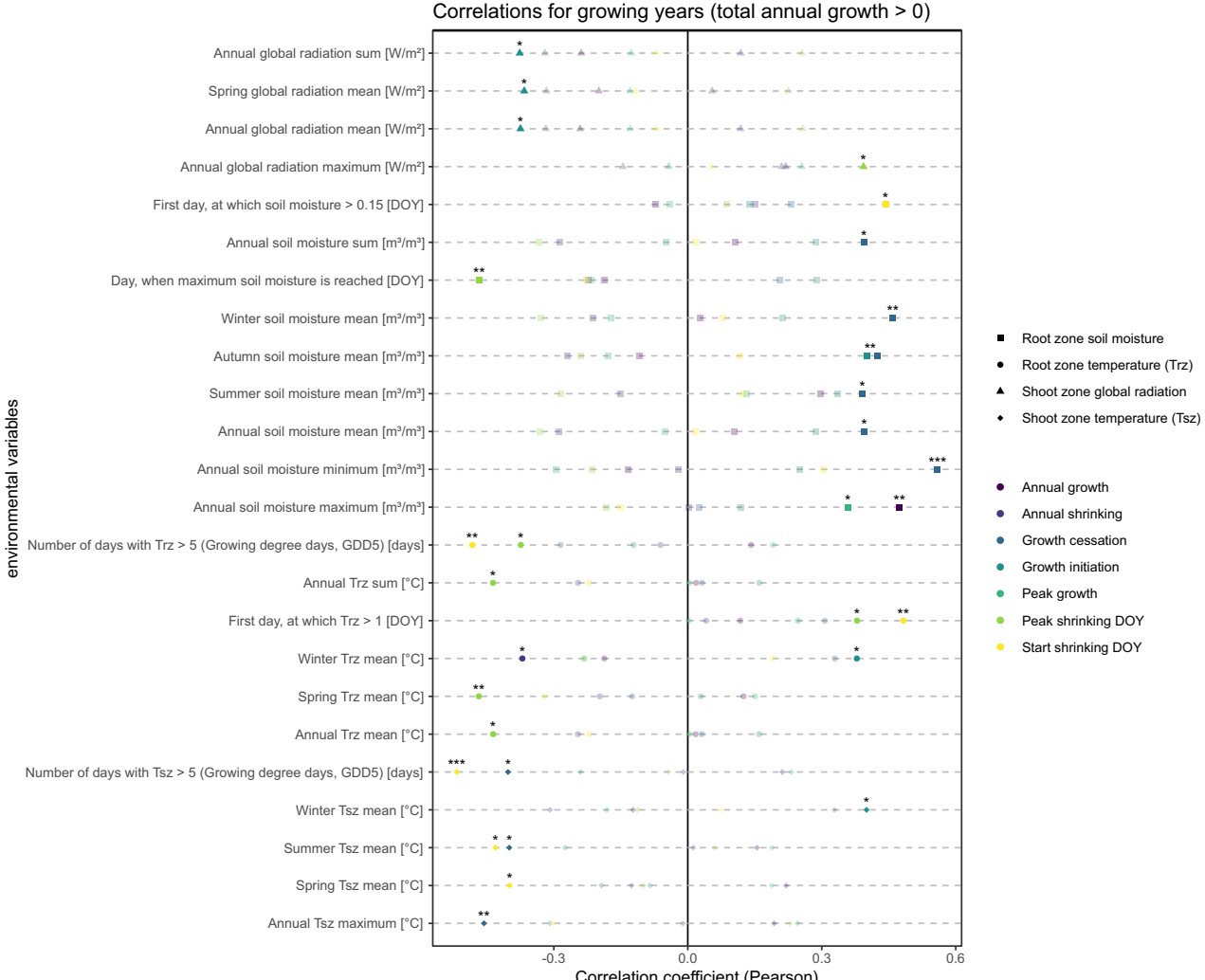

Figure 6: Pearson's correlation coefficients for growth parameters with key environmental parameters, characterizing on -site environmental conditions (for years with a total annual growth > 0). The growth parameters included: a) Total an nual growth, defined as irreversible stem increment in comparison to the previous year, b) Total annual shrinking, defined as stem shrinki ng during the winter months, after the end of the growing season, c) Timing of the main growing season (start and end), d) Peak growth, defined as the maximum daily growth rate, e) Timing of the winter shrinking phase, including the start of stem shrink ing (usually cooccurring with the maximum annual stem diameter) and peak shrinking, or the day when minimum winter stem diameter is reached. All dates were given as day of the year (DOY). Only those environmental parameters that correlated significantly with at least one growth parameter are included in the plot. Nontransparent colors indicate significance (*** = p-value < 0.001, ** = p-value < 0.01, * = p-value < 0.05).

# 4 Discussion

## 4.1 Intra-annual patterns of stem diameter change

In this study, we demonstrated that our focal species *Empetrum hermaphroditum* displayed distinct annual patterns of stem diameter change in response to near-ground environmental drivers and in close accordance with distinct conditions caused by the local topographical characteristics of the studied wind-blown ridge positions. These unique environmental conditions are a feature of the heterogeneous topography that characterizes alpine terrain (Scherrer and Körner, 2011) and are thus distinctly different from the surroundings. They include high exposure to global radiation and very little, temporary snow cover during the winter months, associated with very low temperatures (Wundram et al., 2010). Moreover, a layer of lichens limits water transport from the soil to the atmosphere, reducing evaporative loss to a minimum and, as such, reducing the danger of summer drought (Löffler, 2005). These conditions varied comparatively little between the study regions (Fig. 2 and Fig. A1), and led to very similar seasonal growth patterns and timing of growth (Fig. A5), suggesting that their influence on growth conditions was stronger than the governing regional environmental signal, which was not clearly reflected in our growth data, as suggested by Bär et al. (2008).

Thus, we confirm findings by Bienau et al. (2014), suggesting that *E. hermaphroditum* had consistent response patterns to on-site environmental drivers and our results endorse the crucial role of topography in determining growth response (Ropars et al., 2015). In contrast to the oceanic-continental gradient, our study showed high inter-plant growth variability (Fig. A1 and Fig. A4), which has been previously described in *E. hermaphroditum* (Bär et al., 2008) and could be a result of the nanoscale of internal growth variability within the multi-stemmed plant itself (Bär et al., 2007).

Our findings regarding the seasonally differentiated response to near ground environmental conditions (Fig. 5) highlight the importance of winter conditions for early growth. This indicates that for our sampled evergreen species the degree to which photosynthetic activity was effective in synthesizing carbohydrates during the winter months was especially important. Such continued photosynthetic activity was found in *E. hermaphroditum*, as well as several other evergreen shrub species before (e.g. Bienau et al., 2014; Wyka and Oleksyn, 2014; Blok et al., 2015). It is forced by the exposure of the plant to high incoming solar radiation, which typically occurs at ridge positions that lack a protective snow cover (Saccone et al., 2017). Accordingly, we identified winter and spring radiation as a strong driver, which caused the start of the growth phase (Fig. 5). This photosynthetic activity during winter causes continued water transport under extreme temperatures, increasing the risk of cavitation (Tyree and Sperry, 1989; Venn and Green, 2018), while long and severe ground frosts might limit access to soil moisture, and frost-triggered droughts might thus result in tissue damage caused by an internal water deficit (Mayr et al., 2006). However, *E. hermaphroditum* at our studied sites proved mostly frost hardy, drought tolerant and highly adapted to these conditions (Carlquist, 1989; Hacke et al., 2001), with winter stem diameter change closely linked to soil moisture

availability and singular thawing events, suggesting that the sampled specimens were able to utilize available liquid water even under extreme conditions (Fig. A7). Low temperatures during the winter months prevented cell production and differentiation, resulting in a carbon overflow (Körner, 2015; Saccone et al., 2017), which gives *E. hermaphroditum* the ability to start growth activity as soon as liquid water is available in the root zone (Starr and Oberbauer, 2003). In general,
carbon assimilation may be greater or lower than the demand for plant functions, resulting in periods of carbon surplus and deficit, respectively (e.g. Martínez-Vilalta et al., 2016). Though there is never a critical depletion of carbohydrates in alpine plants, resources at or shortly after snowmelt and, thus, at the onset of the growing season, are still diminished - due to respiratory consumption during winter (Körner 2021). Any surplus of carbohydrates, gained by photosynthetic activity in late winter/spring due to high solar irradiance is, thus, likely to be beneficial for growth throughout the growing period. The
ability to use additional photosynthetic opportunities throughout the year is similar to mechanisms found in *Juniper thurifera* (Gimeno et al., 2012) and, for high latitude vascular plants and shrubs, photosynthetic activity was previously observed under extreme thermal conditions (Semikhatova et al., 1992; Lundell et al., 2008). If confirmed, the ability to preserve resources produced during the winter months would give *E. hermaphroditum* a local advantage compared to broadleaved species at the same sites. However, it is most likely a unique feature of the snow free ridge positions and thus not present at
other micro-topographic positions, where energy budget and water balance is strongly altered (Dahl, 1956; Fritts, 1976; Löffler et al., 2006; Pape et al., 2009).

## 4.2 Total annual growth

Our results showed that total annual growth was largely determined by peak growth, indicating that the overall duration of the growth phase was less important for overall growth than the daily growth rate. Thus, total annual growth can be
interpreted as a function of daily growth. Comparing the environmental drivers, annual growth can be best predicted by soil moisture and, to some extent, by global radiation, whereas shoot and root zone temperatures have minor explanatory power. In inter-annual comparison, this was especially evident in 2018, the year experiencing the highest temperatures. Despite these conditions, the average total growth was considerably lower than in 2016 (17 μm compared to 65 μm). A possible explanation might lie within the spring conditions (March, April, and May) (Fig. 4). In 2018, temperatures rose to
comparatively high values in April and early May (Fig. 2), with soil temperatures above 0 °C and consequent thawing processes evident in our soil moisture data. This early warm phase was followed by a short cold snap and ground frost. This would explain the early growth start in 2018, and the absence of high growth rates despite favorable summer conditions, indicating that an early growth start may not be efficient in terms of total growth, if conditions in early summer prevent the survival of the formed cells. For most snow-free ridges, budburst and flowering are not influenced by snowmelt and can,
therefore, occur early on, causing high vulnerability to late frost events (Weijers et al., 2018b). This is in accordance with the findings of Choler (2018) and Weijers et al. (2018a), who suggested the strong influence of sub-zero temperatures in spring, counteracting improved conditions during summer. Here, it is worth noting that the contrasting year, 2016, experienced the

highest number of days with soil temperatures above 5 °C and temperatures rose quickly and steadily to that threshold without reaching continuously higher values during the summer (Fig. 1). This could indicate that optimum growth conditions lie within a soil temperature span of 5 to 10 °C, which is in accordance with previously reported temperature thresholds for alpine plant distribution (Körner and Paulsen, 2004; Rossi et al., 2008; Steppe et al., 2015). On the other hand, our regression analysis indicated no direct relationship between total growth and near-surface temperatures and no clear thermal growth limit, suggesting more complex connections, probably influenced by the evident temperature extremes at our chosen sites (Körner and Hiltbrunner, 2018). Thus, in our studied alpine environment, we cannot confirm high temperatures as the main general driver of shrub growth, as was assumed in several previous studies (i.e., Elmendorf et al., 2012; Hollesen et al., 2015; Ackerman et al., 2017; Weijers et al., 2018a).

Previous studies commonly used free atmospheric air temperature measured at 2 m above ground and ring width measurements (e.g. Bär et al. 2007 and 2008; Hollesen et al., 2015; Weijers et al., 2018a). A direct comparison of annual growth derived from our dendrometer measurements and such ring width measurements at the studied sites revealed high synchrony (Fig. A7). Here, the ring width data was linked to summer temperature as well, suggesting that the assumed temperature-growth relation holds partly true at our sampled sites (Fig. A8). Dendrometer data have the potential to reveal much deeper insights into complex functional aspects of growth, and in combination with on-site environmental data might help rethinking climate growth relations. Further studies are necessary here to fully explore how dendrometer measurements compare to traditional measurement methods and which additional information can be gained. Still, the comparative data presented in Fig. A8 clearly shows that both have the potential to reveal important aspects of stem variability and growth.

Overall, we found high variability in annual growth between specimens and years, with some specimens experiencing zero growth in more than one year. This occurred in 31% of the dendrometer curves, mostly in 2017 (15%). We therefore attributed these dormant years to comparatively long periods of snow cover during the previous winter, which might have prevented *E. hermaphroditum* from photosynthetic activity and resource accumulation, and thus, may have limited a crucial precondition of growth success. This assumption was supported by a highly negative correlation between stem diameter variation and the number of snow-free days (R = -0.60, p = 0.024, Fig. A9) during these years. The effects of winter snow cover on shrub growth are a critical topic in arctic and alpine ecology, with findings ranging from positive (Blok et al., 2015; Addis and Bret-Harte, 2019) to negative (Schmidt et al., 2010) growth responses, depending on snow depth and vegetation type. In accordance with Buchwal et al. (2013), we assumed that during years of no apparent radial growth, dwarf shrubs might prioritize growth in the more protected and long-living belowground segments, instead of investing in the more vulnerable shoots. This ability to reduce cambial activity to a minimum and cease above-ground wood formation is a trait common among woody plants (Wilmking et al., 2012). However, this implies that our studied specimens were locally

adapted to snow-free conditions at their exposed positions, and consequently, might not be able to cope with such unexpected growth conditions.

## 4.3 Peak growth

We found peak growth, the maximum daily growth rate, closely linked to soil moisture and usually occurring in connection with the soil moisture maximum, highlighting the overall importance of the root zone soil moisture as the key driver of growth in *E. hermaphroditum* (Fig. 6). This was evident in the strong contraction and expansion patterns of the stems, most likely controlled by active or passive water level variability within the plant, linked to the extreme thawing and freezing processes prominent at our study sites. The predominant role of peak growth in controlling total growth suggested that the shrubs were usually able to invest in new cells following the cell water level rise caused by thawing conditions, most likely affected by prior carbohydrate storage (see above). This highlights a key role of the extreme winter temperatures, causing winter desiccation and consequent water stress even if soil water contents are usually sufficiently high during summer (Tranqulini, 1982; Mayr et al., 2006). In trees, the timing of maximum growth in cold environments has been linked to day length (Rossi et al., 2006; Duchesne et al., 2012). This cannot be confirmed for our monitored shrubs because of the high variability between specimens, showing a far broader range in the Julian day at which peak growth occurred than observed in trees. Therefore, peak growth is most likely controlled by other factors in shrubs than those assumed for trees, where available soil moisture is not limiting during the photosynthetically active period in spring and early summer. Instead, our results suggest that soil moisture availability played a key role.

## 4.4 Growth initiation and cessation

We found the overall link of growth initiation and cessation to the micro-environment comparably low, indicating that growth duration was most likely influenced by a multitude of environmental variables with large differences between years and sites (Fig. 4 and Fig. 6). From the positive correlation of growth initiation with global radiation during spring, we concluded that growth initiation might be driven by the constantly increasing radiation with the astronomic rise of the angle of the sun. At this time of the year, energy transfer from global radiation into thermal heat was low, but radiation was high enough for photosynthetic activity, which might explain the decoupling of thermal and radiation drivers of growth initiation. Under snow free conditions, it is likely that, in our study design, global radiation is directly linked to soil thawing close to the soil surface, which was previously linked to growth onset (Descals et al., 2020). Furthermore, growth initiation in *E. hermaphroditum* was positively linked to winter temperatures in our study (Fig. 6), indicating that low winter temperatures were correlated with an early start of the growing season, in contrast to common assumptions relating high temperatures during late winter to an early growth start (Dolezal et al., 2020). This highlights the influence of high radiation on energy storage during periods of an absent snow cover, which is usually accompanied by low temperatures, whereas mild winters are often associated with cloudy, humid weather, and snow cover on the ridges.

For growth cessation, decreasing day length was determined an unlikely trigger (Heide, 1985), because of high inter-stem variability. Instead, shoot zone temperatures played a role in determining when xylogenesis ceased, but a critical temperature threshold, as present in many trees (Rossi et al., 2007; Rossi et al., 2008; Deslauriers et al., 2008) and found for xylem growth of alpine rhododendron shrubs (Li et al., 2016), could not be determined for *E. hermaphroditum*. Growth cessation was linked to summer temperatures, suggesting that high peak temperatures during the summer might delay growth cessation by the thermal promotion of carbohydrate storage, enabling growth continuity even under unfavorable thermal constraints during autumn (Fig. 6). Additionally, we identified soil moisture as the main driver for the end of the growing season. Yet, we could not determine a minimum soil moisture threshold, which would lead to growth cessation. A contrasting pattern was shown in 2016, with a prolonged growing phase after dry conditions during early summer. Thus, we concluded that *E. hermaphroditum* growth was strongly dependent on the availability of water, which is why dry conditions during spring and early summer were less crucial and did not lead to immediate growth cessation, because evaporation was not fully active at these times of the year leaving enough exploitable moisture available. This accounts for the species dependency on a damp climate and high rainfall (Bell and Tallis, 1973). In a close relationship with the end of growth, the stem diameter started to shrink, marking the beginning of the winter shrinking period. Overall, our results regarding growth timing demonstrated the complexity of these processes and we cannot confirm a clear relation of growing season length and overall growth (Rammig et al., 2010; Blok et al., 2011; Prislan et al., 2019).

### 4.5 Stem shrinking

A phase of winter stem shrinking has been described in trees (Winget and Kozlowski, 1964; Zweifel and Häsler, 2000). The distinct and strongly pronounced phase we found in our sampled specimens, however, might be described as a unique feature of shrub growth, which we documented for the first time in this study. The reason why this phase has not been described earlier might be attributable to the methods for shrub growth measurements used in the past, which were insufficient to document intra-annual variability at the appropriate time scale. High-precision dendrometers can reveal these patterns in growth variability, demonstrating the large amount of additional information gained from this method compared to traditional measuring methods.

In trees, radial stem shrinkage has been related to sap flow and tree water content (Winget and Kozlowski, 1964; Zweifel and Häsler, 2000; Zweifel et al., 2006; Tian et al., 2019). When temperatures sink below approximately −5 ∘C, extra-cellular water begins to freeze, inducing the osmotic withdrawal of intra-cellular water and, thus, cell and ultimately stem shrinkage (Zweifel et al., 2000; King et al., 2013). As our observed shrinking phase occurred during such periods of extremely low temperatures during winter and was negatively correlated with root zone temperatures during these months, we assumed a similar relationship with the extreme subzero temperatures and consequent freezing conditions present at our study sites. As

such, stem shrinkage could be interpreted as a result of freezing processes causing living cell shrinkage because of water losses, as commonly observed in trees and other woody plants (Neuner, 2014; Charra-Vaskou et al., 2016). As subzero temperatures are a major environmental stress factor, alpine shrubs have developed a strategy to avoid frost damage, especially where the protective effects of snow cover are missing (Kuprian et al., 2014; Neuner, 2014). We assume that *E. hermaphroditum* similarly uses cell dehydration to actively protect living cells from the consequences of freezing (e.g., ice nucleation), causing the radial stem contraction evident in our data. The start of the shrinking period was linked to the day when peak growth occurred, but finding a singular event causing the shrinking process to start proved difficult without significant connections with the first frost events or autumnal soil moisture declines. We concluded that surviving during extreme winters was the main principle governing *E. hermaphroditum* growth when existing at alpine ridge positions, causing unique adaptations to local micro-site conditions.

## 5 Conclusions

Here, we show that high-precision dendrometers are suitable measuring instruments for identifying growth patterns in dwarf shrubs. For our focal species *Empetrum hermaphroditum,* the method yielded several novel insights into phenology and growth physiology. As an evergreen shrub at exposed and therefore mostly snow-free positions, *E. hermaphroditum* appeared capable of continuing photosynthetic activity throughout the year and to thus aggregate resources for use in early cell formation, as observed in other evergreen plants (Wyka and Oleksyn, 2014). This provides a competitive advantage over deciduous species in the same habitat, limiting the risk of losing resources through competition. To sustain the continued metabolism throughout the year, we found that *E. hermaphroditum* was highly dependent on available moisture, and thus, highly adapted to local microsite conditions. Furthermore, our findings confirmed the positive effects of temperatures on shoot growth in *E. hermaphroditum* to some extent (Chapin and Shaver, 1985; Shevtsova et al., 1997; Bråthen et al., 2018), yet, a clear link between near-surface thermal conditions and growth was lacking, with the overall growth mechanism defined by moisture availability, as well as solar radiation. Hence, temperatures mainly play a role in freezing and thawing processes, on an intra-annual scale. We can thus confirm that while there is a link between shrub growth and warming conditions, it is most likely not uniform and highly variable over spatial and temporal scales (Elemendorf et al., 2012).

Overall, the fine-scale data provided by dendrometer measurements proved highly important, since they allowed for a detailed growth analysis, revealing a growth mechanism that is highly adapted to the local micro-environmental conditions at our studied exposed ridge positions. This mechanism explains the wide distribution and competitive ability of the species at these sites (Bienau et al., 2014; Bienau et al., 2016; Löffler and Pape, 2020) and highlights the prominent role of on-site near-ground environmental conditions in controlling growth processes (Zellweger et al., 2020). In a changing climatic regime, this might become a disadvantage, complicating the adaptation to warming winters and longer snow-covered periods, coupled with prolonged dry periods during summer (Hollesen et al., 2015; Weijers, Pape et al., 2018), which might

cause early growth cessation. In accordance with previous studies (Milner et al., 2016; Virtanen et al., 2016; Wheeler et al., 2016; Saccone et al., 2017), our results suggested that winter conditions and altered snow regimes represented one of the most serious threats to evergreen shrub growth in tundra ecosystems. We conclude that because of the high local adaptation and dependency on specific winter radiation conditions and soil moisture availability of this species, *E. hermaphroditum* will not be able to persist at exposed positions in a changing climate or respond with longer periods of dormancy to warming

conditions. This will potentially promote the spread of competing deciduous species and thus contribute to the arctic-alpine greening trend.

## Data availability

All underlying data pertinent to the results presented in this publication are publicly available in a data publication in "ERDKUNDE---Archive for Scientific Geography" (https://www.erdkunde.uni-bonn.de). DOI:

10.3112/erdkunde.2021.dp.01.

## Author contribution

JL had the idea, designed the research platform, conducted the field work, and together with RP ran the long-term project. SD analyzed the data, lead the writing of the manuscript and arranged the figures, with contributions from RP and JL.

## Competing interests

The authors declare that they have no conflict of interest.

## Acknowledgements

The authors thank Eike Albrecht, Niklas Beckers, Elise Dierking, Nils Hein, Stef Weijers, and Dirk Wundram for collaboration within our LTAER project, Ole Øvsteng and Anders Svare for hospitality, and both the landowners and Norwegian authorities (Vågå and Stranda municipalities) for overall support. Parts of this study were supported by the

570 Deutsche Forschungsgemeinschaft (DFG) (grants LO 830/16-1, LO 830/32-1).

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

**Appendix A**

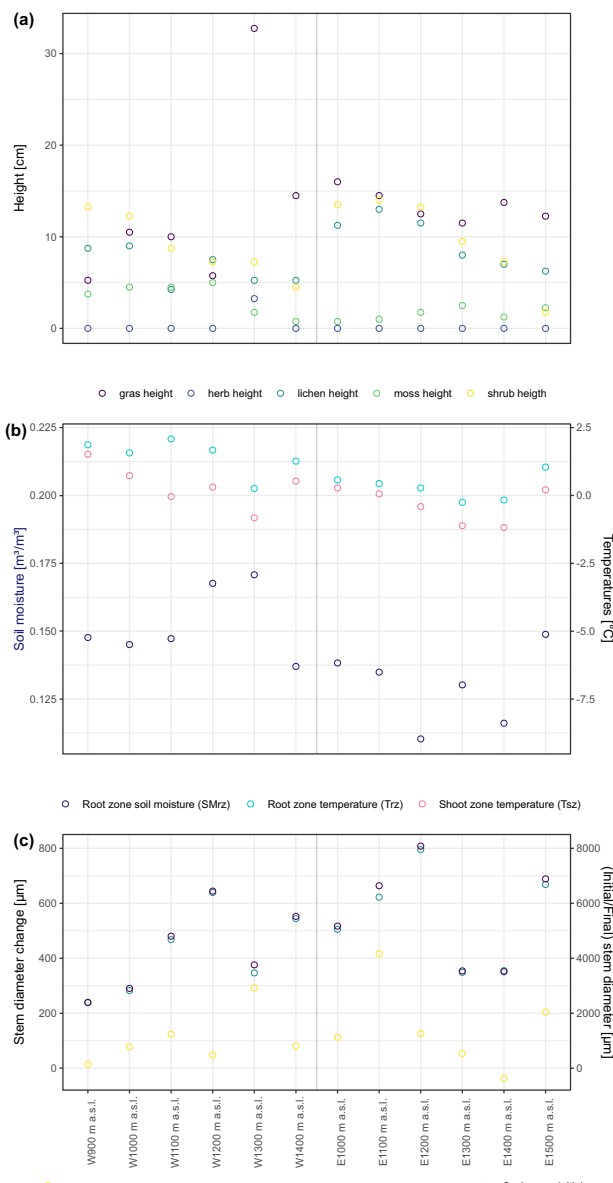

**Figure A1: Summary of canopy height at the 12 studied sites by major vegetation type (a). Trees are not included, because there were to trees present. (b) shows micro-environmental conditions averraged over the studied period, as well as minima and maxima for temperatures measured within the shoot- and root zone and for soil moisture in the root zone. (c) summarizes total stem diameter change as well as stem diameter at the start and end of the studied period 2015-2019).**

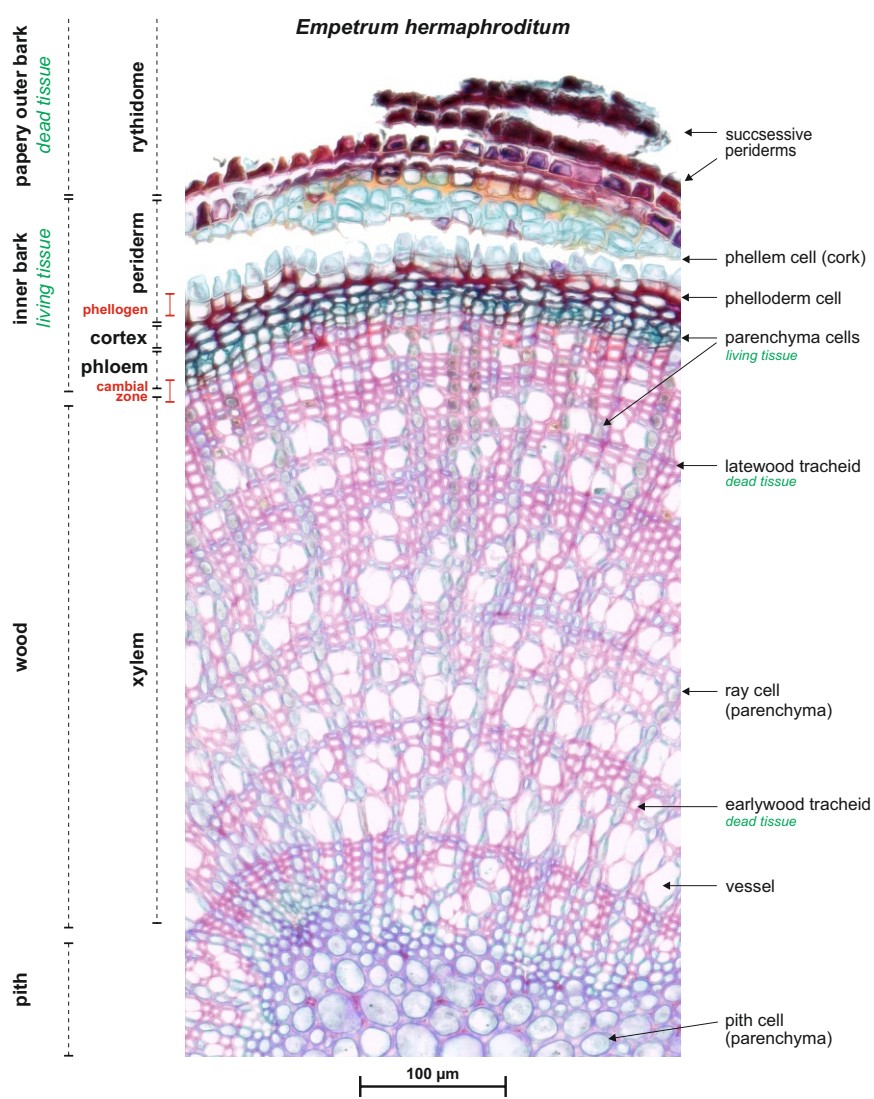

**Figure A2: Radial micro-slide of a stem from Empertum nigrum ssp. hermaphroditum (1:1000 magnification). The outermost layers of the papery outer bark, which repeat the successive pattern of the shown periderm, got lost while cutting. In our dendrometer approach, we removed the outer layers of the bark, most likely down to the phellogen. As shown here, the loose bark structure allows removal without severe damage of the inner tissue. We aimed at mounting our dendrometer sensor as close to the still protected cambial zone, to achieve data on  hysiologically active stem diameter variability such as growth, excluding swelling and shrinking of the passive outer bark tissue.**

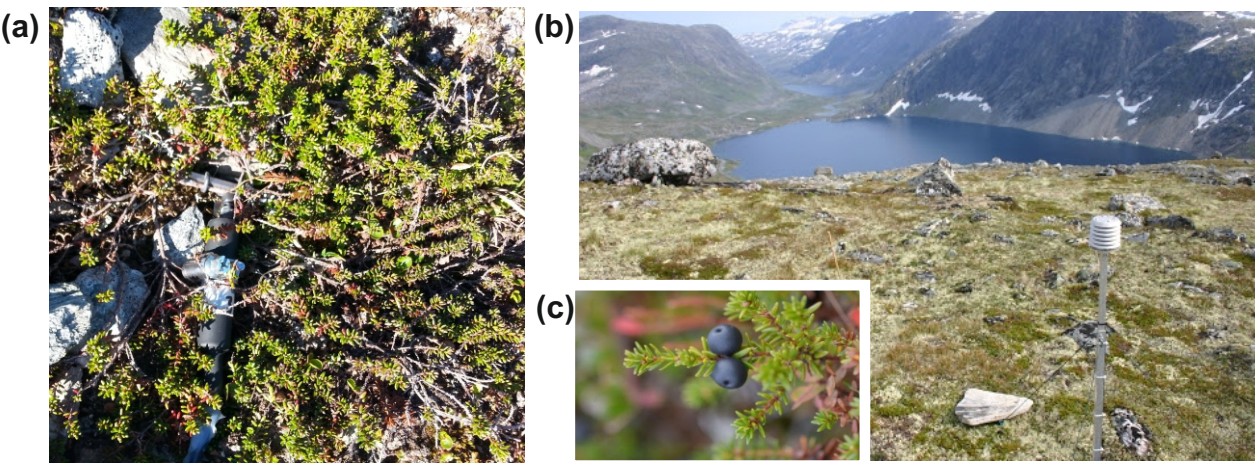

**Figure A3: Dendrometer set up (a), the studied species E. hermaphroditum in the studied region in Central Norway (b), and the species fruits and leafs (c).**

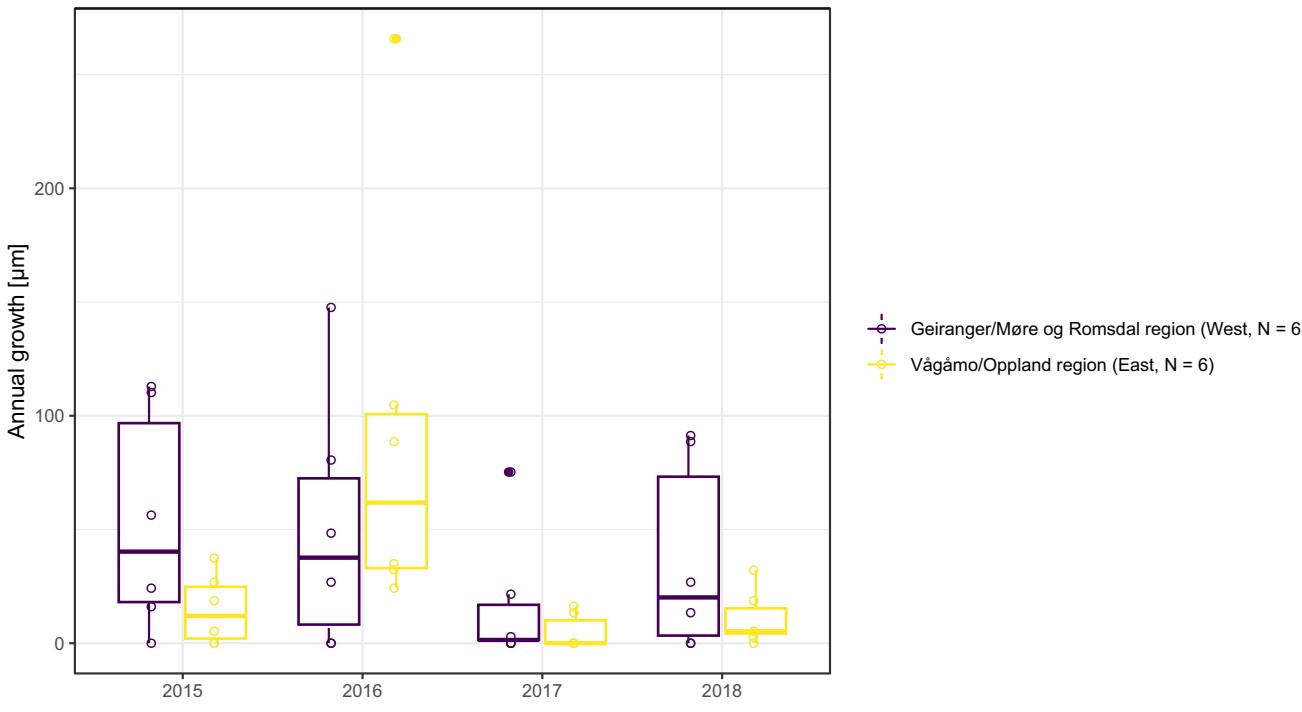

**Figure A4: Boxplots of observed annual growth within each of the two study regions (Vågåmo/Oppland region (East, N = 6) and Geiranger/Møre og Romsdal region (West, N = 6)).**

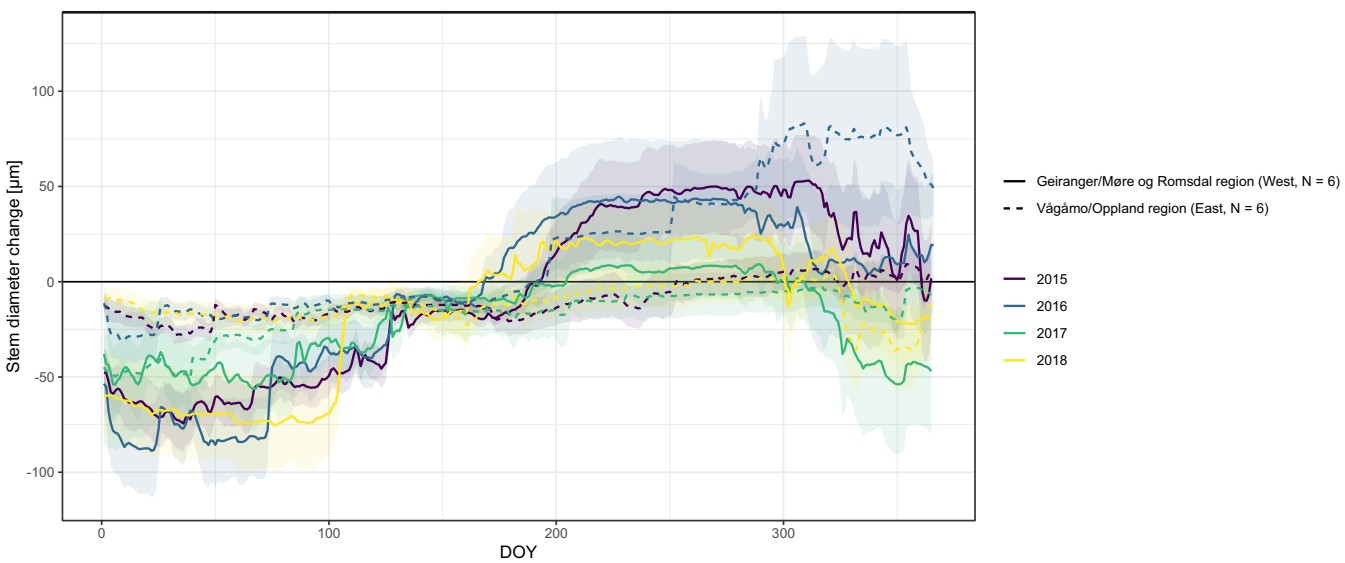

**Figure A5: Observed stem diameter change averaged over the monitored specimens within each of the two study regions (Vågåmo/Oppland region (East, N = 6) and Geiranger/Møre og Romsdal region (West, N = 6)). Transparency indicates standard deviation.**

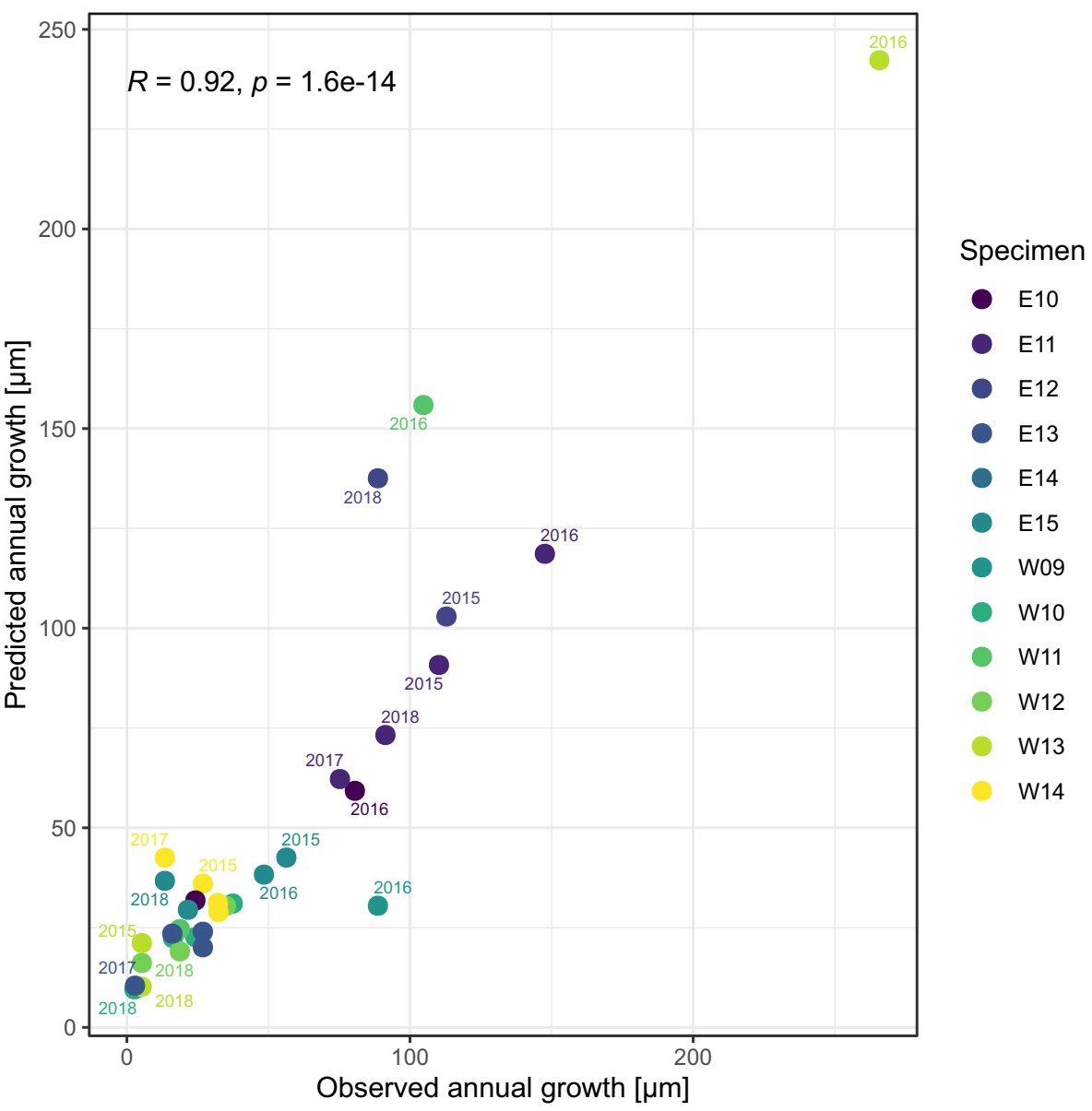

**Figure A6: Observed annual growth and annual growth predicted by a linear regression model including timing (growth initiation and growth cessation), as well as peak growth as independant variables. Colours indicate the monitored specimens at the individual sites (E = East, W = West, numbers indicate elevation (100 m a. s. l.).**

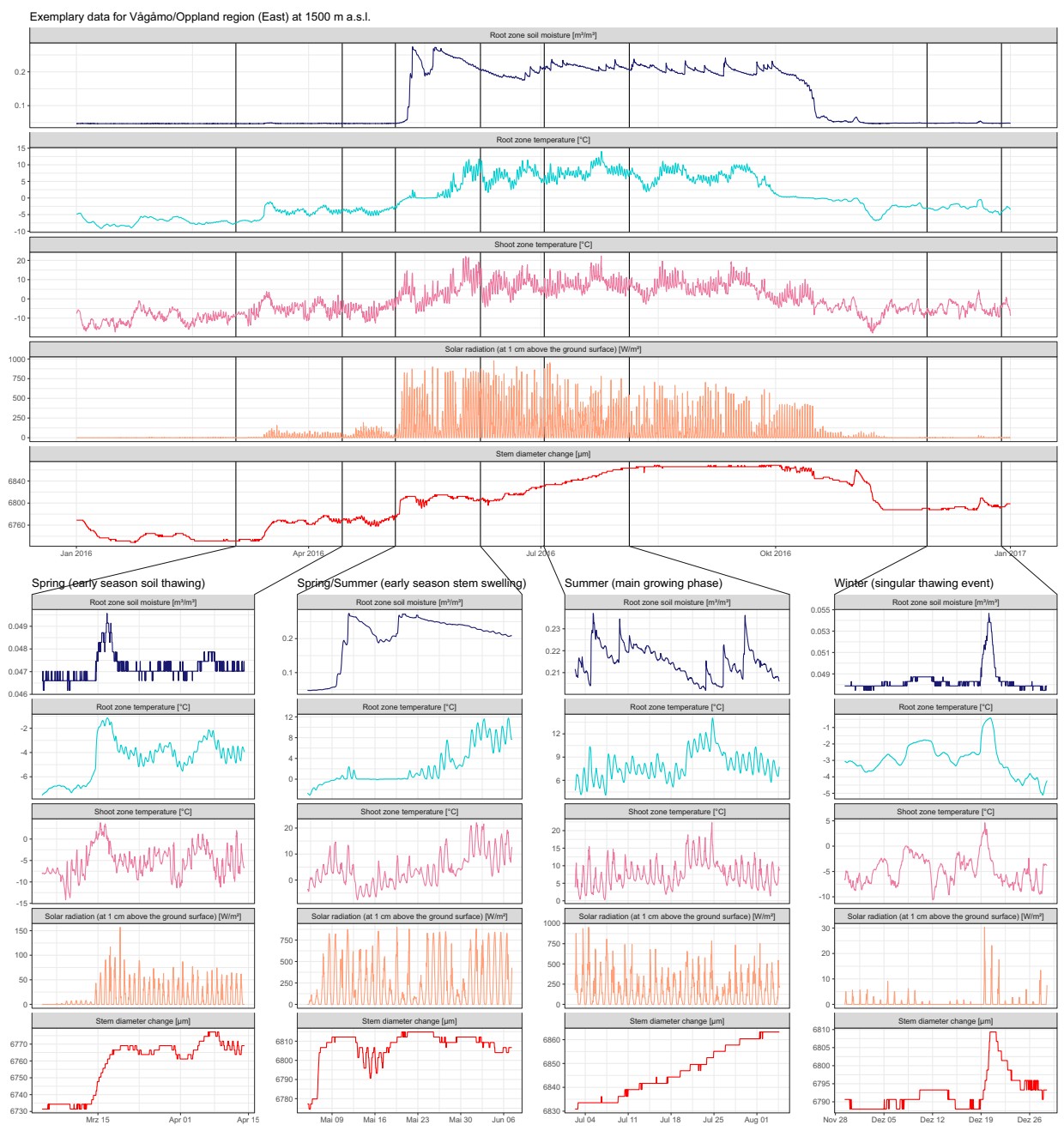

**Figure A7: Raw data for one exemplary specimen from the Vågå/Innlandet region at 1500 m a.s.l. Here, we present hourly data of stem diameter changes and the respective micro-environmental conditions. The four sections show important phases of the annual stem diameter variability and their relation to the micro-environment in detail. Coupling of soil moisture and stem diameter during the winter and spring months, when water induced stem swelling and shrinking occurs, and decoupling during the main growing phase is clearly evident. Additionally, the direct response of stem diameter to singular soil thawing events in winter is clearly visible in the curves.**

**(a)**   Ring width index and dendrometer data

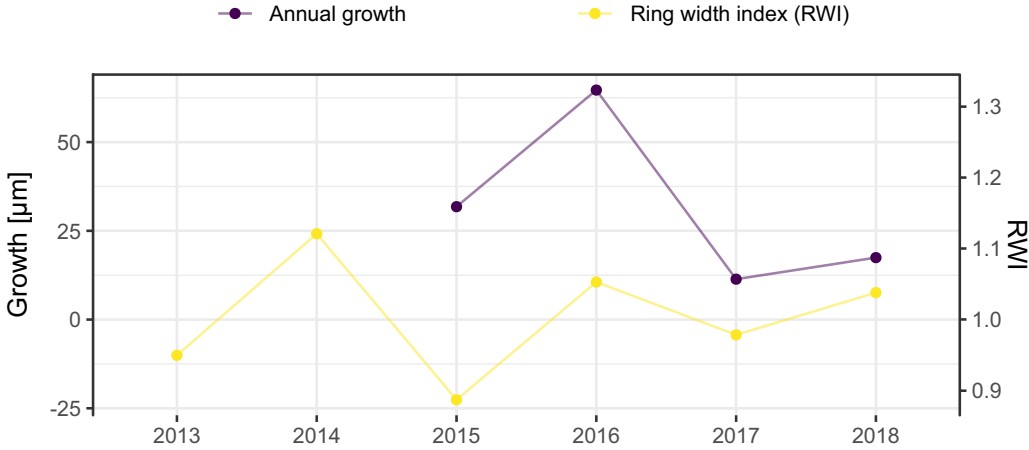

**(b)**   Mean summer temperatures (June, July, August)

**Figure A8: Comparison of annual growth measured using dendrometers (see Material and Methods), and ring width derived by measuring annual growth rings of 12 specimens from similar positions on exposed ridges, presented here as ring width index (RWI). Ring width was measured from multiple micro-slices per specimen (following Bär et al., 2006) (a), (b) shows free atmospheric air temperatures measured at 2 m above ground in both study regions. Such temperature data is commonly used for comparison of climate-growth relationships in dendroecological studies. With this figure we aimed to reproduce previous studies (Bär et al., 2006 and 2007) for comparison with our dendrometer measurements.**

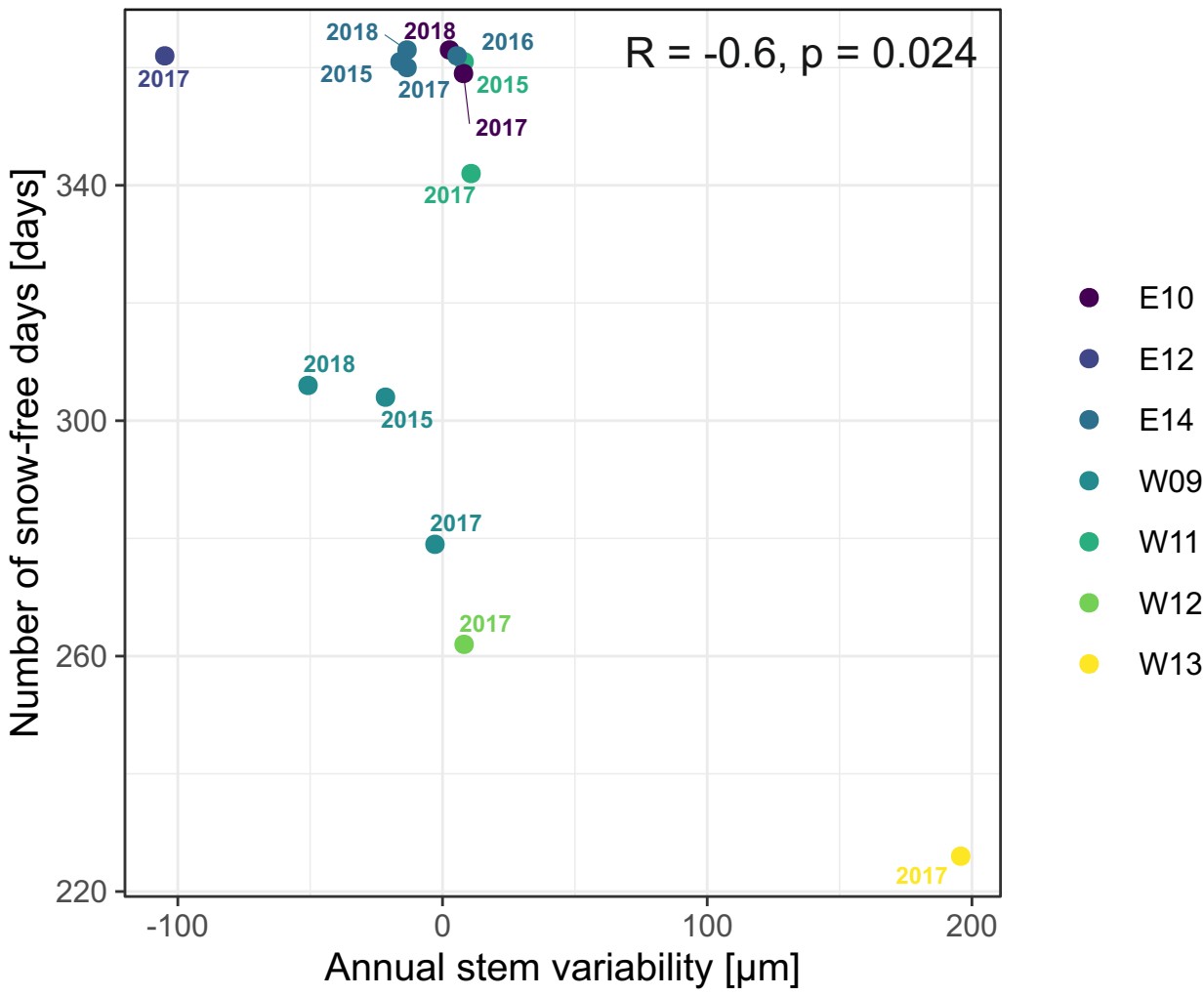

**Figure A9: Number of snow-free days and annual stem diameter variability in years in which no irreversible growth occurred (total annual growth = 0). Colours indicate the monitored specimens at the individual sites (E = East, W = West, numbers indicate elevation (100 m a. s. l.).**