# Peer review of "The application of dendrometers to alpine dwarf shrubs – a case study to investigate stem growth responses to environmental conditions"

_Biogeosciences, 2021_

## Author Response (AR1)

We here present a summary of all changes made in the revised manuscript, together with a summary of our point-by-point replies to the reviewer's as already posted in the interactive discussion alongside our original manuscript.

**Changes made (summary):**

Clarification of the study design:
- We included details on the monitoring setup and technical implementations of the study design, and gave additional information on the placement of the dendrometers and handling of outer bark tissue.
- We also gave some additional background regarding the study design and trial periods before the start of the study period.
- We added some additional information on the position of the tree line in the studied regions and the elevational levels of our studied sites.
- To further improve clarity for the reader we included an additional Figure showing a radial micro-slide of an *E. hermaphroditum* stem with details on stem anatomy and bark structure.

Regarding methodical ambiguities:
- One of the main challenges proposed by our study design was the separation of water-related stem shrinking and swelling and growth-induced stem increment. We gave additional details on how we approached this challenge. Since our chosen approach was chosen based on visual interpretation of our data we included an additional Figure presenting exemplary raw data to the reader.
- We also addressed major concerns of one of the reviewers regarding sample size. - We validated our chosen dendrometer approach by direct, exemplary comparison with traditional ring width data from our studied sites, presented in an additional Figure.
- We expanded on the text in the Discussion section to further explain and strengthen our interpretation of the observed results during the winter months, giving additional references, supporting this interpretation.
- We revised the Material and Methods section with special emphasis on clarity and wording.

Writing style and wording:
- We rewrote passages with imprecise wording.
- We revised the Abstract.
- We rethought our use of some of the terms used and replaced or removed them as suggested by the reviewers (e.g. point dendrometers) - We made sure that the calculated parameters are clearly defined and consistently named throughout the text. - We rethought the placement of some passages in the text and moved them to more appropriate chapters accordingly.

Revision of the Figures:
- We made some major and minor adjustments to the Figures and Figure legends, improving clarity.
- We added three new and relevant Figures to the supplementary material (see above).

- We added new information to Figure A1, to highlight our interpretations regarding the elevational gradient

**Point-by-Point responses to the reviewers:**

Reviewer 1 (Comment 1)

*Due to considerable methodological deficiencies and a lack of physiological understanding of plant growth limitation in the alpine zone, I cannot recommend publication of this manuscript.*

The authors would like to thank Reviewer 1 for detailed and constructive input. There are two main points of concerns, a) bark, which was of course removed before measurement, and b) averaging procedure, which would have undermined the strength of our unique dataset. Both issues can be ruled out, as they were obvious misunderstandings. Moreover, we reworked the entire manuscript according to very helpful comments and further detailed suggestions (see below). As such, we addressed shortcomings in articulating the approach of our study, rewrote passages with imprecise wording, and added important arguments, references and explanations. The authors are convinced that their extensive changes now justify publication in Biogeosciences, and by its a novel approach, strong methodology, unique dataset and unexpected results the paper will stimulate the discussion on our physiological understanding of arctic-alpine shrubs growth. As to obvious discordance with existing literature, we clearer indicated the speculative nature of some interpretations and argue that our findings give reason to rethink certain aspects of current knowledge.

*General comments:*
*In this paper authors analysed the influence of environmental factors on radial growth and intraannual growth patterns of the prostrate dwarf shrub Empetrum nigrum ssp. hermaphroditum, which is widespread in arctic and alpine ecosystems. Elevational gradients spanning 500 m were selected above treeline in a humid and continental climate region in central Norway. To determine key dates of intra-annual growth dendrometers were mounted on one major stem per plant (two regions x six*
*samples x two region, i.e., one sample per elevation result in a total of 12 dendrometer records during four successive years). Regression and correlation analyses were applied to calculate the influence of microclimate on growth parameters. Authors found that soil moisture, solar radiation and winter conditions are the main drivers of radial stem growth of Empetrum nigrum ssp. hermaphroditum.Results of this study are unexpected, as it is generally assumed that in the temperate climatic zone growth processes at and above treeline are primarily limited by temperature during the growing season. Although authors present a unique data set of dendrometer*
*records of an alpine shrub species together with site specific environmental data during four years, I have several major concerns regarding study design and analysis:*

*(i) I suppose that due to selection of only one individuum per elevation, dendrometer recordsgathered along the elevational gradient were averaged in both regions, i.e., dendrometer data sampled along a temperature gradient of 3°C (500 m elevational gradient x 0.6 °C temperature lapse rate per 100 m elevation) were averaged to obtain a mean*

*dendrometer series per region (n=6 per region), which were correlated with mean values of environmental variables. This averaging procedure, however, precludes a detailed and meaningful analysis of the influence of temperature on shrub growth at and above treeline (cf. comments to p. 9, line 205 and p. 10, Fig. 2).*

The reviewer is wrong assuming that the individual dendrometer curves (and associated environmental measurements) have been averaged prior to further analyses. While designing the study, we were well aware of common misconceptions regarding the utilization of, e.g., temperature data (cf. Körner & Hiltbrunner 2018) and therefor carefully chose physiologically meaningful variables without averaging, specifically to ENABLE (and not to preclude) a meaningful analysis. As focusing solely on analyses that also make sense ecologically seems so obvious and natural, we admittedly failed to state and clearly describe such our approach more explicitly. Based on the reviewer´s critique, we expanded on our previous statement in the Material and Methods section (lines 174/175 of the revised version):
"We first calculated these growth-defining parameters for each individual curve to assess the variability between sampled specimens. These values entered in the following statistical analyses individually."
Moreover, we included an additional statement in lines 249-252 of the revised version:
"For these, as well as the following statistical analysis, we used site-specific values for each sampled specimen to overcome common misconceptions regarding the utilization of, e.g., temperature data (cf. Körner & Hiltbrunner 2018), We carefully chose physiologically meaningful variables without averaging, specifically to enable a meaningful analysis (see also Löffler and Pape 2020). "Throughout the manuscript, whenever data were averaged, this was done after analyses only and primarily for visual purposes, to increase readability and interpretability of the graphics and reduce the size of the final tables. We revised Figure A1 to show variability between our studied sites more clearly and conveyed a better understanding of the patterns along the elevational gradient to the reader.

*(ii) Diameter dendrometers were mounted over dead outer bark, which is a highly hygroscopic tissue. The possible influence of daily/seasonal changes in air humidity on stem diameter variations were not taken into consideration (cf. comments to p. 5, line 135).*

This again revealed to be a misunderstanding: We are very aware of the effects the highly hygroscopic bark has on diameter measurements. Consequently, our dendrometers were NOT mounted over dead outer bark. We removed the outer bark before mounting the dendrometers. As we assumed this to be common knowledge and practice, we unfortunately missed to state this explicitly. To highlight this step, we included a short paragraph in the Material and Methods section (lines 134-136 in the revised version):
"During this process, we removed the outer bark to place the sensor directly on the cambium. This ensures that hygroscopic shrinkage and swelling of dead tissues from the bark do not influence the diameter measurements. Such processes have been previously addressed in trees (Zweifel and Häsler, 2000; Gall et al., 2002; Ilek et al., 2016)."
In general, Empetrum hermaphroditum has a very thin bark, which is easily removed without the danger of damaging the cambium.

*(iii) There are numerous inconsistencies in data interpretation which need to be clarified to justify publication.*

We clarified inconsistencies in accordance with the following comments and suggestions.

*Specific comments and suggestions:*
*p. 1, line 15: Elevational effects on stem growth were not analysed in this study, because dendrometer records were averaged. This procedure precludes the analysis of effects of decreasing temperature with elevation on growth.*

This is not true. Actually, we expected to find growth response with elevation, hypothesized that these would be due to thermal constraints, and were surprised not to see such a coupling in the data. All our data were measured along the elevational gradient, and based on our temperature measurements at each site we see the decreasing temperature with elevation. We revised Figure A1 accordingly to show this to the reader. However, the elevational trend found in the temperature data is neither reflected in our stem diameter data, nor in the soil moisture data (see Figure A1). The
decoupling of stem diameter variation from the temperature trend along our elevational gradient, together with the decoupling of soil moisture from the elevational gradient, suggests that alpine temperatures are not limiting growth in our shrub species at the studied sites. Alpine ridge sites are often believed to be dry as to the wind-exposed nature. As has been shown in previous studies (e.g. Löffler 2005),  soil moisture is not a limiting factor during the summer period, when plants are
exposed to atmospheric summer drought. This decoupling of the soil from the atmosphere is due to a dense lichen carpet, which dries out, has no root coupling to the ground like other plants have, effectively limits descending water transport from the soil to the atmosphere, and as such reduces evaporative loss to a minimum. Along with our general novel findings on soil moisture as a strong driver of growth in *E. hermaphroditum*, we here show that frost droughts during winter are
obviously restricting our shrub species, and that this phenomenon is similarly represented along all alpine ridges despite elevational and regional climate differences. We added this in chapter 4.1 of the discussion section:
"This decoupling of stem diameter variation from the temperature trend along our elevational gradient (Fig. A1) suggests that alpine temperatures are not limiting growth in our shrub species at the studied sites. Furthermore, we found soil moisture similarly decoupled from the elevational gradient, suggesting a reduced significance of this gradient at the micro-scale, at which our near-ground environmental parameters were measured."
Additionally, as mentioned previously, dendrometer records were not averaged before entering into the statistical analysis (see above).

*p. 1, line 18: There are inconsistencies here: According to Table 3, onset of growth did not start before DOY 173. Furthermore, mean shoot and root temperature were below 0°C in March-April 2015-2017 and < 5°C in May 2018. How can it be possible that these correlations have come about? Please clarify.*

The dendrometer curves show stem increment in spring, before the start of the growing phase (growth initiation) (see Fig. 3). This is not defined as part of the growing phase, because it does not exceed the previous year's stem diameter and therefore cannot be seen as growth, but rather as stem recovery from the autumn/winter shrinking period during the previous year. Still, this recovery phase influences total annual growth and is possibly related

to thermal conditions during winter and spring, thus reflected as such in the correlation results. Additionally, in Table 3 and Fig. 1 and 2, mean values were presented, while individual-site values entered into the analysis. We expressed this more clearly in line 251: "For these, as well as the following statistical analysis, we used site-specific values for each sampled specimen..."
Overall, growth initiation showed a positive correlation with winter temperatures (see Fig. 6 and discussion).

*p. 1, line 19: Photosynthetic activity during winter at daily mean temperatures < -10 °C is highly speculative and rather implausible.*

Even though mean temperatures are temporarily below -10°C, higher temperatures are reached during the midday and there are short, warmer periods which experience soil thawing (see Figure 1, which we revised to include daily maximum and minimum temperatures). This is because of pronounced exposure and missing snow cover at all our studied ridge positions. We used nearground air temperature as potential driver of plant growth, which does not directly reflect the temperature in the plant. Especially during periods of clear sky and full global radiation, canopy temperature is known to be much higher than our temperatures measured at 15 cm above ground.
We do not suggest that active cell growth occurs during these conditions. However, photosynthetic activity of vascular plants in late winter and under extreme conditions was reported before, and our suggestion is thus backed by the literature (e.g. Semikhatova et al., 1992; Starr and Oberbauer, 2003; Lundell et al., 2008), which we also referred to in the Discussion section. Please see also below (comment regarding p. 24, line 459).

*p. 3, line 75: "caused by induced water" – unclear meaning, please reword.*

This addition was removed here to improve clarity.

*p. 3, line 78: "inter-annual" or "intra-annual" – please check.*

We checked this and the correct term here is intra-annual.

*p. 3, line 84: Dendrometers used in this study record stem diameter variations, whereas point dendrometers record radial stem variations. Please correct.*

The term point dendrometer is not uniformly defined in literature and by manufacturers. Usually, it refers to dendrometers which measure stem variations at one (radius) or two (diameter) points, to distinguish them from band dendrometers, which measure stem circumference. We measured stem diameter, and our dendrometers are called point dendrometers by the manufacturer (probably
because of their technical principle, which allows mounting them to measure radial variations). We fully agree that the term point dendrometer might be confusing here and therefore removed it throughout the text.

*p. 3, line 88: Please explain meaning of "mechanistic adaptation".*

We here used the term "mechanistic adaptation" to distinguish our measuring (physiological) approach from the methods commonly used in dendroanatomy, which focuses on structures and anatomical signatures. However, since this term is probably not clear to all readers, we here changed it to "growth patterns", which reflects the main objective of our work more accurately.

*p. 4, line108: Please indicate that sites at 900 and 1000 m asl are (natural?) treeline sites. Has the elevation of the tree line been influenced by humans? Are study sites selected along the elevational gradient influenced e.g., by grazing?*
*Furthermore, it is quite important to indicate slope and exposure of all study sites, as well as plant structure (height) of selected shrubs, because all these factors affect temperature that the plants experience (cf. Körner 2021).*

The sites at 900 and 1000 m a.s.l. are all alpine sites (without trees), but they are near-treeline, which is found at ~800 m a.s.l. in the oceanic study region and at ~1000 m a.s.l. in the continental study region). We included these aspects in the text (lines 108-111 of the revised version) "..., in accordance with the tree line in this region, which is located at about 750 to 800 m a.s.l.. In the continental region, we used 1000, 1100, 1200, 1300, 1400, and 1500 m a.s.l.. Here, the tree line is situated slightly higher, at about 1000 m a.s.l. (Rößler et al., 2008; Rößler and Löffler, 2007). Thus, all of our studied sites were located above the tree line." The treeline in Scandinavia has indeed been influenced by humans over long periods of time. Recently, only very extensive sheep herding is practiced, and *E. hermaphroditum* is usually not affected by grazing (see Weijers and Löffler, 2020), a fact which we included in chapter 2.2 Species and Specimens. For more information on climate and human influence on the treeline in the study regions, please also see Rößler and Löffler, 2007 and Rößler et al., 2008, now cited in line 110 of the revised text.
As suggested, we included height of the plants in Figure A1. Slope and exposure were not included, since they were naturally identical at the ridge positions with slope = ~0°, which indicates no exposure.

*p. 4, line 114: Please indicate mean height of shrub-canopy along elevational gradients.*

We included mean height of shrubs, lichens, mosses, graminoids, and forbs at the studied sites in a revised version of Figure A1.

*p. 5, line 128: DRO-dendrometers measure stem diameter variations and therefore are different from point dendrometers. To avoid misunderstanding, please correct.*

We agree that the term point dendrometer might be misleading and removed it from the entire text (see above).

*p. 5, line 135: Please consider that the "daily mean approach" yields time series of daily stem diameter variations, which include both water- and growth-induced diameter changes. Several authors found that bark is a highly hygroscopic tissue (e.g., Ilek et al. 2016, Gall et al. 2002). Therefore, stem diameter changes recorded by dendrometers mounted on dead outer bark as in this study may be affected by evaporation and absorption of water from the bark tissue. Dead outer bark*

*should be (completely) removed to reduce hygroscopic shrinkage and swelling of dead tissues on dendrometer records. To be able to unequivocally relate stem diameter fluctuations to environmental parameters and accurately determine stem water deficit, hygroscopic processes must be taken into account even when dendrometers were mounted over even thin dead outer bark layers (cf. Oberhuber et al. 2020). Therefore, authors need to demonstrate the lack of influence of changes in diurnal and seasonal air humidity on changes in stem diameter. Because hygroscopic effects could partly explain unexpected climate-growth relationships found in this study (i.e., influence of winter freezing conditions, soil moisture availability, solar radiation) I suggest adding records of daily (seasonal) changes of relative air humidity within study regions.*

Our dendrometers were not mounted on dead bark (see our comment above). Rather, the bark was removed before mounting the dendrometer. We are aware of the problems caused by placing the dendrometer on top of the bark and included a short text in the Material and Methods section, citing the suggested literature:
"During this process, we removed the outer bark to placed the sensor directly on the cambium. This ensures that hygroscopic shrinkage and swelling of dead tissues from the bark do not influence the diameter measurements. Such processes have been previously addressed in trees (Zweifel and Häsler, 2000; Gall et al., 2002; Ilek et al., 2016)."

*p. 5, line 144f: The phrases "…hydrological processes", and "reversible shrinking and swelling associated with the stem water deficit" are unclear, please reword.*

For clarity, we reworded this sentence as follows:
"To separate GRO from water-related expansion and contraction of the stem, we first excluded reversible shrinking and swelling associated with stem water fluctuations from the original data using the approach proposed by Zweifel (2016)."
The term "stem water fluctuations" now better describes what we meant here.

*p. 5, line 146: I suggest adding "the final growth curve", which was used to determine relationships between stem growth and environmental variables.*

This must be a misunderstanding possibly due to inconsiderate wording. "The final growth curve" does not refer to one individual curve, but rather to the final curves for each individual specimen. All of these curves were used to determine relationships between stem growth and environmental variables. We reworded this in the text:
"... the final growth curves were derived from the original measured data..."
Additionally, we included the described cumulative curves in Figure 3.

*p. 6, line 169-174: Sentence starting with "Buchwal et al. (2013)…." and the following sentence belong to discussion section.*

We disagree. This sentence belongs to the explanation of our methical approach regarding the specimens that did not experience any growth, and is therefore important at this point.

*p. 6, line 174: Please explain "non-growing seasons" or reword.*

We reworded this sentence:

"We did not calculate growing seasons for these years, and the analyses proceeded Separately"

*p. 6, lines 184ff: Please add manufacturer of soil moisture and radiation sensors. Furthermore, was the radiation sensor mounted on an open site or below "canopy".*

Our loggers as well as all temperature, soil moisture and global radiation sensors were manufactured by ONSET which is indicated in the text (p. 7, line 196), Furthermore, the global radiation sensor was mounted close to the plant and was not affected by the canopy. We edited the text accordingly:

"Additionally, we measured the shoot zone global radiation (W/m²) at 1 cm above the ground surface in close proximity to the plant (hereafter GRSZ) using ONSET's HOBO type SLIB-M003 silicon pyranometer (±10 W/m² accuracy). We made sure that those measurements were not affected by the canopy."

*p. 7, Figure 1: I suggest showing not only daily mean values of shoot and root zone temperatures but also daily mean maxima and minimum temperatures.*

We revised Figure 1 to include daily mean maximum and minimum temperatures.

*p. 8, Legend of Table 1: "inter-stem variability" or "site variability"? Please check.*

"Site variability" is the more accurate wording here. Was changed accordingly.

*p. 9, line 205: Has the abbreviation GDD10 been explained before?*

GDD10 stands for the number of days with shoot zone temperature > 10 (growing degree days). The abbreviation is explained in Table 3. For clarity, we added the explanation here as well.

*p. 9, lines 202-211: This paragraph belongs to results section.*

We disagree. This paragraph describes our collected data (Material).

*p. 9, line 205: Do authors have any explanation why temperatures vary only slightly between sites? Along an elevational gradient spanning 500 m (900 to 1400 m asl and 1000 to 1500 m asl) and taking a temperature lapse rate of 0.6 °C per 100 m elevation into account, a temperature difference of 3 °C between the lowest and highest study site is to be expected. Furthermore, also soil moisture shows low variability although selected study regions belong to the humid and continental climate zone. Please discuss these findings.*

There is a clear elevational temperature gradient (see our previous comment regarding p.1, line 15), both for near-ground air and soil temperatures, but the difference between the lowest and highest study sites is on average slightly lower than proposed by the linearly regressed adiabatic lapse rate, here. We expanded Figure A1 to include the environmental conditions along the elevational gradient. The reasons are daily variations in the adiabatic lapse rate induced by nocturnal inversions, and variations over specific weather phenomena

and seasonality (e.g. Löffler et al. 2006; Pape et al. 2009; Wundram et al. 2010). We included this in line 216. In general, our results highlight that the measured near-ground conditions follow common patterns of ambient air temperature measured at standard heights (e.g. 2 m above ground) only partially. Soil moisture shows no elevational gradient and overall low short- and medium-term variability, but there is a strong seasonality in the soil moisture data, with clear indication of winter frost droughts at the alpine ridges. All studied alpine ridges from oceanic and continental climate regions had similar patterns along the elevational gradient. We used this above discussed finding to explain missing elevational relations of growth and soil moisture (see our previous comment regarding p.1, line 15: Alpine ridge sites are often believed to be dry as to the wind-exposed nature. As has been shown in previous studies (e.g. Löffler 2005), soil moisture is not a limiting factor during the summer period, when plants are exposed to atmospheric summer drought, and this decoupling of the soil
from the atmosphere is due to a dense lichen carpet, which dries out, has no root coupling to the ground like other plants have, effectively limits descending water transport from the soil to the atmosphere, and as such reduces evaporative loss to a minimum. Along with our general novel findings on soil moisture as a strong driver of growth in *E. hermaphroditum*, we here show that frost droughts during winter are obviously restricting our shrub species, and that this phenomenon is similarly represented along all alpine ridges despite elevational and regional climate differences.)

*p. 9, lines 220ff: This sentence belongs to discussion section.*

We disagree. This sentence explains why and how we included estimated times of snow coverage into our analyses, even though we did not measure snow directly.

*p. 10, Figure 2: It is quite surprising that mean monthly shoot and root zone temperatures reach similar maxima (around 8-10 °C) during the supposed main growing season (June-September). This is most likely an effect of averaging temperature records along the elevational gradient or selection of sites showing different slope and/or exposure. The extended elevational transect would be an ideal*
*study design to investigate the influence of shoot and root zone temperatures on growth of this dwarf shrub. Unfortunately, limited sample size (one dendrometer record per elevation) precludes any analysis in this regard.*

Shoot and root zone temperatures follow similar patterns throughout the year (see Figure 1), which is why maxima are reached at similar times. However, the absolute values (see y-axis scale) are of course different. To make this more visible, we aligned Figure 2a and 2b, showing both graphs with the same y-axis scale. We agree that an extended elevational gradient should principally better address the thermal limits of shrub growth at its upper alpine distribution. Here, we will have further data from additional dendrometers which we have already installed (not included here due to
shorter time series). We disagree that the analysis of the elevational growth gradient was limited by the sample size, here. Actually, our result regarding the elevational gradient was that there was no thermally driven growth gradient. However, we agree that the study design could be expanded to include, for example, sites representing the thermal limit of the species.

*p. 11, line 245: "with the daily growth for each season" – Stem growth does not occur during all seasons (cf. Fig. 3)? Please clarify meaning of "daily growth" in this respect.*

To clarify, we changed "daily growth for each season" to "daily stem change for each season". This is consistent throughout the text: We defined growth in chapter 2.4 Analysis of Seasonal Growth Patterns as "growth-induced irreversible stem expansion".

*p. 11, Table 2: Please explain inverse relationship between growth cessation and total annual growth: the later growth stops during the year, the lower the annual increment?*

Please keep in mind here, that Table 2 presents the results of a multiple partial regression. As such, the resulting coefficients represent the relationship between the dependent variable (total annual growth) and each independent variable, while controlling for all other variables. The contribution of growth cessation to the model (see Table 2) was very low, indicating an overall low influence of growth cessation on total annual growth, which we indicate in the results section. Because of this role within the full model, such a conclusion is not reasonable from this result. As discussed in detail in the discussion (chapter 4.4 Growth Initiation and Cessation): Growth cessation was related to soil moisture and temperature, yet we could not identify a threshold. Peak growth usually occurred early during the growing season, indicating a lesser role of the late growing season for total annual growth.
In the discussion we included a short additional statement to clarify that our results highlight the complexity of such relations rather than support the assumption that total growth is directly linked to growing season length:
"Overall, our results regarding growth timing demonstrated the complexity of these processes and we cannot confirm a clear relation of growing season length and overall growth (Rammig et al., 2010; Blok et al., 2011; Prislan et al., 2019)."

*Also add explanation of abbreviation "Part" in legend.*

"Part" is not an abbreviation here. It is statistical terminology and stands for Part correlation, which is also sometimes referred to as semipartial correlation, which was added for clarity (Table 2). For interpretation: The semipartial can also be viewed as the decrease in $R^2$ that results from removing the independent variable from the model.

*p. 14, Figure 3: Determination of stem water deficit is most likely influenced by bark hygroscopicity (see comments to p. 5, line 135). I suggest removing Figure 3d. Furthermore, is "stem diameter variability" or are "stem radial variations" depicted in Figure 3b. In Table 4 values of "stem radial growth" are shown. Please clarify and be consistent in using "stem diameter variations" or "stem radial variations" throughout the text.*

Bark hygroscopicity should not be an issue, since we have removed the outer bark before mounting the dendrometers (see above). We therefore did not remove Figure 3d. Figure 3b depicts stem diameter variability, as stated in the Figure (y-axis) and the legend: "Averaged measured daily stem diameter variability and Gompertz models fitted to zero growth curves." Additionally, we changed "stem radial variations" in Table 4 to "stem diameter variation". In general, we used stem variation throughout the text to distinguish between the directly measured variations and growth, which we defined in chapter 2.4 Analysis of Seasonal

Growth Patterns: "total annual growth, defined as growth-induced irreversible stem expansion (GRO)".

*p. 15, Table 4: Please check value of "stem radial growth" in 2017, which amounts to "1485".*

Here, "." was missing. Stem radial growth in 2017 was 14.85.

*p. 15, line 285f, Figure 4: All 25 environmental parameters explained total annual growth? How is this to be explained? Not all relationships are statistically significant.*

This refers to the results of the regression model, which included all environmental parameters and was indeed able to explain annual growth to a very large extent:
"Together, all 25 chosen microenvironmental parameters (Table 3) were able to explain the total annual growth to a very large extent, even though not all parameters contributed significantly."
Further analysis and a breakdown of the contribution of each variable follows.

*p. 16, line 291f: "main growing phase during spring" – see comment to p. 1, line 18 (no spring growth according to Table 3). Please clarify.*

Mean growth initiation is indeed not in spring, but as suggested by the standard deviation, variability between years and specimens was high, with many individual specimens starting growth in spring or early summer. Additionally, some specimens started stem increment in spring, which was not termed growth here, because we defined growth as irreversible stem expansion following Zweifel (2016). We assume that this early stem increment might be related to refilling processes rather than formation of new xylem and cambial growth (see Mayr et al., 2006) and included this in chapter 2.4 Analysis of Seasonal Growth Patterns. To avoid confusion, we reworded the phrase:
"Comparisons of seasons also showed that micro-environmental influences had strong explanatory power during spring, when many of the plants started stem increment."

*p. 16, line 296: Is "R" Pearson correlation coefficient? If yes, change to "r" throughout the Manuscript.*

We changed "R" to "r" when we refer to Pearson's correlation coefficient throughout the text.

*p. 16, line 298: Influence of soil moisture availability on "growth processes" throughout "all four seasons" is highly speculative because (i) leaf water potentials were not determined, (ii) mean values of soil moisture given in Fig. 1 are not indicating drought stress during the growing season in both study regions, and (iii) most alpine plants have a small fraction of deep roots reaching up to 1 m depth. Furthermore, which "growth processes" were influenced throughout the year?*

The influence of soil moisture on growth is strongly emphasized by our results, which are presented here. The soils from our study regions are usually only about 30 – 35 cm deep (solid bedrock below), and we found maximum root density at about 5 – 20 cm. As such, deep roots can be ruled out here. Moreover, our measurements of the soil water content in 15 cm depth indicate that our soils in both climate regions are not experiencing any summer

drought (see above), but they are nonetheless influenced by drought during the winter months, caused by intensive soil freezing. During such winter periods, intensive soil frosts may severely limit water uptake and might result in frost drought. Our chain of arguments here is that an evergreen alpine species at the ridges, which is not covered by snow during the winter, is temporarily exposed to solar radiation and results in transpiration losses. The activated photosystem of the plant will deliver carbohydrates during the cold months (frost protection, reserve fabrics), being of advantage to an evergreen species, e.g. to grow better after a mild but sunny winter and spring. In reverse, if frozen ground limits water uptake, temporary snow cover and cloudy weather limit winter metabolism and little reserve fabrics have been accumulated during the previous season, growth will be limited during the current season. Altogether, our findings are quite unexpected at first glance, but turn out to be sound when taking all the different aspects into consideration. We discussed our above chain of arguments in details in the Discussion section (see below), being aware of the partly speculative nature of the interpretation of our statistical analysis. As such, we again argue that our findings give reason to rethink certain aspects of current knowledge based on our novel physiological records on stem diameter variations.

*p. 16, line 300: Sentence starting with "This result…" belongs to discussion section.*

We moved this sentence to the discussion section (chapter 4.4 Growth Initiation and Cessation).

*p. 18, Figure 5: It is highly surprising that Pearson correlation coefficients between "radial stem diameter and micro-environmental data" are almost all highly significant throughout the year including lag phases extending from three days to one year. Did authors check for normal distribution of data? Furthermore, Deslauriers et al. (2007) suggested including only the main period of growth to assess relationships between environmental variables and dendrometer data. Therefore, to determine climate–growth relationships, only the most linear growth phase should be considered, i.e. correlations should be calculated for e.g. the period ±14 days around the inflection point of the Gompertz model. Furthermore, to determine environmental influences on growth, radial stem increments extracted from dendrometer records according to e.g. Zweifel (2016) should be used in the correlation analysis rather than "daily values of stem variability" (which include reversible shrinking and swelling of the bark). Figure 5a: "Correlation with mean values": which mean values, annual means? Please add and also indicate meaning of significance symbols in legend.*

We agree that our results are quite surprising. The data are normally distributed. We chose not to reduce the data to the growing season, and rather include each season separately, because in our case there is important information to be gained for example from the stem shrinking observed during autumn and winter, which is not included in the growing phase, and which is also not included in "radial stem increments". Figure 5a refers to the whole year (as opposed to the seasons). Legend was changed. To convey this reasoning more clearly to the reader, we included a short statement in the Material and Methods section:
"Here, we chose to include the complete dendrometer series, rather than reducing the data to the main growing season (Deslauriers et al., 2007), to capture seasonal stem shrinking and swelling throughout the year."

*p. 20ff: The discussion sections needs to be substantially revised based on comments given above. Extended speculative data interpretations should be avoided.*

We revised the discussion section based on all the detailed constructive comments.

*p. 20, line 322: Sub-heading is not appropriate – please reword.*

We reworded the heading of chapter 4.1 ("Intra-annual growth patterns") to "Intra-annual patterns of stem change".

*p. 20, lines 324 and 330: Topographical effects on growth of Empetrum were not analysed in this study, because all sites were located on exposed ridge positions.*

It is true that we did not analyze topographical effects on growth in this study. Here, we refer to topography to highlight the fact that the studied exposed ridge positions are characterized by unique micro-environmental conditions (which we describe in the text). These conditions are a result of fine-scale topography within the studied regions, which distinguishes the ridge positions from their nearest environment. We clarified this in the text:
"In this study, we demonstrated that our focus species Empetrum hermaphroditum displayed distinct annual growth patterns in response to near-ground environmental drivers and in close accordance with distinct conditions caused by the local topographical characteristics of the studied wind-blown ridge positions. These unique microenvironmental conditions are a feature of the heterogeneous topography that characterizes alpine terrain (Scherrer and Körner, 2011) and are thus distinctly different from the surroundings. They include high exposure to global radiation and very little, temporary snow cover during the winter months, associated with very low temperatures (Wundram et al., 2010)"

*p. 20, line 335: "intra-plant growth variability" – in Fig. A1 and A3 inter-plant variability is shown, please check wording.*

Checked. "intra-plant growth variability" was changed to "inter-plant growth variability".

*p. 20, line 329: "very similar seasonal growth patterns" – In Fig. A4 I see large differences in stem diameter changes between regions in 2015 and 2018. Please clarify.*

That is true. Stem diameter changes vary in amplitude, yet, the seasonal pattern (shrinking during autumn and winter, growth during summer) and timing of the phases is surprisingly similar. We added the aspect of timing in the text ("... and led to very similar seasonal growth patterns and timing of growth (Fig. A4), ...")

*p. 22, line 386f: "negative correlation between stem diameter variation and number of snow-free days" – "stem diameter variation" is different from stem growth; please clarify.*

Throughout the whole text, we refer to stem growth as defined in the Material and Method section ("...defined as growth-induced irreversible stem expansion (GRO)"), which is based on Zweifel (2016). Stem diameter variation includes all measured stem changes. These terms are used accordingly throughout the whole text.

*p. 22, line 391: "to prepare for the following winter" – Why preparing for winter period? Please explain in more detail.*

We revised this sentence:
"In accordance with Buchwal et al. (2013), we assumed that during years of no apparent radial growth, dwarf shrubs might prioritize growth in the more protected and long-living belowground segments, instead of investing in the more vulnerable shoots."

*p. 22, line 408f: Regarding importance of soil moisture availability for growth, see comment to p. 16, line 298.*

As above, the key role of soil moisture in influencing growth processes is surprising and as such novel to the literature. Here, we carefully argue along with our strong statistical results acknowledging generally accepted physiological mechanisms. Our studied sites experience strong seasonality in water availability due to extreme winter conditions. The relation of growth and soil moisture availability, rather than temperatures, is one of the key findings presented here. We revised the text accordingly:
"This highlights a key role of the extreme winter temperatures, causing frost drought and consequent water stress even if soil water contents are usually sufficiently high during summer (Tranqulini, 1982; Mayr et al., 2006)."

*p. 22, line 413f: Onset of budbreak and plant growth in spring in the alpine zone is generally related to increase in temperature and photoperiod. A relationship between photosynthetic activity and growth onset is mere speculation and not substantiated by any previous study in the scientific literature (cf. Körner 2021).*

We agree that the interpretation of our statistical finding on the growth relationship to photosynthetic activity is speculation. We here propose a kind of nano-scale effect of soil thawing caused by radiation and respective heat transfer into the upper soil layer, which we assume to be in turn related to growth onset:
"Under snow free conditions, it is likely that in our study design global radiation is directly linked to soil thawing close to the soil surface, which was previously linked to growth onset (Descals et al., 2020)."
In our study design such effects were best captured by radiation sensors, since soil temperatures and soil moisture were measured at 15 cm below ground, and air temperature 15 cm above ground, respectively.

*p. 23, lines 418-440: Speculative discussion should be condensed. Regarding frost induced stem shrinkage in the literature please see and cite Zweifel and Häsler (2000) and King et al. (2013). Furthermore, regarding importance of carbon assimilation for growth onset and soil moisture availability for growth cessation, it is well known that in alpine plants no critical depletion of carbohydrates occurs in the seasonal cycle, and in temperate climate zones alpine plants initiate winter bud formation tightly coupled to the photoperiod, respectively (cf. Körner 2021).*

Zweifel and Häsler propose a mechanism of bark dehydration, which is responsible for winter shrinking, however, since we removed the bark, this is not the case for our data (see

above). However, we revised our text on frost induced stem shrinkage and included more literature on similar effects in trees:

"In trees, radial stem shrinkage has been related to sap flow and tree water content (Zweifel et al., 2006; Tian et al., 2019). When temperatures sink below approximately −5 ◦C, extra-cellular water begins to freeze inducing the osmotic withdrawal of intra-cellular water and thus cell and ultimately stem shrinkage (Zweifel et al., 2000; King et al., 2013)."

Regarding depletion of carbohydrates, we propose an early-season carbon surplus caused by photosynthetic activity in winter and early spring. In general, through such processes carbon assimilation may be greater or lower than demand for plant functions, resulting in periods of carbon surplus and deficit, respectively (e.g. Martínez‑Vilalta et al., 2016). There's never a critical depletion of non-structural carbohydrates (NSC), but still diminished resources at or shortly after snowmelt and, thus, at the onset of the growing season - due to respiratory consumption during winter (Körner 2021). A surplus of NSC, gained by photosynthetic activity in late winter/spring due to high solar irradiance is, thus, likely to be beneficial for growth throughout the growing period. We included this statement in line of the revised text:

"In general, carbon assimilation may be greater or lower than demand for plant functions, resulting in periods of carbon surplus and deficit, respectively (e.g. Martínez‑Vilalta et al., 2016). There's never a critical depletion of carbohydrates in alpine plants, but still diminished resources at or shortly after snowmelt and, thus, at the onset of the growing season - due to respiratory consumption during winter (Körner 2021). A surplus of carbohydrates, gained by photosynthetic activity in late winter/spring due to high solar irradiance is, thus, likely to be beneficial for growth throughout the growing period."

*p. 24, line 459: "photosynthetic activity throughout the year" – It is highly implausible for photosynthesis to occur throughout the year, i.e. at mean temperatures below -10°C (Table 1, Fig. 1).*

This is discussed in detail above (see comments on p. 1, line 19 and p. 22, line 413): Even though mean temperatures are below -10°C, higher temperatures are reached during the midday and there are short, warmer periods which experience soil thawing (see Figure 1, which we revised to include daily maximum and minimum temperatures). This is because of the high exposure and missing snow cover at our studied positions. Additionally, we used near-ground air temperature as potential driver of plant growth, which does not directly reflect the temperature in the plant. Especially during periods of clear sky and full global radiation, canopy temperature is known to be much higher than our temperatures measured at 15 cm above ground. We do not suggest that active cell growth occurs during these conditions. However, photosynthetic activity of vascular plants in late winter and under extreme conditions was reported before and our suggestion is thus backed by the literature (e.g., Starr and Oberbauer, 2003; Lundell et al., 2008; Semikhatova et al., 1992, also cited in the Discussion section).

*p. 24, lines 475ff: Climate warming has caused an increase in growth of shrubs in arctic and alpine ecosystems. I would assume that a further rise in temperature would primarily promote the spread of shrubs (and trees) in these temperature limited ecosystems rather than impair it.*

This is a common assumption in literature (e.g. Chapin et al., 2005; Post et al., 2019). In recent years, however, the spreading trend of shrubs in these ecosystems and the respective greening has been recognized as highly heterogeneous and complex, with the direction of change is still poorly understood (e.g. Elemendorf et al., 2012; Abott et al., 2016; Nielsen et al., 2017; Macias-Fauria et al., 2020). Thus, our view of temperature-growth relations is challenged. Competition might play a role here. Our results for *E. hermaphroditum* do most likely not apply to other species, but they suggest that at ridge positions, *E. hermaphroditum* growth might be limited with temperature increase, with important implications for species composition within these ecosystems.

*References cited:*
*Deslauriers A, Rossi S, Anfodillo T (2007) Dendrometer and intraannual tree growth: what kind of information can be inferred? Dendrochronologia 25, 113–124.*

*Gall, R, Landolt, W, Schleppi, P, Michellod, V, Bucher, JB, 2002. Water content and bark thickness of Norway spruce (Picea abies) stems: phloem water capacitance and xylem sap flow. Tree Physiol 22, 613–623.*

*Ilek, A, Kucza, J, Morkisz, K, 2016. Hygroscopicity of the bark of selected forest tree species. iForest 10, 220–226.*

*King G, P Fonti, D Nievergelt, U Büntgen, D Frank (2013) Climatic drivers of hourly to yearly tree radius variations along a 6 °C natural warming gradient. Agr For Meteorol 168:36-46.*

*Körner C (2021) Alpine plant life. Functional ecology of high mountain ecosystems. Springer.*

*Oberhuber W, M Sehrt, F Kitz, 2020, Hygroscopic properties of thin dead outer bark layers strongly influence stem diameter variations on short and long time scales in Scots pine (Pinus sylvestris L.). Agr For Meteorol 290, 108026.*

*Zweifel R (2016) Radial stem variations - a source of tree physiological information not fully exploited yet. Plant Cell Environ 775, 231–232.*

*Zweifel R, R Häsler (2000) Frost-induced reversible shrinkage of bark of mature subalpine conifers. Agr For Meteorol 102:213-222.*

Additionally included references in the revised version:
Descals, A., Verger, A., Filella, I., Baldocchi, D., Janssens, I. A., Fu, Y. H., Piao, S., Peaucelle, M., Ciais, P., and Peñuelas, J.: Soil thawing regulates the spring growth onset in tundra and alpine biomes. Sci Total Environ, 742, 140637, doi:10.1016/j.scitotenv.2020.140637, 2020.

Ilek, A., Kucza, J., and Morkisz, K.: Hygroscopicity of the bark of selected forest tree species. IForest, 10, 220–226, doi:10.3832/ifor1979-009, 2016.

King G., Fonti, P., Nievergelt, D., Büntgen, U., and Frank, D.: Climatic drivers of hourly to yearly tree radius variations along a 6 °C natural warming gradient. Agr For Meteorol, 168, 36-46, doi: 10.1016/j.agrformet.2012.08.002, 2013.

Körner, C.: Alpine plant life: Functional plant ecology of high mountain ecosystems, 3rd edition, Springer, Berlin, Heidelberg, 500 pp., 2021.

Lundell, R., Saarinen, T., Åström, H., and Hänninen, H. The boreal dwarf shrub Vaccinium vitis-idaea retains its capacity for photosynthesis through the winter . Botany, 86, 491–500, doi:10.1139/B08-022, 2008.

Martínez-Vilalta, J., Sala, A., Asensio, D., Galiano, L., Hoch, G., Palacio, S., Piper, F. I., and Lloret, F.: Dynamics of non-structural carbohydrates in terrestrial plants: a global synthesis. Ecological Monographs, 86, 495–516, doi:10.1002/ecm.1231, 2016.

Mayr, S., Hacke, U., Schmid, P., Schwienbacher, F., and Gruber, A.: Frost drought in conifers at the alpine timberline: xylem dysfunction and adaptations. Ecology, 87, 3175–3185, doi:10.1890/0012-9658(2006)87[3175:FDICAT]2.0.CO;2, 2006.

Prislan, P., Gričar, J., Čufar, K., Luis, M. d., Merela, M., and Rossi, S.: Growing season and radial growth predicted for Fagus sylvatica under climate change. Climatic Change, 153, 181–197, doi:10.1007/s10584-019-02374-0.

Rammig, A., Jonas, T., Zimmermann, N. E., and Rixen, C.: Changes in alpine plant growth under future climate conditions. Biogeosciences, 7, 2013–2024, doi:10.5194/bg-7-2013-2010, 2010.

Rößler, O., Bräuning, A., and Löffler, J.: Dynamics and driving forces of treeline fluctuation and regeneration in Central Norway during the past decades. Erdkunde, 62, 117–128, doi:10.3112/erdkunde.2008.02.02, 2008.

Rößler, O., and Löffler, J.: Uncertainties of Treeline Alterations Due to Climatic Change During the Past Century in the Central Norwegian Scandes. Geoöko, 28, 104-114, 2007.

Tranquillini, W.: Frost-Drought and Its Ecological Significance, in: Lange, O. L., Nobel, P. S., Osmond, C. B., and Ziegler, H. (Eds.), Encyclopedia of Plant Physiology, New Series: 12 / B. Physiological Plant Ecology II: Water. Relations and Carbon Assimilation. Springer Berlin Heidelberg, 79–400, doi: 10.1007/978-3-642-68150-9_12, 1982.

Zweifel R., and Häsler, R.: Frost-induced reversible shrinkage of bark of mature subalpine conifers. Agr For Meteorol 102, 213-222, doi:10.1016/S0168-1923(00)00135-0, 2000.

Zweifel, R., Item, H., and Hasler, R..: Stem radius changes and their relation to stored water in stems of young Norway spruce trees. Trees Struct Funct, 15, 50–57, doi:10.1007/s004680000072, 2000.
* * *
Reviewer 1 (Comment 2)

*Recommendation to revised version of the manuscript: Reject*

*Authors substantially revised their manuscript, clarified inconsistencies and they supplemented the text when this was necessary for a better understanding. However, I have still major concerns regarding methodological aspects as well as data interpretation, which need to be resolved to warrant publication.*

*Response to general argumentation of the authors:*
*"The authors are convinced that their extensive changes now justify publication in Biogeosciences, and by its a novel approach, strong methodology, unique dataset and unexpected results the paper will stimulate the discussion on our physiological understanding of arctic-alpine shrubs growth."*
*I readily agree with the authors that current knowledge should be questioned and discussed to enable scientific progress. But this discussion can only take place on the basis of solidly collected data. In the following I will once again present my arguments against the publication of this manuscript in its present (revised) form, as there are methodological deficiencies that significantly affected data collection and subsequent analysis.*

The authors would like to thank R1 for further constructive comments and suggestions to improve our manuscript. There were three aspects to be clarified: a) still existing doubts on accurate dendrometer measurements, b) representativeness of a dendrometer curve from a multi-stem shrub, and c) seeming mismatch of our new findings with results from (our) previous studies. We are positive that these concerns can be resolved, and we revised several parts of the manuscript, added three new figures to the supplement, and extended the interpretation of our findings, accordingly.

*Response to authors' comments on my previous specific points of concern (line numbers refer to the manuscript before revision):*
- *Averaging of dendrometer records: Authors clarified this point and they have sufficiently complemented the methods section.*
- *Mounting of diameter dendrometers: The authors now give more detailed information on mounting of dendrometers. Authors add in the Materials and Methods section that the dendrometers were not mounted on bark as I previously assumed from their description ("We mounted our dendrometers on one major above-ground stem…"), but "…we removed the outer bark to place the sensor directly on the cambium."*

*In this regard, I ask for clarification or elaboration on the following points:*
*First, please cite a study showing that dendrometers can be mounted directly on the cambial tissue without seriously affecting it. For several reasons, it is standard to mount point dendrometers on the living phloem, not on cambial tissue, which consists of a few cell layers only. Mounting on the cambium - if you can manage to do it that way - would inevitably lead to damage, mechanically or through dehydration. Furthermore, "girdling" of the phloem would block transport of carbon and hormones, which are necessary for cambium activity to occur.*
*Secondly, Figure A2 shows a photo of the way diameter dendrometers were assembled in this study. I admit that the resolution does not allow a clear statement, but it looks like that the diameter dendrometer was mounted directly on the stem without removing the outer bark. However, authors state in their reply that "In general, Empetrum hermaphroditum has a very thin bark, which is easily removed without the danger of damaging the cambium." That's fine, but I wonder, how authors could manage to mount DRO diameter dendrometers directly on the cambium. This type of dendrometer consists not only of a circular sensor head with a*

*diameter of c. 5 mm, but also of a rectangular fixing plate to be mounted on the opposite side of the sensor head. By default this plate is c. 2 cm long and 5 mm wide. Hence, this part of the stem should also have been removed (without damaging the cambium!) to ensure that hygroscopic effects are not influencing dendrometer records.*

R1 is right, that measurements on the cambium are hard to be implemented. We rephrased our text accordingly to avoid further misunderstandings. So far, to our knowledge, the use of dendrometers for the study of stem physiological activity (including growth) has been restricted to trees. Mounting dendrometers for the first time to dwarf shrub stems was a challenging task and indeed led to several ideas and potential technical solutions to be tested. Our dendrometer project initially started in 2008, many years before the here referred study period. During a test period, we developed the study design in different alpine regions. We invested in trials over several years to a) choose the best sensor type, b) find the final best option of mounting the dendrometers onto the shrub stems, and c) to proof that we measure active physiological activity (cambial activity) instead of passive swelling and shrinking of dead tissue. All the dendrometers included in our recent manuscript were running for at least one year before the start of the study period in 2015, to ensure that they produce meaningful data and that growth of the sampled plants is not impaired by the dendrometer mounted to the stem (added in the manuscript: "To ensure that the dendrometers run properly and produce exploitable data unaffected by the mounting of the dendrometers and bark removal, we tested the study design for several years, before selecting data series for our analysis."). As such, some stems were too young, and too soft and thin and died as a result of the "surgery", some stems suffered from the intervention (e.g. leaf loss). In both cases dendrometers were mounted to another specimen and the biased data were not used for analyses. Overall, branch structure and tissue properties of Empetrum hermaphroditum proved to be well suited for the installation and proper functioning of the chosen sensor type. From the reviewer's comment we assume that the procedure of mounting the dendrometers itself needs some clarification. We think that our new supplement figure A will help the readers follow our methodological and technical procedure based on the terminology used in our text. We removed dead outer bark material (periderm) directly where the sensor and the fixing plate were placed (the reviewer´s description of the sensor is accurate). As such, we ensured close contact of the dendrometer with the stem and minimized the effects of hygroscopic shrinkage and swelling of the dead outer bark on the final records (e.g. Oberhuber et al., 2020). Our overall aim was to get as close to the living tissue as possible. We made sure not to remove more protective bark material than necessary, and we tried to avoid damaging the living tissue (see anatomical structure in micro-slice picture in our new Fig. A). We are aware that removing the bark might be accompanied by a small wound to the stem. If we had any indication suggesting that continued life and growth were affected during our trial phase, we removed the specimen from the dataset, and mounted the dendrometer at another specimen. As such, the twelve specimens on which our recent manuscript is based, are actually part of a much more comprehensive study.

In general, the procedure we replicated on shrubs is common practice in trees (e.g. Oberhuber et al., 2020; Wang et al., 2020; Grams et al., 2021). However, the process of removing the dead outer bark was greatly facilitated by the unique anatomy of E. *hermaphroditum* in comparison to trees: Our shrubs are characterized by a thin but clearly pronounced living tissue, and selective removal of the outer bark is easy and can be done

without the use of special equipment or cutting (new Fig. A). In most cases, parts of the successive periderms of the rhytidome probably down to the outer phellem were easily removed with the outer bark, when the stem was moistened before the "surgery", as suggested by the anatomical structure shown in Fig. A. Usually, the procedure was not accompanied by any leaking fluids from the plant, proving that the living tissue was not damaged. In comparison to similar interventions on tree stems, removing the outer bark on our shrubs was, on the one hand, easy to handle as to the papery structure of the bark, but on the other hand, also tricky to see with bare eyes as to its micro-scale structure.

(Fig. see Appendix A to the revised manuscript (Fig. A2))

**Figure A: Radial micro-slide of a stem from Empertum nigrum ssp. hermaphroditum (1:1000 magnification). The outermost layers of the papery outer bark, which repeat the successive pattern of the shown periderm, got lost while cutting. In our dendrometer approach, we removed the outer layers of the bark, most likely down to the phellogen. As shown here, the loose bark structure allows removal without severe damage of the inner tissue. We aimed at mounting our dendrometer sensor as close to the still protected cambial zone, to achieve data on physiologically active stem diameter variability such as growth, excluding swelling and shrinking of the passive outer bark tissue.**

To clarify the mounting process in the manuscript we specified:

"We mounted our dendrometers on one major above -ground stem horizontal to the ground surface on randomly chosen specimens, which were as close to the assumed root collar as possible. During this technical process, we removed the dead outer bark to place the sensor as close to the living tissue as possible, following a common practice for dendrometer measurements of trees (e.g. Oberhuber et al., 2020; Wang et al., 2020; Grams et al., 2021). This ensures that hygroscopic shrinkage and swelling of dead tissues from the outer bark do not influence the diameter measurements. Such processes have been previously addressed in trees (Zweifel and Häsler, 2000; Gall et al., 2002; Ilek et al., 2016), and comparative studies revealed a complex interplay of xylem as well as phloem growth and pressure induced size changes, which simultaneously affect radial stem change and are thus captured by the dendrometers (Turcotte et al., 2011; Zweifel et al., 2014b; Oberhuber et al., 2020)."

To further highlight our findings regarding water induced shrinking and swelling and growth, we would like to present a closer look at our raw data in a new supplement figure (Fig. B). Here, we show that stem diameter changes during winter and spring are clearly coupled with soil moisture as well as soil freezing and thawing effects, suggesting that we indeed measured effects of active physiological stem shrinking and swelling. During the main growing phase however, our dendrometers captured a clear phase of stem increment, independent on available soil moisture, which we interpreted as growth.

(Fig. see Appendix A to the revised manuscript (Fig. A7))

**Figure B: Raw data for one exemplary specimen from the Vågå/Innlandet region at 1500 m a.s.l. Here, we present hourly data of stem diameter changes and the respective micro-environmental conditions. The four sections show important phases of the annual stem diameter variability and their relation to the micro-environment in detail. Coupling of soil moisture and stem diameter during the winter and spring months, when water induced stem swelling and shrinking occurs, and decoupling during the main growing phase is clearly**

**evident. Additionally, the direct response of stem diameter to singular soil thawing events in winter is clearly visible in the curves.**

*Limited sample size, i.e., one dendrometer record per plant and elevation:*
*In their reply to this major issue, authors argumented that "Actually, our result regarding the elevational gradient was that there was no thermally driven growth gradient."*
*The presentation of unexpected results is not an argument in favour of the small sample size.*

*Furthermore, dendroecological studies on* E. hermaphroditum*, which two of the authors of this manuscript co-authored (Löffler was co-author in all papers cited below), revealed high intra-plant growth variability and authors also pointed out the necessity of a high number of samples for determining radial growth of the dwarf shrub under study.*

*Main points of previous dendroecological (i.e., tree ring) studies:*

*Bär et al. (2006) found that "*E. hermaphroditum *shows highly individual growth histories. Thus, cross-dating of growth curves is restricted to several radii within an individual and to mean curves of individuals growing at the same micro-site. Wedging rings and missing rings as well as eccentricity and asymmetric geometry of the stem constrict the synchronisation of growth curves."*

*Bär et al. (2007) pointed out that "For a proper synchronization of the growth rings, serial sectioning was applied in order to deal with the high internal growth variability and the high proportion of discontinuous rings."*

*Bär et al. (2008) stated in their last sentence: "Hence, carefully synchronized and well replicated ring-width series of dwarf shrubs from alpine regions can be used as sensitive indicators for reconstructing past climate in vast regions beyond the polar and alpine tree limits."*

*Furthermore, in lines 336ff of the study under review authors state that "In contrast to the oceanic-continental gradient, our study showed high inter-plant growth variability (Fig. A1 and Fig. A3), which has been previously described in* E. hermaphroditum *(Bär et al., 2008) and could be a result of the nanoscale of internal growth variability within the multistemmed plant itself (Bär et al., 2007)."*

*Therefore, it seems to be quite obvious that a single point measurement (or diameter record in this case) does not represent radial growth (and hence intra-annual growth patterns derived from it) of a multi-stemmed shrub at a given elevation. Extremely low growth rates (< 100 µm) are likely to increase the uncertainties of a "single measurement".*

Addressing the limited sample size along the elevational gradient, there seems to remain a misunderstanding. We agree that the sample size is too small to make any overall assumptions about the elevational gradient. Therefore, in our manuscript we do not consider the results regarding the elevational gradient among main results of our study. Instead, we would like to highlight here that we found surprisingly similar annual growth patterns and

statistically significant response patterns to the micro-environment in twelve individually sampled plants from differing sites along an elevational gradient (see manuscript): "These conditions varied comparatively little between the study regions (Fig. 2 and Fig. A1), and led to very similar seasonal growth patterns and timing of growth (Fig. A6) …".
We would like to highlight here that our approach is fundamentally different from traditional measurement methods for shrub growth in that we are observing site-specific processes of stem diameter change, while the ring-width-approach from previous studies looks at the results of these processes (ring widths). The traditional approach is usually presented with a larger sample size and does not include variability between micro-sites or individual specimen. Here, we focus on fine-scale patterns instead, which we found surprisingly synchronized, due to the similarities of the environmental conditions, which are a result of the topographical position. Our sampled specimens seem to be uniquely affected by these specific micro-environmental conditions found at the sampled ridge positions. Here, further comparative studies are necessary, to clarify how growth patterns might differ in a heterogeneous alpine environment e.g. along topographic gradients. We are currently working on these questions within our research project. However, the findings presented here already highlight the strength of the measuring approach in identifying common fine-scale patterns across sites. Also, dendrometer measurements on trees have proven that meaningful results can be drawn from comparatively small sample sizes per species (e.g. Duchesne et al., 2012 (n = 3); Liu et al., 2019 (n = 11); van der Maaten et al., 2018 (n = 5)).

High inter-plant growth variability holds true, as seen from the magnitude of observed growth varied between sites. Overall patterns, including winter stem shrinking and summer growth, however, were found to be similar in the majority of sampled specimens. Such patterns have not been discussed previously in literature, and they cannot be derived using traditional sampling methods, including the ring-width approach used by Bär et al. (2006, 2007, 2008). Here, it is important to distinguish between growth derived from wood anatomical traits (growth rings), and the growth processes measured by dendrometers. A comparative study would further validate our dendrometer approach. We therefore simultaneously sampled several specimens from the same positions studied with our dendrometers and conducted micro-slice based anatomical measurements on growth rings, using the same approach from our earlier studies on *E. hermaphroditum* (Bär et al. 2006). We are currently working on a study attempting such a comparison, and first results suggest that the radial growth measured by our dendrometers is indeed mirrored in the wood anatomy data (see Fig. C).

We are aware of the intra-plant variability, which might affect our results. However, the fact that we were able to derive meaningful growth curves, which followed similar patterns from 12 individual specimens suggests that these patterns represent an important aspect of shrub growth. Additionally, our ring -width data presented in the new Fig. C was derived from multiple stems (following Bär et al., 2006 and 2007). Here, high synchrony with the stem diameter change derived from the dendrometer data further validates the approach.

(Fig. see Appendix A to the revised manuscript (Fig. A8))
**Fig. C: Comparison of annual growth measured using dendrometers (see Material and Methods), and ring width derived by measuring annual growth rings of 12 specimens from similar positions on exposed ridges, presented here as ring width index (RWI). Ring width was measured from multiple micro-slices per specimen (following Bär et al., 2006) (A), (B) shows free atmospheric air temperatures measured at 2 m above ground in both study regions. Such**

**temperature data is commonly used for comparison of climate-growth relationships in dendroecological studies. With this figure we aimed to reproduce previous studies (Bär et al., 2006 and 2007) for comparison with our dendrometer measurements.**

*Frost drought as a major determinant of shrub growth: In their reply authors state that "we here show that frost droughts during winter are obviously restricting our shrub species, and that this phenomenon is similarly represented along all alpine ridges despite elevational and regional climate differences."*

*If frost drought in late winter is a major issue, which significantly affects growth of* E. hermaphroditum*, how can it be that at the same time this shrub species "remains photosynthetically active during the snow-free period" (see lines 18ff)? As a result of freezing temperatures water transport is either severely reduced or completely interrupted – this would certainly impair carbon assimilation. I would also expect that stomata are closed to prevent excessive water loss as long as soils are frozen. Authors are also stating in their reply that "intensive soil freezing" occurs during winter months. Therefore, please show data or cite a paper that supports your interpretation, i.e., relevant carbon assimilation is possible during periods when soils are frozen.*
*Furthermore, did authors observe any leave damages caused by frost drought, i.e., browning and subsequent shedding of leaves in spring, which would indicate that severe drought stress occurred during winter (lines 344ff: "..and frost-triggered droughts might result in tissue damage caused by an internal water deficit.") . If winter drought is an important issue for this shrub species as suggested by authors, I would expect that at least at the highest elevation and in the more continental study region signs of frost drought are clearly visible.*

We did indeed observe the described effects of frost drought, i.e., browning and subsequent shedding of leaves in spring in several cases after harsh, cold and stormy winters, but usually the plants at the ridges were affected only partially (we did not observe entirely dead specimens). Such effects did not follow the expected patterns suggested by the reviewer. Instead, lower alpine elevation sites had strongest indications (in East and West), and this observation match stronger protective icing effects with elevation and the reverse effects of inversion weather conditions, with the lowest temperatures at the low-alpine ridges (Löffler et al. 2006).

In general, we agree that stomata would be closed to prevent excessive water loss as long as soils are frozen. However, if our evergreen species is forced to photosynthetic activity during clear sunny weather conditions in the winter months, this means that in order to effectively use the relatively high radiation measured during this time, the stomata will open for CO2 intake and transpiration loss. This will make the plant highly dependent on the availability of liquid water from the roots and vulnerable to drought frost damage, when ground frost is severe. The strong dependency of stem diameter variability on liquid water availability is clearly visible in our raw data (see new Fig. B): During the winter months stem increment is clearly coupled with soil moisture and individual thawing events.

 As such, we expanded on the text to further explain and strengthen our interpretation of the observed results during the winter months:
"Our findings regarding seasonally differentiated response to near ground environmental conditions (Fig. 5) highlight the importance of winter conditions for early growth. This

indicates that for our sampled evergreen species at the chosen sites, which experienced only short periods of snow cover that otherwise would be likely to influence the growth response, the degree to which photosynthetic activity was energetically effective in synthesizing carbohydrates during the winter months was especially important. Such continued activity was found in *E. hermaphroditum*, as well as several other evergreen shrub species before (e.g. Bienau et al., 2014; Wyka and Oleksyn, 2014; Blok et al., 2015). Photosynthetic activity is forced due to exposure to high solar radiation reaching the evergreen plants at the ridge positions where a protective snow cover is missing. This causes continued water transport under extreme temperatures, increasing the risk of cavitation (Tyree and Sperry, 1989; Venn and Green, 2018). Long and severe ground frosts might limit access to soil moisture, and frost-triggered droughts might thus result in tissue damage caused by an internal water deficit (Mayr et al., 2006). However, *E. hermaphroditum* at our studied sites proved mostly frost hardy, drought tolerant and highly adapted to these conditions (Carlquist, 1989; Hacke et al., 2001), with winter stem diameter change closely linked to soil moisture availability and singular thawing events, suggesting that the sampled specimens were able to utilize available liquid water even under extreme conditions (new Fig. B*)."

A clearer understanding of the stem anatomy of *E. hermaphroditum* might also be of help here, which is why we included the following in chapter 2.2 Species and Specimen:
"The species belongs to the Empetraceae family of heathlike shrubs. Its stem anatomy was described by Carlquist (1989) and is characterized by a narrow vessel diameter, which can be interpreted as a form of adaptation to drought or physiological drought due to cold as it impedes embolism formation. In general, the family is known to match extreme environments by adapting stem anatomy (Carlquist, 1989)."
This highlights the adaptive capability of species of the Empetraceae family, including adaption of stem anatomy to winter drought caused by soil freezing.

*A clarification is needed as to why in a previous dendroecological study co-authored by Pape and Löffler contradictory results regarding growth limitation of hermaphroditum by climate factors were found. Bär et al. (2008) reported that "This study indicates that mean summer (June–August) temperatures determine the width of the growth rings of Empetrum hermaphroditum irrespective of topoclimate."*

*It is highly implausible that determination of climate-growth relationships based on dendroecological techniques (inter-annual) vs. dendrometer records (intra-annual) lead to such contrary results as reported in this study. Please clarify and discuss this issue in a revision.*

Both studies (Bär et al., 2008 and our present study) do not only differ in the deployed technical approach. Instead, the previous study used a different dataset, which is only partly comparable to the one used here. For instance, the samples presented in our new paper were taken from two, instead of one, study regions, spanning a much wider environmental gradient. At the same time, we restricted our sampled sites to exposed ridge positions in the new dendrometer study, which experience very similar micro-site conditions, regardless of the regional climate signal. We discuss this in details in the first paragraph of the discussion section. Soil moisture is one of the key factors determining these micro-conditions (mainly due to snow distribution). Because such fine-scale soil moisture measurements were not

included in the study from 2008, their effects could not be assessed. As such, the results of both studies as a whole might suggest that while temperatures play a key role in determining growth processes, on a micro-scale soil moisture and snow conditions might be more important within a topographically heterogeneous environment, especially at positions where snow drift plays a major role. We are currently exploring this topographical variability to be presented in future publications. In general, we considered comparability of our dendrometer approach to previous study designs for several years, which is why we also collected ring width data at our studied sites. We included some of this data in the new figure C and are currently working on a comparative study. In the manuscript we included a short discussion of the figure and its implications in the discussion section:

"Thus, in our studied alpine environment, we cannot confirm high temperatures as the main general driver of shrub growth, as was assumed in several previous studies (i.e., Elmendorf et al., 2012; Hollesen et al., 2015; Ackerman et al., 2017; Weijers et al., 2018b). Such previous studies commonly used free atmospheric air temperature measured at 2 m above ground and ring width measurements (e.g. Bär et al. 2007 and 2008). A direct comparison of annual growth derived from our dendrometer measurements and such ring width measurements at the studied sites revealed high synchrony (Fig. C*). Here, the ring width data was linked to summer temperature as well, suggesting that the assumed temperature-growth relation holds partly true at our sampled sites (Fig. C*). Dendrometer data have the potential to reveal much deeper insights into complex functional aspects of growth, and in combination with on-site environmental data might help rethinking climate growth relations. Further studies are necessary here to fully explore how dendrometer measurements compare to traditional measurement methods and which additional information can be gained. Still, the comparative data presented in Fig. C* clearly shows that both have the potential to reveal important aspects of stem variability and growth."

*Figure numbers will be adjusted in the final revised version of the manuscript.

*References cited:*
*Bär, A., Bräuning, A., and Löffler, J (2006): Dendroecology of dwarf shrubs in the high mountains of Norway – A methodological approach, Dendrochronologia, 24, 17–27.*

*Bär, A., Bräuning, A., and Löffler, J. (2007): Ring -width chronologies of the alpine dwarf shrub Empetrum hermaphroditum from the Norwegian mountains. IAWA J, 28, 325–338.*

*Bär, A., Pape, R., Bräuning, A., and Löffler, J. (2008): Growth -ring variations of dwarf shrubs reflect regional climate signals in alpine environments rather than topoclimatic differences, J Biogeogr, 35, 625–636.*

Additional references included in our newly revised manuscript (and above):
Carlquist, S.: Wood and Bark Anatomy of Empetraceae; Comments on Paedomorphosis in Woods of Certain Small Shrubs, Aliso, 12, 497–515, 1989.

Duchesne, L., Houle, D., and D'Orangeville, L.; Influence of climate on seasonal patterns of stem increment of balsam fir in a boreal forest of Québec, Canada. Agricultural and Forest Meteorology, 162-163, 108–114, doi:10.1016/j.agrformet.2012.04.016, 2012.

Grams, T. E. E., Hesse, B. D., Gebhardt, T., Weikl, F., Rötzer, T., Kovacs, B., Hikino, K., Hafner, B. D., Brunn, M., Bauerle, T., Häberle K.-H., Pretzsch, H., and Pritsch, K.: The Kroof experiment: realization and efficacy of a recurrent drought experiment plus recovery in a beech/spruce forest, Ecosphere, 12, doi:10.1002/ecs2.3399, 2021.

Hacke, U. G., Sperry, J. S., Pockman, W. T., Davis, S. D., and McCulloh, K. A.: Trends in wood density and structure are linked to prevention of xylem implosion by negative pressure, Oecologia, 126, 457–461, doi:10.1007/s004420100628, 2001.

Liu, X., Wang, C., and Zhao, J.: Seasonal Drought Effects on Intra-Annual Stem Growth of Taiwan Pine along an Elevational Gradient in Subtropical China, Forests, 10, 1128, doi:10.3390/f10121128, 2019.

Oberhuber, W., Sehrt, M., and Kitz, F.: Hygroscopic properties of thin dead outer bark layers strongly influence stem diameter variations on short and long time scales in Scots pine (Pinus sylvestris L.). Agric For Meteorol, 290, 108026, doi:10.1016/j.agrformet.2020.108026, 2020.

Tyree, M. T., and Sperry, J. S.: Vulnerability of Xylem to Cavitation and Embolism, Annu Rev Plant Physiol Plant Mol Biol, 40, 19–36, doi:10.1146/annurev.pp.40.060189.000315, 1989.

van der Maaten, E., Pape, J., van der Maaten-Theunissen, M., Scharnweber, T., Smiljanic, M., Cruz-García, R., and Wilmking, M.: Distinct growth phenology but similar daily stem dynamics in three co-occurring broadleaved tree species, Tree Physiology, 38, 1820–1828, doi:10.1093/treephys/tpy042, 2018.

Venn, S. E., and Green, K. Evergreen alpine shrubs have high freezing resistance in spring, irrespective of snowmelt timing and exposure to frost: an investigation from the Snowy Mountains, Australia, Plant Ecol, 219, 209–216, doi:10.1007/s11258-017-0789-8, 2018.

Wang, Y., Wang, Y., Li, Z., Yu, P., and Han, X.: Interannual Variation of Transpiration and Its Modeling of a Larch Plantation in Semiarid Northwest China, Forests, 11, 1303, doi:10.3390/f11121303, 2020.

Zweifel, R., Drew, D. M., Schweingruber F., and Downes, G. M.: Xylem as the main origin of stem radius changes in Eucalyptus. Funct Plant Biol : FPB, 41, 520–53, doi:10.1071/FP13240, 2014b.

\*\*\*\*\*\*\*\*\*\*\*\*\*\*\*\*\*\*\*\*\*\*\*\*\*\*\*\*\*\*\*\*\*\*\*\*\*\*\*\*\*\*\*\*\*\*

Reviewer 2 (Comment 1)

*Dear authors, dear Editors,*
*it was a pleasure reading the manuscript "A new mechanistic understanding of*
*ecophysiological patterns in a widespread alpine dwarf shrub – Refining climate-growth*
*relationships", which presents a pioneering work at the emerging interface of dendroecology*
*and wood anatomy (of which more could be added to the paper) that appears particularly*

*timely in the context of understanding the causes and consequences of "Arctic greening" (as we still don't understand the abiotic and biotic drivers of shrub growth at hourly rather than integrated ring width resolution). Furthermore, I was delighted to follow the interactive online discussion, during which challenging scientific questions were raised and – most importantly – answered.*

*The data used and methods applied (i.e. a network of hourly-resolved dendrometer and in-situ meteorological/environmental measurements from several Empetrum nigrum sites along elevational gradients in central Norway) provide detailed insights into the growth-climate response of an important alpine and Arctic shrub species. The two thorough replies to the critiques of R1 not only confirm the robustness of the study, but also contribute to an improvement of the quality of the manuscript, which clearly goes beyond previously published dendrochronological evidence of shrub growth.*

*Once all comments and suggestions of R1 are either considered or counterargued, my recommendation to the authors is to further improve the writing style of their paper (maybe even inviting further authors?), which is still rather vague and imprecise at several occasions. This being said, I have no doubt that a revised version of this manuscript will make a strong contribution to the wider community of global change ecology/biology and biogeography, and stimulate discussion within and between dendroecology and wood anatomy (where high-resolution dendrometer evidence from shrubs is still lacking).*

*The main reasons for publishing this study are at least two "firsts": 1) hourly-resolved shrub growth dendrometer in tandem with in-situ climate and environmental measurements, and 2) winter climate affects stem growth of one of the focal species of recent Arctic and alpine greening.*

*I am looking forward to read a fully revised article and, again, congratulate the authors to their achievements in what I consider a ground-breaking study that has the potential to stimulate further research.*

We would like to thank all reviewers for their contribution to the discussion and helpful comments on our manuscript.

We revised our manuscript accordingly, paying special attention to writing style and clarity, and have the revised version ready for submission. Furthermore, we believe we were able to address all issues raised in the new manuscript, as well as in our detailed answers to reviewer 1.

\*\*\*\*\*\*\*\*\*\*\*\*\*\*\*\*\*\*\*\*\*\*\*\*\*\*\*\*\*\*\*\*\*\*\*\*\*\*\*\*\*\*\*\*\*\*

Reviewer 3 (Comment 1)

In this paper, alpine dwarf shrubs are studied with the help of point dendrometers and micrometeorological measurements. The work definitely has a certain novelty, as the application of this set of methods to an alpine shrub is new, as well as the attempt to study dendrometer data on an hourly resolution. The work and its results fit into the current attempts to understand growth processes at high temporal resolution and thus get closer to the underlying physiological mechanisms (see also Zweifel et al., 2021).

I support the publication of the paper in Biogeosciences Discussions, however the current version of the manuscript suffers from some shortcomings that need to be addressed first.

*1) The title is far too bold and needs to be changed, as the paper does not present a new mechanistic understanding of ecophysiolopgical patterns (what is even meant by this term?). Novel aspects of the work are: the application of automated point dendrometers to shrubs and the approach of converting stem radius data into high temporal resolution growth data. It is a case study with limited general relevance, but nicely demonstrates the potential of the methods used.*

*A title such as the following proposal would better fit the content of the paper:*

*The application of point dendrometers to alpine dwarf shrubs - a case study to investigate stem growth responses to environmental conditions.*

Thank you for your thoughts on the title, we changed it to the suggested title: "The application of dendrometers to alpine dwarf shrubs - a case study to investigate stem growth responses to environmental conditions."
We removed "point" in accordance with Review1.

*2) The authors mix three currently available methods to separate dendrometer data into the components of irreversible growth and reversible water-related stem tissue dynamics.*

*First, there is the approach of Deslauriers et al. 2007 and Van der Maaten et al. 2016, which treat each individual day independently of the historical evolution of stem radius changes. Therefore, this approach records any absolute increase in stem size over 24 hours as growth.*

*This is in contrast to the zero-growth approach of Zweifel et al. 2016, where accumulated shrinkage must be replenished over longer periods of time (days and weeks) before any additional stem increase is considered growth. Essentially, the zero-growth approach assumes that no growth is possible during periods of stem shrinkage. All daily increments add up to an annual increment represented by the total annual stem size change measured by the dendrometer. In the case of the Deslauriers et al. approach, daily increments sum to more than the increment measured by the dendrometer over one year. The reason for this is that stem size increases during periods of stem shrinkage are counted as growth.*

*The third approach is the Gompertz growth function, which takes up the commonly assumed growth form over a season. The Gompertz function assumes constant and uniform growth throughout the growing season, which is clearly not true, as shown in Zweifel et al. 2021 and also in this paper (Fig. 3).*

*Further, the Gompertz function is not a reliable way to find the onset of growth because the Gompertz function is fitted to the original dendrometer curve and thus neglects that the stem is first rehydrated before growth begins. Furthermore, the nature of the Gompertz function implies a slow growth start at the beginning of the season, which obviously does not fit the growth pattern, as can be well seen in Fig. 3b. This approach leads to too early growth starts in spring and might also be the reason why the authors set a threshold for initial growth at 20% of annual ring growth.*

*Anyway, all approaches may have advantages and disadvantages, the problem with this work is that there is no clear line that tells me as a reader which analyses and which figures are based on which approach. Also, I don't see how the Deslauriers approach and the zero growth approach are compatible in the same study.*

Thank you for the comprehensive summary of the three approaches. This is indeed missing from our text. In general, the zero-growth approach proposed by Zweifel et al. 2016 was the main approach we used to calculate growth parameters. We did, however, apply the "daily mean" approach described by Deslauriers to calculate daily values from our measured hourly data. Following the zero growth assumption, we account for the fact that rehydrating before growth begins occurs by excluding this additional increment and starting to measure growth when the stem diameter exceeded the previous year's maximum (Fig. 3b). The Gompertz curve thus does not fit the pattern perfectly in winter when we recorded stem shrinking, but describes growth processes during the main growing season fairly accurately, which is why we think its use is justified to derive information on timing from the fitted models (Fig. 3b shows that the growth start suggested by our calculations from the Gompertz curves is reasonable). The use of the Gompertz approach to define growth start and end is well tested for dendrometer measurements (e.g. Duchesne et al., 2012; Liu et al., 2018; Drew and Downes, 2018; van der Maaten et al., 2018).

We made sure to clearly state (and justify) the use of these approaches in the Material and Methods section: "There are several methods currently available to separate growth from water-related expansion and contraction of the stem. To define total annual growth, we chose the approach proposed by Zweifel (2016), which excludes reversible shrinking and swelling associated with stem water fluctuations, assuming zero growth during periods of stem shrinkage. Total annual growth is thus derived from the original measured data by calculating the cumulative maxima (Zweifel et al., 2014a; Zweifel, 2016). We recorded additional stem increment before the start of the growing season in spring. Because this increment did not exceed the previous year's maximum stem diameter, we assumed that it might be related to refilling processes rather than formation of new xylem and cambial growth (see Mayr et al., 2006) and therefore, in accordance with the previously described zero growth assumption, did not define those processes as growth. Following this approach, we were able to define growth-induced stem increment during the main growing season. To further define this growing season and derive accurate dates for growth start and end, we fitted sigmoid Gompertz models to the resulting growth curves."

Additionally, we revised the legend accompanying Fig. 3b: "(b) Averaged measured daily stem diameter variability and fitted Gompertz models (a goodness-of-fit (GoF) measure was calculated using the least-squares method). Models were fitted to zero growth curves derived from the original measurements as cumulative maxima (thin lines), assuming zero growth during phases of prolonged stem shrinkage. In this way annual growth and, consequently, growth start is directly linked to growth during the previous year and additional rehydrating processes before the start of the main growing season are excluded."

We thus clearly convey to the reader how we combined the zero growth and Gompertz model approach to separate growth-induced and water-related stem diameter variation and accurately define the main growing season.

*3) The statement (L435ff) that winter shrinkage of woody stems has never been reported is false. See e.g. Winget & Kozlowski, 1964; Zweifel & Häsler, 2000; Sevanto et al., 2012.*

We included the suggested references and revised our statement from L435ff. We did not intend to state that winter shrinkage has never been reported for woody plants. However, a pronounced phase of winter stem shrinking in shrubs specifically (as found in our data) has not been described before (to our knowledge).The revised part now reads: "A phase of winter stem shrinking has been described in trees (Winget & Kozlowski, 1964; Zweifel & Häsler, 2000). The distinct and strongly pronounced phase we found in our sampled specimens, however, might be described as a unique feature of shrub growth, which we documented for the first time in this study."

*4) Legends of figures and tables must be completed with all abbreviations that occur. In addition, the data basis (model or measured data, temporal resolution) should be stated in each case. A legend must be readable on its own.*

We revised tables, figures and legends, keeping this in mind.

*5) The analysis in Fig. 4 states that the authors are able to perfectly predict annual growth from radiation with 100% accuracy! This must be wrong!!!*

The value $R^2=1$ resulted from rounding of the original value. We corrected this.

*6) Several recent studies show the importance of VPD for growth (Novick et al., 2016; Grossiord et al., 2018; Peters et al., 2021; Zweifel et al., 2021). It would add weight to this study if VPD were included. If I understand the measurement setup correctly, the authors have this data.*

We did indeed measure relative air humidity at some sites, from which we could infer VPD. We have, however, only one complete series for each region, with no information on how this data might vary between the sites. One of the strong points of the current study is the fine-scale environmental data measured directly at each site. We do not think we can match this scale for the whole study period with the VPD-measurements we currently have. However, we agree that this could potentially contain novel information and therefore would like to include it in a future study.

*7) It might be a matter of style, but why are so many results shown already in M&M?*

This mainly relates to the environmental data presented in chapter 2.5 Environmental data collection and growth conditions. We present this data in the Material and Methods section, since it mainly describes our collected environmental data and no results from our analysis.

*References*

*Grossiord C, Sevanto S, Limousin JM, Meir P, Mencuccini M, Pangle RE, Pockman WT, Salmon Y, Zweifel R, McDowell NG. 2018. Manipulative experiments demonstrate how long-term soil moisture changes alter controls of plant water use. Environmental and Experimental Botany 152: 19-27.*

Novick KA, Ficklin DL, Stoy PC, Williams CA, Bohrer G, Oishi AC, Papuga SA, Blanken PD, Noormets A, Sulman BN, et al. 2016. The increasing importance of atmospheric demand for ecosystem water and carbon fluxes. Nature Climate Change 6(11): 1023-1027.

Peters RL, Steppe K, Cuny HE, De Pauw DJW, Frank DC, Schaub M, Rathgeber CBK, Cabon A, Fonti P. 2021. Turgor - a limiting factor for radial growth in mature conifers along an elevational gradient. New Phytologist 229(1): 213-229.

Sevanto S, Holbrook NM, Ball MC. 2012. Freeze/Thaw-induced embolism: probability of critical bubble formation depends on speed of ice formation. Frontiers in plant science 3: 107.

Winget CH, Kozlowski TT. 1964. Winter shrinkage in stems of forest trees. J For 62: 335-337.

Zweifel R, Häsler R. 2000. Frost-induced reversible shrinkage of bark of mature, subalpine conifers. Agricultural and Forest Meteorology 102: 213-222.

Zweifel R, Sterck F, Braun S, Buchmann N, Eugster W, Gessler A, Haeni M, Peters RL, Walthert L, Wilhelm M, et al. 2021. Why trees grow at night. New Phytologist doi 10.1111/nph.17552

---

## Author Response (AR2)

"The manuscript entitled "The application of dendrometers to alpine dwarf shrubs – a case study to investigate stem growth responses to environmental conditions" addressed an important issue in relation to intra-annual growth dynamics in shrubs and its environmental drivers. The manuscript has received a plethora of relevant comments from previous reviewers, which have been carefully addressed by the authors. I highly appreciate the topic as it dives into the eco-physiological mechanisms which could further clarify the Artic greening observations, and enlighten us more on potential environmental drivers which regulate growth. After reviewing the manuscript, and enclosed referee reports, I however still felt like improvements could be made. Particularly in relation to structure, methodological explanations and the presentation of the results."

Abstract:
"Within the manuscript the authors mention multiple times that high-precision dendrometers have not been used on shrubs before. Although I agree that it is less common, the authors have already published a manuscript in the past where dendrometer measurements have been collected on shrubs (i.e., https://esajournals.onlinelibrary.wiley.com/doi/10.1002/ecs2.3688; but see also González-Rodríguez et al., 2017). I would thus argue that one should be more nuanced with these statements. Also, presenting this more as a case-study within the abstract would be better for the manuscript."

Thank you for the suggestions. The mentioned manuscript was not yet published when we first submitted this manuscript to Biogeosciences, which is why we emphasized the novelty of the method here. We rephrased the Abstract at several points, and it now reflects the state of the current research regarding dendrometer measurements of shrub growth more accurately. We also included our previous study in the Introduction.

Introduction:
"Within the first paragraph of the introduction it might be worth to spend some more time on explaining the observed greening patterns in more detail. Greening can occur because of higher photosynthetic activity of leaves, besides the general higher abundance of shrubs in an area (as noted by the authors). Although woody growth can be stimulated by having additional carbon assimilates, the link between growth and photosynthesis has received some critique which might be worth mentioning as an additional uncertainty (see Fatichi et al. 2019 New Phytologist: https://doi.org/10.1111/nph.15451). Here one could raise the question whether the growth is indeed favoured by similar environmental drivers as the ones postulated for driving greening."

"In the second paragraph I would emphasize that most of the referred studies use inter-annual variability of (stem/shoot) growth and that to fully understand the climatic response window we have to obtain intra-annual variability in growth patterns. This could then be combined with the statements made in the third paragraph about shrub-ring series, as right now this seems separated."

"Moreover, the authors should clarify the link between water use and growth. There is relevant literature on these issues which has not been cited or discussed. One could consider including some statements on mechanistic models showing the link between water relations and cell formation (i.e., De Schepper & Steppe 2010 Journal of Experimental Botany: https://doi.org/10.1093/jxb/erq018; Peters et al. 2021 New Phytologist: https://doi.org/10.1111/nph.16872; Cabon et al. 2020: https://doi.org/10.1111/nph.16456)."

"Also, the authors should explain in more detail what the physiological relevance is of understanding the swelling dynamics. Do the authors assume that it provides and indirect measure of transpiration and thus assimilation, or is this more related to identifying periods of water stress?
As a large emphasis is place on the physiological mechanisms within the manuscript, I believe that it should be clear to the reader as to why specific measurements are physiologically relevant. At the moment it reads more like a methodological argument (i.e., it is an easily extractable parameter), in addition to the fact that the swelling patterns are not clearly mentioned within the objectives."

Thank you for your thoughts on the Introduction. We carefully revised the text, including the suggested references (as well as some additional ones, relevant in this context). We additionally emphasized the importance of gathering fine-scale, intra-annual data, as well as physiological processes governing radial stem increment, including cell formation and water relations. Here, we restructured the original text substantially. However, as this study was mainly conducted as a case study we kept the methodical focus. As indicated below, we removed the Species section from the Materials and methods section and included some more detailed information on our focal species in the Introduction instead.

Regarding the main aims of the study, we hope it becomes clear from the revised text (in the Introduction, the Material and Methods, as well as in the Discussion) that we aimed to find common growth patterns, independent from the variability in environmental conditions across the elevational gradient, yet closely linked to the specific characteristics of the positions at exposed ridges, which did not vary between sites.

Materials and methods:
"When reading the MM I find a slight disconnect with the introduction. For example it is not mentioned why having multiple elevational bands for monitoring is important. Especially, as noted by referee 1, there is only 1 individual per site which is critical. This requires clarification within the introduction, as apparently the variability in environmental conditions was more important when setting up the sites than the replication of shrubs per site. Also, within the materials and methods there is the species section where the authors explain the selection of the species with relevant literature. In my view this belongs in a shortened version into the introduction, as it explains the reader the merits of the study (i.e., why select this species)."

"In accordance with referee 3, I am still slightly confused about the different methods used for extracting growth from water relations. From the response of the authors it seems they extracted daily growth rates using the method proposed by Deslauriers and excluded solely rehydration patterns before growth initiation (based on the zero-growth concept; Zweifel et al. 2016). This approach however fully ignores the fact that stem shrinkage can also occur during the growing season, where the approach by Deslauriers would again overestimate the annual growth rates, compared to the method proposed by Zweifel. As the method by Zweifel allows for the extraction of hourly/daily growth patterns as well (see R package presented in Knüsel et al. 2021 Forests: https://doi.org/10.3390/f12060765), I am really confused as to why the author combined these two approaches, without concretely testing the difference. Has this been done? Also, in the response to the comments from referee 3 they noted that they combined the zero-growth and Gompertz model, while in the methods they clearly state they use the dendrometeR package to extract peak growth rates. Please provide a more structured and clear line as to which growth and water related parameters are extracted and how. Also, again little room is provided to explain the exact "swelling" (or water related) parameter which was mentioned to be relevant within the introduction."

"Finally, for the methods I found it distracting to already have results presented in such great detail (as noted by referee 1 and 3). One can decide to either make a dedicated paragraph within the results section to explain the environmental variability, or one can move these figure to the supporting information."

Regarding our methods for separating growth from water relations, there still seem to be some major misunderstandings, which is why we revised the text substantially. We used the dendrometeR package solly to derive daily values from our hourly stem diameter measurements, as daily fluctuations were not our main interest and aggregating the hourly data facilitated calculation processes within the R statistical software. While there are multiple other tools available in R to do this, the package provided a quick alternative which fitted the structure of our data. For all further analysis we based our understanding of growth on the approach proposed by Zweifel (2016), assuming no growth during periods of stem shrinkage. However, this does not mean that shrinking processes were not included in our analysis, we simply chose to separate them from growth using the described approach. Here, we revised the chapter entirely to avoid further misunderstandings regarding our methods. We made sure

that the introduction of each of our growth parameters is followed by a detailed explanation, including definition, approach and physiological background.

Additionally, we moved large parts of the Material and Methods section to the Introduction, as well as to a new paragraph within the Results chapter, as suggested.

Results:
"Within Figure 3 the stem water deficit is clearly presented, yet no analyses has been performed on these patterns outside of the winter months (which has not been clearly explained in the MM). This seems strange as within the introduction the authors state: "Additionally, the time series derived from dendrometer measurements offer information not only on radial stem growth, but also on stem water relationships with higher quality and resolution than previously attainable". If there is no intention to look at water relations or swelling/shrinkage outside of the winter months, I would specifically mention this and explain why, instead of highlighting it as generally relevant physiological information. This lower relevance of water relationships extracted from the dendrometers is again highlighted by the fact that within Table 4 the authors mention in the header "Growth parameters" while it also includes some shrinkage parameters (this is also the case for Figure 6)."

"The analyses and the representation of Figure 5 could be improved. On the x-axis a fixed shift is presented, where the selection of the intervals has not been explained in the methods. Also, it is not clear what is meant with stem variability. Does this refer to growth or another parameter extracted from the data? Moreover, there is the possibility to perform continuous moving-window correlation analyses (as presented in Castagneri et al. 2017 Annals of Botany: https://doi.org/10.1093/aob/mcw274) which could be more informative then fixed periods."

"Finally, within the results it is often unclear whether correlations have been performed with daily or annually aggregated values. This should be clearly described in all figure legends and within the descriptive text, as these analyses operate with highly different sampling sizes."

We revised the Results chapter, including some parts previously included in the Material and Methods section. While it is true that we focused more on growth than on the described shrinking processes, swelling and shrinking were included in multiple of our analysis. For example, shrinking parameters were included in all of the regression analyses. Here, the term "growth parameters" might be misleading, which is why we revised the text, differentiating between growth parameters and parameters of stem change. From the calculated stem water deficit (previously Fig. 3, new Fig. 1) it becomes clear that the main phase of stem shrinking occurs during the winter months, which is why we focused on this phenomenon later on (see Discussion).
Additionally, we replaced Fig. 5, including a moving window analysis as suggested. Here, we experimented with including varying time periods for the environmental data, similar to the original analysis. However, the results revealed no significant new insights gained from including these time spans, which is why we did not include them in the final analysis. Instead, we focused on direct influences of the on-site environmental measurements on daily rates of stem change, thus including stem shrinking and swelling processes.
We also made sure, to indicate more clearly, if daily or annually aggregated values were used in the analysis.

Discussion:
"Within the introduction a nice structure is presented, namely: "1) explain major growth patterns and their variation between years and specimens, 2) identify the most important on-site environmental drivers controlling these patterns, and 3) gain insights into potential response to environmental change". I was expecting a similar structure to become apparent in the discussion. However, within the discussion a more parameter centred structure is utilized. I am wondering whether it then would be clearer for the reader to add an initial section where these key points are shortly addressed before diving into each parameters. Then one should also introduce the discussion structure.

The main objective is clearly defined: "The main objective of our work is thus to gain detailed understanding of the growth patterns of one common arctic-alpine dwarf shrub (Empetrum nigrum ssp. hermaphroditum) and their relation to the micro-environment". However, within the discussion ample attention is provided to the distinct differences between macro- and micro-environmental conditions which boils down to the conclusion that topography is crucial in determining growth responses. Yet, due to the sampling design (solely 1 sample per site) and presented analyses, I wonder whether this is the strongest result one can present from this study. I find the discussion on temporal dynamics of the growth responses to environmental conditions more relevant."

"In relation to the discussion on the lower relevance of atmospheric air temperature on growth, I do not find the response of the authors satisfying. First the authors state: "A direct comparison of annual growth derived from our dendrometer measurements and such ring width measurements at the studied sites revealed high synchrony. Here, the ring width data was linked to summer temperature as well, suggesting that the assumed temperature-growth relation holds partly true at our sampled sites". So if this is a clear observation, then why are the author so convinced that there is no direct connection between total growth and near-surface temperature. Multiple mechanistic studies show the relevance of temperature on enzymatic kinetics within the cambium (i.e., see temperature module in the Cabon et al. 2020 model). Moreover, there is no discussion on the fact that the difference might also be caused by solely including four years of dendrometer data, or due to the fact the dendrometers incorporate both the production of xylem and phloem cells, while tree-ring studies only consider the xylem. Without such careful considerations I tend to agree with the previous concerns raised by referee 1."

"More general I miss an overarching discussion on the limitations of the study. I think this is particularly relevant as I agree with referee 3 that this should be considered as a case study highlighting the potential of using dendrometer data on shrubs. Referee 3 for instance makes a strong point with the fact that VPD is a critical environmental factor which should be studied in the future (see Novick et al., 2016; Grossiord et al., 2018; Peters et al., 2021; Zweifel et al., 2021). Also, referee 3 makes a good point on the fact that the Gompertz fitting has its limitation, while other methods to extract the start and end of the growing season do exist (see Knüsel et al. 2021 Forests: https://doi.org/10.3390/f12060765). Some of the concerns raised by 1 referee are also valid. For example, 1 dendrometer per site/elevation is a limitation which prevented the analyses on the impact of elevation on growth parameters. Moreover, the fact that only one stem is measured per shrub does generate the question on how large the within plant variability is and how this would impact the results. All such careful considerations should be mentioned in a dedicated section, where clear recommendations should be presented to guide future research efforts."

"Finally, I would refrain from using references within the conclusion and just highlight the most important findings and considerations."

While revising the Discussion with the remarks on structure in mind, we came to the conclusion that the previous structure with the emphasis on parameters might not be the best choice, as several aspects were coming up multiple times in the text and some parts were therefore not as clear as they could be. We therefore revised the structure of the Discussion entirely, keeping the main aims introduced in the Introduction in mind. With this revised structure, an additional Conclusion chapter seemed unnecessary. We therefore removed it and incorporated the aspects from the previous Conclusion in the Discussion section.

Also, we put more emphasis on temporal dynamics and the complex temperature- and soil moisture-relations found in our data, showing that while we found no evidence for highly influential temperature thresholds, temperatures still played a role in determining growth patterns, but this role is more complex than previously assumed.

Regarding the methodical concerns, we included some of these considerations in the Discussion and in the Material and Methods chapter. Also, we made sure to phrase the implications and possibilities for future research more clearly, and included some additional statements, pointing out the limitations of the approach.

---

## Author Response (AR3)

Comments to the author:
Dear Dr. Löffler and co-authors,

I have now received the report from an additional reviewer, given that the previous reviewers indicated that they did not want to re-assess your manuscript. The reviewer points out that your paper needs some further improvement and makes some excellent suggestions, which should help improve the clarity of your paper. Please
1) explain more clearly how the statistical analyses relate to the specific questions asked,
2) structure the discussion around the main questions posed in the introduction,
3) outline more prominently the limitations of the study,
4) use consistent terminology throughout the manuscript and
5) take into account the additional specific comments of the reviewer.

I look forward to seeing your revisions. Given that the review process of your paper has been exceptionally lengthy I point out that the revised manuscript will be reconsidered only if your revisions address the issues raised in a profound and conclusive manner.

Best regards,
Michael Bahn

Many thanks to the reviewer and the editor for their detailed comments and suggestions, which helped a lot to further improve the manuscript.
In accordance with the suggestions, we again revised the manuscript substantially. Specifically, we 1) improved on some methods used and rewrote the text on the statistical analyses in the Material and Methods section. Furthermore, we 2) restructured the Discussion around the main questions posed in the introduction and 3) included a paragraph on the limitations of the study. We additionally revised 4) the terminology throughout the manuscript, and 5) addressed the detailed comments by the reviewer.

**Review for Dobbert et al. The application of dendrometers to alpine dwarf shrubs – a case study to investigate stem growth responses to environmental conditions**

The manuscript of Dobbert et al. deals with intra-annual growth dynamics of alpine dwarf shrubs obtained by dendrometer measurements. The applied methodology is new since dendrometer measurements on shrub species have not been conducted so far. The topic is important since growth dynamics at the alpine treeline are changing due to climate warming. The manuscript addresses the eco-physiological mechanisms of growth which might help to understand dynamics that are observed at the alpine treeline.

The manuscript has received already a lot of comments through two rounds of major revisions, which have been addressed in the revised version and should be

acknowledged. However, I still feel that improvements could be made, particularly in relation to the methodological description and discussion.

Methods: I have spent some time trying to understand what kind of statistical analyses have been made, and why. However, I could not fully reconstruct what was done in detail, since analyses are made with a lot of different response variables and explaining factors, and on different time scales (annual values, daily values). For example, I did not understand why a multiple regression analysis was done and afterwards a prediction from a linear regression model. Further, why didn't you use a mixed effect model with site as random factor since all sites have been measured four times. It would be helpful to clearer describe the statistical methods and what kind of research question should be answered with the respective analysis. See also detailed comments.

We agree that the Methods section (especially chapter *2.5 Climate-growth relations and potential drivers of radial stem change,* on the statistical analysis) needed some clarifying. As suggested, we now included linear mixed effect models with site as random effect (see also our response to the detailed comments below). In general, our statistical analysis followed the following structure:

1. Identification of growth defining parameters in order to assess major growth patterns and their variation between years and specimens. We here used Gompertz models to accurately define critical dates of growth. Subsequently, we used linear mixed effect models (in the revised version) to assess the relation of these growth-defining parameters and their importance for total annual growth.
2. Linking the previously defined annual parameters to environmental parameters derived from raw environmental measurements. This was done using a simple correlation analysis, with the aim to identify the most important environmental drivers controlling growth patterns.
3. Examining intra-annual patterns of stem diameter shrinking and swelling in response to fine-scale, local environmental conditions (daily values). Here, we performed moving window correlations.

To convey this governing structure more clearly to the reader, we substantially revised the Material and Methods chapter. In chapter *2.5 Climate-growth relations and potential drivers of radial stem change*, we restructured the text according to the three main methods deployed and made sure to point out the aim of the respective analysis in relation to the research questions introduced in the Introduction.

Discussion: You define three research questions at the end of the Introduction. I think it would help the reader to build-up the discussion along these three main questions, or at least summarize in the beginning of the discussion the main findings of the study, which will be discussed then. At the moment I have the feeling that the discussion is not well structured and is not based on the findings that were presented in the results section. For example, you start the discussion with a topic that was even not shown in the results (zero-growth years), and you refer to new Figures in the Supplement. I would recommend to discuss the findings that were presented in the results section.

Also, I have the impression that patterns of stem growth and stem diameter changes (including stem water relations) are sometimes mixed-up, especially within the discussion.

In accordance with your detailed suggestions we restructured the Discussion section entirely. After a short paragraph focusing on the methodical implication of the study we now present three subchapters, each focusing on one of the questions posed within the Introduction. Also, we moved all parts presenting additional results from the supplementary material to the Results section and made sure that all findings prominently discussed in the Discussion are also clearly mentioned in the Results. Additionally, we paid special attention to our use of the terms stem growth and stem diameter changes (as well as similar terms).

Further, I think the limitations of the study as nicely summarised by the previous reviewer, should be placed more prominent within the discussion.

After restructuring the Discussion chapter, we now start with a clearer focus on the methodical implications of our study and included the limitations in a separate paragraph at the beginning of the Discussion. Here, we discuss the limitations mentioned by the previous reviewer, including the challenge of extracting physiologically meaningful data, potential intra-plant variation, and the potential of including VPD as a critical environmental factor.

Finally, terms should be used consistently, e.g. you use various different terms for stem diameter variations (stem diameter change, stem diameter data, stem change, stem diameter variability /variation, annual stem variability)

As suggested, we refrained from using multiple terms here and consistently use "stem diameter variability" to refer to the measured stem diameter variations. However, when we refer to the total change in stem diameter from the start to the end of the year (stem diameter at the end of the year - stem diameter at the beginning of the year) we consistently use the term "stem diameter change" to differentiate the two.

Detailed comments:

L16: can you explain what growth-defining parameters are? Do you refer to phenology?

We here referred to the parameters used throughout the study to define radial stem growth. They are defined in chapter *2.3 Analysis of seasonal growth patterns,* in the Material and Methods section. As this might not be clear to the reader while reading the Abstract, we use the term "radial stem growth" in the revised version.

L16: response of what?

To clarify, we changed this sentence. It now reads:

> "We found high inter-plant variability in overall radial stem growth, but strong similarities in response patterns to the local environment.".

L18: remain

We believe "remains" is the grammatically correct term here, as we refer to a singular evergreen species (*E. hermaphroditum*).

L18: Why can we conclude this from a dependency of growth start to winter temperature and ground freezing?

We reworded this part of the Abstract as it might not be clear in the previous form. It now reads as follows:

> "Our results suggest that the evergreen species is highly adapted to the specific local conditions, remaining partly photosynthetically active during the snow-free winter, which facilitates carbohydrate accumulation for early-season physiological activities."

The conclusion that *E. hermaphroditum* might stay partly photosynthetically active during the winter months stems from a combination of several aspects of our results: Due to the snow free conditions at the monitored ridge positions, the evergreen species is faced with unique environmental conditions, receiving continued solar radiation which might force photosynthetic activity. At the same time, we found the species capable of avoiding negative effects of extreme temperatures through a process of radial stem shrinking, and an early growth start was linked to low winter temperatures, suggesting *E. hermaphroditum* is adapted to profit from cold winters with associated high solar radiation. This is discussed in detail in the Discussion section.

L23: Can you be more precise? What does this finding mean for the debate on greening and browning?

As detailed in the final paragraph of the Discussion, our findings regarding drivers of shrub growth have important implications for future distribution and potential spread/decline of the species, which in turn influences the processes of arctic greening and browning. We therefore included this in the Abstract:

> "We identified soil moisture availability and winter freezing conditions as the main drivers of stem diameter variability, which might negatively affect the species distribution in a warming climate, thus forwarding the ongoing debate on the functional mechanisms and complexity of vegetational shifts in arctic and alpine regions. "

This also ties back to the first sentence of the Abstract.

Methods:
L138: I don't see this in Fig. A3
Fig. A3 (a) illustrates the dendrometer set up as described here, showing that the dendrometer was not installed inside the radius of other larger shrub species or near large stones. Because of the complex set up and vegetation in the field it was challenging to show a good photo here.

L143: can you explain briefly the daily mean approach? Why didn't you use the daily maximum?

Originally, we tested both approaches for our data, calculating daily mean and maximum values (calculated from 24 hourly measurements for each day, from 0:00 to 23:00). For the purpose presented here (annual growth patterns), we found the results highly similar, which is in accordance with results by Deslauriers et al. (2007). They found differences in both approaches mainly in daily amplitude, which was not of interest for our study. Consequently, we chose the daily mean approach, believing it to be the simpler of the two. Yet, considering the zero growth approach deployed later in our study for calculating total annual growth, the daily maximum approach might be slightly more appropriate here, which is why we changed this in the revised manuscript and added a short explanatory sentence in the text. As evident from Figure 1, this change in methodology resulted in only minor changes in the resulting curves.

L173: I do not understand why you derived the dates from the modelled curve and not from your original (growth) data?

Using sigmoid models to determine growth start and cessation has proven successful in a number of past studies, especially, since these models can be biologically interpreted (e.g. Duchesne et al., 2012; Van der Maaten et al., 2018; Liu et al., 2019). Due to the additional measuring of reversible stem hydraulic dynamics by dendrometers, raw dendrometer data are known to deliver rough estimates of cambial activity and less reliable critical dates like onset and cessation of growth (Deslauriers et al., 2007; van der Maaten et al., 2018; Cruz-García et al., 2019). This is especially true for our measurements of shrub growth, because, in comparison to trees, they showed a less clearly defined growing season, with a phase of stem shrinking during the winter months, and consequent stem expansion in spring. The chosen modelling approach therefore proved the most reliable to clearly define the main growing season, derive the critical dates and ensure high comparability between years and specimens. We included this statement in the text.

L174: Why did you chose a threshold of 20% as growth start, this is quite a lot. How was the 'careful testing' done and which criteria did you use to validate your threshold. In Fig. 1 it looks like that the growth season starts when the growth rate is already close to its maximum.

Our testing was mainly done visually by direct comparison of raw environmental data (for each specimen) and the resulting growing season. One of our main concerns here was to make sure that the defined growing season does not include radial stem expansion resulting from the stem shrinking observed during the winter months and probably linked to stem water dynamics, which is why we initially chose a higher threshold for growth start. However, since the difference between a 10 % and a 20 % threshold was minor, we changed it in the revised version. In Figure 1, we included lines to aid the visual interpretation of the figure.

L216: can you give the response variable (annual growth and peak shrinking according to Table 1) and the independent variables? In table 1 only 3 independent variables are given, in Line 219 you write six measures, please clarify.

In Line 219 we refer to the six measures of collinearity, which are implemented in the mctest package (Imdad and Aslam, 2018), and which we deployed to assess collinearity between independent variables. We revised this entire chapter and performed linear mixed effect models, replacing the linear regression (see below).

L223: Is it correct that you have N=12*4=48 for the correlations? Why didn't you use mixed models with sites as random effects? How did you account for the site effect in your analysis?
This is correct. We agree that mixed models might be a better fit here and replaced this analysis in the revised version. We now used linear mixed effect models with site (including study region and elevation) as random effects. The results are presented in Table 1.

L223-25: I do not understand what was done here. What was the research question for this analysis and what is the difference to the correlation analysis that was done before?
The results of this analysis (formerly presented in Fig. 4) is indeed to some extent redundant with the results from the correlation analysis. We therefore excluded it in the revised version.

L224: What window size did you chose and did you test for the effect of the chosen size?
We did indeed test several values for window widths (presented in an earlier version of this manuscript) and finally chose 3 days, which showed the overall most significant results. We included this in the revised text.

Results:
L238: Since you do not describe solely growth pattern, I would use stem diameter variation instead
We chose to use "stem diameter variability" here (see above).

L244: What do you mean by timing of shrinking period? Is it the start or the peak or end?
The start of the shrinking period. We included this in the text.

L259: What do you mean with total growth pattern? Did you test these statements statistically? I do not think that year-to-year variability is similar between the two regions. In region 1, the year 2017 has lowest growth, while the other three years are comparable, in region 2, growth is exceptional in 2016, while it is low in the three other years.
We do agree that this statement should be more specific. We changed it to "...growth patterns and timing of the growing season were similar in the two studied regions", which is more accurate and shown in Figure A4 (e.g. late growth start in 2015). Here it is also important to keep in mind that Fig. A4 shows averaged curves while variation between specimens (indicated by standard deviation in the figure) in general higher than variation between regions, as indicated in the text.

L261: but here (Fig. A1) you do not look at growth patterns, but at stem diameter variation.

We do not refer to Figure A1 in this line. Instead we here refer to Figure A4. We agree, however, that we do look at stem diameter variation, rather than growth. We therefore changed the term here to " patterns of stem diameter variability".

Fig. A4 and A6: It looks like
West and East is swapped (A4: generally higher annual growth for West, and high increment in 2016 for East, in A6 the other way around).

Thank you very much for pointing that out. We revised Fig. A4 accordingly.

Table 3: what do you mean by standard deviation , +/- SE: SE is the abbreviation for standard error, what was calculated? I do not think that these values are meaningful, because they are dependent on the mean. Better would be to calculate e.g., coefficient of variation, if you are interested in the variability of the measure.

Originally, we included the standard error (SE) here, as a measure of variation. However, we do agree that the coefficient of variation (CV) is the better measure here. We therefore calculated it in the revised version to show variability between sampled specimens. CV is now included for Table 1,3, and 4.

L306-308: How was this analysed and where are the results presented? Isn't the model overfitted with 25 explaining variables and only 48 observations?

We agree. The model was initially created to check the overall reliability of our chosen parameters and is complementing other results. As it is not prominently discussed in the discussion section we removed this part from the text.

L310: The moving window analysis does not present climate-growth relations, since you use stem diameter changes and not only growth. This measure has a totally different dynamic as also stem water relations are included.

We agree that climate-growth relations might be the wrong term here. We changed it to "the relation of stem diameter variability and environmental conditions", which is more accurate and in consistence with the terms used throughout the text.

Fig. 4 and 6: What is the difference of the results presented here? Why is annual growth related to radiation and soil moisture in Fig.4, and to soil moisture maximum in Fig. 6, and not at all to radiation?

The analysis presented in Fig. 4 does include different parameters from the correlations presented in Fig. 6. However, the multiple regression analysis is not very robust compared to the correlation coefficients, and the results to some extent redundant. We therefore removed Fig. 4. We revised the text (Methods, Results, and Discussion) accordingly.

L290-300: In my opinion this belongs to discussion

We moved some of the aspects presented in this paragraph to the Discussion section.

Discussion:
L351: Do you refer here to the soil moisture shown in Fig. A1b? But these measures vary between sites and regions. In Fig. 2 it is not visible whether there are changes between sites /regions.
Here we referred to the environmental conditions characterising the exposed ridge positions, described in the previous sentences, rather than variation between sites/regions. For example, all of our study sites in both study regions and along the elevational gradient are experiencing discontinuous snow cover and high solar radiation, compared to their immediate surroundings. As we revised the Discussion, we placed less emphasis on this aspect and also made sure to include it more prominently in chapter *2.1 Study sites.*

L353: But the growth curves of the two regions look totally different at least for the years 2015 and 2018!
We here referred to seasonal growth patterns (including timing of the growing season and shrinking period in winter). However, since this is not completely clear from the graphs, we removed this part from the Discussion.

L354: but which was not tested in your study as you did not include regional climate data. If you argue like that it would be good to include the chain of arguments developed in line 290-300 here.
Since this aspect seemed to be not entirely clear from the presented results, we decided to remove it in the revised version. We did, however, include some of the aspects from line 290-300 later in the text.

L361-369: If discussed so prominently, these data should be presented in the results section.
We included these results in the Results section and discussed them less prominently in the restructured version of the Discussion.

L362: how did you link the zero growth to snow coverage periods and where can this analysis be seen?
This is presented in Fig. A5 (formerly Fig. A9). We moved this to the Results section.

L466: With which analysis can this statement be confirmed?
These results are from the correlation analysis. We revised this part, directly linking it to the results from the year 2016 (high number of days with temperatures > 5°C resulting in high growth). Also, the positive effects of winter temperatures shown in the correlation analysis are discussed earlier in the text.

---

## Author Response (AR4)

Dear authors,

I am pleased to inform you that your revised manuscript will be considered for publication in Biogeosciences after you have satisfactorily addressed the excellent and very constructive comments provided by the reviewer.

When revising your manuscript please make sure that throughout the manuscript, including the Tables and Figures (see e.g. Fig. 1-2 and Fig. 5, Table 4), all units are written exponentially (e.g. W m–2) - see also
https://www.biogeosciences.net/submission.html#figurestables.
Furthermore I ask you to add y-axis titles to Fig. 2 and, for better visibility, to increase line thickness of significant correlation coefficients in Fig. 4.
Finally, could you please rephrase the last sentence of your abstract to make it a clearcut and straightforward conclusion.

Best regards,
Michael Bahn

Thank you very much for the additional comments on our manuscript. We made sure that all units are written exponentially and added axis titles to Fig. 2. Furthermore, we increased line thickness in Fig. 4 and rephrased the last sentence of our abstract as suggested. It now reads:
"Our results highlight the role of water availability in controlling vegetational shifts in arctic and alpine regions, with soil moisture and winter snow conditions as the main drivers of radial stem growth, negatively affecting the species distribution in a warming climate."
Additionally, we addressed the questions and comments of the reviewer (see below).

The authors have substantially revised the ms, addressed all suggestions and overall, I think the ms has much improved. The discussion has been completely restructured and reads now much clearer, and includes also a discussion on the limitations. The description of the methods has also been improved and methods have been adapted. However, I still have some questions regarding the methods, especially the correlation analysis and the moving window analysis, but which probably can easily be answered.
Thank you very much. We addressed your questions (see below), resulting in minor changes in the manuscript.

Detailed comments:
L213: Table 1
Thank you, we meant to refer to Table 1 here.

L226: Do you mean study sites? ('measuring sites`)
Yes, we changed "measuring sites" to "study sites".

L231: I think also for this analysis a mixed effects model would be the appropriate approach.

Thank you very much for the suggestion. We agree that mixed effects modelling is a more appropriate approach here, since variation between sites can be included into the model as a random effect. Accordingly, we now present fixed effects derived from linear mixed effects models in a revised version of Fig. 5.

Comparing the results to the previous correlation analysis, we found that the mixed models mostly confirmed the previous results. In fact, some of the main aspects discussed in the Discussion chapter are more strongly reflected in the results from the mixed models, compared to the correlation analysis. The models revealed for example an influence of the timing of soil thawing (calculated from our environmental data as the first day, at which soil moisture > 0.15) on total annual growth, with earlier soil thawing leading to overall higher growth. We discuss this dependency on freezing and thawing processes in chapter 4.2 Environmental controls of shrub growth in the Discussion section.

In accordance with the new results we made some minor changes in the text, especially in chapter 3.3 Environmental controls of stem diameter variability of the Results section.

L231: Table 3: In Table 3 no micro-environmental drivers, nor the results of the correlation analysis are presented, therefore I think this reference is not placed well.
We meant to refer to Table 1 here. We included the reference here because the micro-environmental drivers which entered into the analysis are summarised in Table 1. We changed the text to clarify: "…from the potential micro-environmental drivers presented in Table 1"

L239: Is it correct that you use a window size of only 3 days or is this a typo? I find this quite small, you have then a sample size of N=3 per window, which is too small to do a correlation analysis? Probably I misunderstand something. I also did not find information how you did account for the different specimen and sites? Did you perform a moving window analysis for each specimen, and average the results in the end, which I think would be the right approach to do, or did you pool everything together?
It is true that the moving window size for this analysis is 3 days. We agree that this might seem very short. We chose this size after carefully testing window widths of up to six months (as mentioned in the manuscript). Overall, window width had surprisingly little effects on the results. To show that in the revised manuscript, we included results for a window width of 30 days (dashed lines) for direct comparison. We changed the text in the Material and Methods section and the figure legend accordingly. We did perform a moving window analysis for each specimen and averaged the results as described.

Table 1: Abbreviations for Soil moisture and Global radiation is missing
Thank you, we included the abbreviations in the table.

Table 2: Dependant = dependent. Why is the pearson's correlation coefficient included in the Table, this does not belong to the mixed model. Either remove or specify in the Table caption.
Does the Model R2 include both the random and the fixed effects? Can you also give the partial R2 of the random and fixed effects (e.g. with MuMin:: r.squaredGLMM(model)

Changed "Dependent". We agree that the Pearson's correlation coefficient is not necessary here and removed it from the table.
The Model R2 was indeed calculated using the r.squaredGLMM-function and thus represents the variance explained by the entire model (conditional R2). We now added the Partial R2 of the fixed effects (marginal R2) as suggested.

Figure 1: I don't understand what the black bars should indicate. Is this spring and fall?
Yes, they indicate meteorological seasons as specified in the figure legend.

L322: Negative relation to temperature in spring is really unexpected. I think it would be worse to discuss this finding.
The negative relation to temperature in spring is possibly linked to the species vulnerability to cold snaps (as described in Weijers et al., 2018). However, our findings were not very strong (see Fig. 2) and are also not confirmed by the mixed effects models, which is why we refrained from discussing this aspect and rather removed it here.

L327: which threshold?
The 5 °C-threshold. We changed the sentence to make this more clear ("...temperatures rising quickly and steadily to 5 °C…").

L328: How did you determine the upper range of 10°C?
We included both the number of days with temperatures rising above 5 °C, as well as the number of days with temperatures rising above 10 °C into the analysis (growing degree days, GDD5 and GDD10). As we observed the best growing conditions in 2016, the year with the highest number of days with temperatures above 5 °C, but no clear link to GDD10 (for example in 2018, which had an exceptionally high number of days with temperatures above 10 °C), we came to the conclusion presented in the text:
"During 2016, the year with highest radial stem growth, we also measured the highest number of days with soil temperatures above 5 °C, with temperatures rising quickly and steadily to that threshold without reaching continuously higher values during the summer".

L332: This is a suggestion and belongs to the discussion
This is indeed discussed in the discussion section (chapter 4.2 Environmental controls of shrub growth). We therefore removed it here.

Fig. 4: Did I interpret it correctly that the black bars in Fig.4 and Fig. 1 are opposite (black bar is spring and fall in Fig. 1 and summer in Fig. 4)? I would suggest to make this consistent.
Yes, thank you, we made this consistent.

---

## Author Response (AR5)

Dear Dr. Löffler and co-authors,

please make some final revisions as follows:

1) The revised conclusion of the abstract is not appropriate as you did not study vegetational shifts. Please rewrite your conclusion once more, e.g. "We conclude that soil moisture availability and winter snow conditions are the main drivers of radial stem growth of the studied species and could negatively affect the species' distribution in a warming climate."

2) Lines in Fig. 4 are still not very clearly visible, please increase line thickness further. In legend of Fig. 4 indicate which lines correspond to the moving windows of 3 and 30 days, respectively.

Best regards,

Michael Bahn

Thank you very much for the final suggestions. We revised the conclusion of the abstract as suggested, increased line thickness in figure 4, and revised the figure legend, including the information on the linetypes.